# FEDERATED EQUILIBRIUM SOLUTIONS FOR GENERALIZED METHOD OF MOMENTS APPLIED TO INSTRUMENTAL VARIABLE ANALYSIS

## ABSTRACT

Instrumental variables (IV) analysis is an important applied tool for areas such as healthcare and consumer economics. For IV analysis in high-dimensional settings, the Generalized Method of Moments (GMM) using deep neural networks offer an efficient approach. With non-i.i.d. data sourced from scattered decentralized clients, federated learning is a popular paradigm for training the models while promising data privacy. However, to our knowledge, no federated algorithm for either GMM or IV analysis exists to date. In this work, we introduce federated IV analysis (FEDIV) via federated GMM (FEDGMM). We formulate FEDGMM as a federated zero-sum game defined by a non-convex non-concave minimax optimization problem. We characterize the solutions to the federated game using Stackelberg equilibrium and show that it satisfies client-local equilibria up to a heterogeneity bias. Thereby, we show that the consistency of federated GMM estimator across clients closely depends on the heterogeneity bias. Our experiments demonstrate that the federated framework for IV analysis efficiently recover the consistent GMM estimators for low and high-dimensional data.

## 1 INTRODUCTION

Federated Learning (FL) (McMahan et al., 2017) is now an established paradigm for training Machine Learning (ML) models over decentralized clients, keeping the data local and private. The applications include important domains such as healthcare (Oh & Nadkarni, 2023), finance & banking (Long et al., 2020), smart cities & mobility (Gecer & Garbinato, 2024), and many others (Ye et al., 2023). The scale of FL has also grown large – see the Nature Medicine report by Dayan et al. (2021) on a global-scale FL to predict the effectiveness of oxygen administration to COVID-19 patients in the emergency rooms while maintaining data locality. However, the current popular FL methods have a crucial limitation due to their standard supervised nature of learning. For example, Liang et al. (2023) suggests that the hypoxia-inducible factors (HIF) (a protein that controls the rate of transcription of genetic information from DNA to messenger RNA by binding to a specific DNA sequence) play a vital role in oxygen consumption at the cellular level. Arguably, the Dayan et al. (2021)'s approach may over- or under-estimate the effects of oxygen treatment as it does not accommodate the influence of HIF levels on oxygen consumption.

A classical approach to address such limitations is Instrumental variables (IV) analysis, which assumes conditional independence between a confounding variable and the outcome while considering its causal effect on the treatment variable. IV analysis can very practically apply to training Dayan et al. (2021)'s ML model wherein the patients' HIF levels work as an IV that influences the effective organ-level oxygen consumption (a treatment variable) but does not directly affect the mortality of the COVID-19 patients (the outcome). IV analysis has been comprehensively explored in econometrics (Angrist & Krueger, 2001; Angrist & Pischke, 2009) with several decades of history, such as works of Wright (1928) and Reiersøl (1945). Its efficiency is now accepted for learning even high-dimensional complex causal relationships, such as those in image datasets (Hartford et al., 2017; Bennett et al., 2019). Naturally, the growing demand for FL entails designing methods for federated IV analysis, which, to our knowledge, is yet unexplored.

In the centralized deep learning setting, Hartford et al. (2017) introduced an IV analysis framework, namely DEEPIV, which uses two stages of neural networks (NN) training – learn the conditional treatment distribution as a NN-parametrized Gaussian mixture for the treatment prediction and then train the outcome model. The two-stage process has precursors in applying the least square regressions in the two phases (Angrist & Pischke, 2009)[4.1.1]. In the same setting, another approach for IV analysis applies the generalized method of moments (GMM) (Wooldridge, 2001). GMM is a celebrated estimation approach in social sciences and economics. It was introduced by Hansen (1982), for which he won a Nobel Prize in Economics (Steif et al., 2014).

Lewis & Syrgkanis (2018) employed neural networks for GMM estimation. Their method, called the adversarial generalized method of moments (AGMM) fit a GMM criterion function over a finite set of unconditional moments. Similarly, Bennett et al. (2019) introduced deep learning models to GMM estimation; they named their method DEEPGMM. DEEPGMM differs from AGMM in using a weighted norm to define the objective function. The experiments in (Bennett et al., 2019) showed that DEEPGMM outperformed AGMM for IV analysis, and both won against DEEPIV. Nonetheless, to our knowledge, none of these methods has a federated counterpart. Notably, both AGMM and DEEPGMM translate to a minimax optimization problem corresponding to a smooth zero-sum game.

The zero-sum game formulated for GMM estimation is essentially nonconvex-nonconcave (Bennett et al., 2019). Such a game corresponds to a sequential game as it may have differing maximin and minimax solutions. Unlike Nash equilibrium, the global minimax points – the Stackelberg equilibria – are guaranteed to exist for nonconvex-nonconcave games. However, finding a global minimax point is generally NP-hard, necessitating solving for a surrogate local equilibrium (Jin et al., 2020).

Now, considering the federated version of this problem, a fundamental challenge arises in establishing that a federated minimax optimization algorithm retrieves a local Stackelberg equilibrium of the federated zero-sum game. Even if it did, it requires showing that the federated equilibrium translates to the client-local setting under heterogeneity. Finally, it entails proving that the client-local equilibrium under heterogeneity is a consistent GMM estimator for its data. In this work, we address these challenges. Our contributions are summarized as follows:

1. We introduce **FEDIV**: federated IV analysis. To our knowledge, **FEDIV** is the first work on IV analysis in a federated setting.
2. We present **FEDDEEPGMM**[1] – a federated adaptation of DEEPGMM of Bennett et al. (2019) to solve FEDIV. FEDDEEPGMM is implemented as a federated smooth zero-sum game.
3. We characterize an **approximate local equilibrium solution for federated zero-sum game**. We show that the limit points of a federated gradient descent ascent (FEDGDA) algorithm include the equilibria of the zero-sum game.
4. We show that an equilibrium solution of the federated game obtained at the server consistently estimates the **moment conditions of every client**. An important insight derived from our results is that the consistency of the GMM-estimators on clients directly depend on the heterogeneity bias.
5. We experimentally validate that even for non-i.i.d. data, FEDDEEPGMM has convergent dynamics analogous to the centralized DEEPGMM algorithm.

This work focuses on the existence results of federated equilibrium solutions, federated consistent GMM estimators, and thereby structurally solving the federated IV analysis problem. The existence of approximate client-local equilibria via federated solution has applications beyond the GMM and IV analysis, to problems such as federated generative adversarial networks (FedGAN) (Rasouli et al., 2020), where a Nash Equilibrium may not exist (Farnia & Ozdaglar, 2020). However, the scope of our discussion does not include FedGAN, or a new federated minimax algorithm, or, for that matter, the convergence theory and scalability. We leave an open problem to characterize and recover a federated mixed-strategy Nash equilibrium, which has enormous applications to diverse domains (Barron, 2024). We compare and contrast our method against related works in Appendix D.

## 2 PRELIMINARIES AND MODEL

In this section, we introduce our basic terminologies tailored to a motivating application as described by Dayan et al. (2021) in the context of the global-scale federated learning.

---

[1]Wu et al. (2023) used FEDGMM as an acronym for federated Gaussian mixture models.

**Client-local causal inference.** We begin by adapting (Bennett et al., 2019) for a client-local setting. Consider a distributed system as a set of $N$ clients $[N]$ with datasets $S^i = \{(x^i_j, y^i_j)\}_{j=1}^{n_i}, \forall i \in [N]$. We assume that for a client $i \in [N]$, the treatment and outcome variables $x^i_j$ and $y^i_j$, respectively, are related by the process $Y^i = g^i_0(X^i) + \epsilon^i, \ i \in [N]$.

Referring to (Dayan et al., 2021), $x^i_j$ and $y^i_j$ represent the clinical features (CBC, D-dimer, oxygen devices, etc.) and the outcome (death, duration of survival over oxygen administration, etc.), respectively, on a client $i \in [N]$.

We assume that each client-local residual $\epsilon^i$ has zero mean and finite variance, i.e. $\mathbb{E}[\epsilon^i] = 0, \mathbb{E}[(\epsilon^i)^2] < \infty$. Furthermore, we assume that the treatment variables $X^i$ are endogenous on the clients, i.e. $\mathbb{E}[\epsilon^i|X^i] \neq 0$, and therefore, $g^i_0(X^i) \neq \mathbb{E}[Y^i|X^i]$. We assume that the treatment variables are influenced by instrumental variables $Z^i, \forall i \in [N]$ so that

$$P(X^i|Z^i) \neq P(X^i). \tag{1}$$

Furthermore, the instrumental variables do not directly influence the outcome variables $Y^i, \forall i \in [N]$:

$$\mathbb{E}[\epsilon^i|Z^i] = 0. \tag{2}$$

Referring to (Dayan et al., 2021), $Z^i$ would represent the HIF levels in the patients on the client $i \in [N]$, which their implementation misses to incorporate.

**Federated causal inference function.** Note that, assumptions 1, 2 are local to the clients, thus, honour the data-privacy requirements of a federated learning task. In this setting, we aim to discover a common or global causal response function that would *fit the data generation processes of each client without centralizing the data*. More specifically, we learn a parametric function $g_0(.) \in G := \{g(., \theta)|\theta \in \Theta\}$ expressed as $g_0 := g(., \theta_0)$ for $\theta_0 \in \Theta$, defined by

$$g(., \theta_0) = \frac{1}{N} \sum_{i=1}^{N} g^i(., \theta_0). \tag{3}$$

**The generalized method of moments (GMM)** estimates the parameters of the causal response function (3) using a certain number of *moment conditions*. Define the *moment function* on a client $i \in [N]$ as a vector-valued function $f^i : \mathbb{R}^{|\mathcal{Z}|} \to \mathbb{R}^m$ with components $f^i_1, f^i_2, \ldots, f^i_m$. Based on equation (2), we define the moment conditions using parametrized functions $\{f^i_j\}_{j=1}^m \ \forall i \in [N]$ as

$$\mathbb{E}[f^i_j(Z^i)\epsilon^i] = 0, \forall j \in [m], \ \forall i \in [N], \tag{4}$$

We assume that $m$ moment conditions $\{f^i_j\}_{j=1}^m$ at each client $i \in [N]$ are sufficient to identify a unique federated estimate $\hat{\theta}$ to $\theta_0$. With (4), we define the moment conditions on a client $i \in [N]$ as

$$\psi(f^i_j; \theta) = 0, \ \forall j \in [m], \tag{5}$$

where $\psi(f^i; \theta) := \mathbb{E}[f^i(Z^i)\epsilon^i] = \mathbb{E}[f^i(Z^i)(Y^i - g^i(X^i; \theta))]$. In empirical terms, the sample moments for the $i$-th client with $n_i$ samples are given by

$$\psi_{n_i}(f^i; \theta) := \mathbb{E}_{n_i}[f^i(Z)\epsilon^i] = \frac{1}{n_i} \sum_{k=1}^{n_i} f^i(Z^i_k)(Y^i_k - g^i(X^i_k; \theta)), \tag{6}$$

where $\psi_{n_i}(f^i; \theta) = \left(\psi_{n_i}(f^i_1; \theta), \psi_{n_i}(f^i_2; \theta), \ldots, \psi_{n_i}(f^i_m; \theta)\right)$ is the moment condition vector, and $\psi_{n_i}(f^i_j; \theta) = \frac{1}{n_i} \sum_{k=1}^{n_i} f^i_j(Z^i_k)(Y^i_k - g^i(X^i_k; \theta))$. Thus, for empirical estimation of the causal response function $g^i_0$ at client $i \in [N]$, it needs to satisfy

$$\psi_{n_i}(f^i_j; \theta_0) = 0, \ \forall i \in [N] \text{ and } j \in [m] \text{ at } \theta = \theta_0. \tag{7}$$

**The optimization problem.** Equation (7) is reformulated as an optimization problem given by

$$\min_{\theta \in \Theta} \|\psi_{n_i}(f^i_1; \theta), \psi_{n_i}(f^i_2; \theta), \ldots, \psi_{n_i}(f^i_m; \theta)\|^2, \tag{8}$$

where we use the Euclidean norm $\|w\|^2 = w^T w$. Drawing inspiration from Hansen (1982), DEEP-GMM used a weighted norm, which yields minimal asymptotic variance for a consistent estimator

$\tilde{\theta}$, to cater to the cases of (finitely) large number of moment conditions. We adapt their weighted norm $\|w\|_{\tilde{\theta}}^2 = w^T \mathcal{C}_{\tilde{\theta}}^{-1} w$, to a client-local setting via the positive semi-definite covariance matrix $\mathcal{C}_{\tilde{\theta}}$ defined by

$$[\mathcal{C}_{\tilde{\theta}}]_{jl} = \frac{1}{n_i} \sum_{k=1}^{n_i} f_j^i(Z_k^i) f_l^i(Z_k^i)(Y_k^i - g^i(X_k^i; \tilde{\theta}))^2. \tag{9}$$

Now considering the vector space $\mathcal{V}$ of real-valued functions $f_i$ of $Z$, $\psi_{n_i}(f^i; \theta) = \left(\psi_{n_i}(f_1^i; \theta), \psi_{n_i}(f_2^i; \theta), \ldots, \psi_{n_i}(f_m^i; \theta)\right)$ is a linear operator on $\mathcal{V}$ and

$$\mathcal{C}_{\tilde{\theta}}(f^i, h^i) = \frac{1}{n_i} \sum_{k=1}^{n_i} f^i(Z_k^i) h^i(Z_k^i)(Y_k^i - g^i(X_k^i; \tilde{\theta}))^2 \tag{10}$$

is a bilinear form. With that, for any subset $\mathcal{F}^i \subset \mathcal{V}$, we define a function

$$\Psi_{n_i}(\theta, \mathcal{F}^i, \tilde{\theta}) = \sup_{f^i \in \mathcal{F}^i} \psi_{n_i}(f^i; \theta) - \frac{1}{4} \mathcal{C}_{\tilde{\theta}}(f^i, f^i),$$

which leads to the following client-local optimization problem:

$$\theta^{\text{GMM}} \in \underset{\theta \in \Theta}{\arg\min} \, \Psi_{n_i}(\theta, \mathcal{F}^i, \tilde{\theta}), \tag{11}$$

where $\mathcal{F}^i = span(\{f_j^i\}_{j=1}^m)$, $\Psi_{n_i}(\theta, \mathcal{F}^i, \tilde{\theta}) = \|\psi_{n_i}(f_1^i; \theta), \psi_{n_i}(f_2^i; \theta), \ldots, \psi_{n_i}(f_m^i; \theta)\|_{\tilde{\theta}}^2$, and the the weighted norm $\|\|_{\tilde{\theta}}$ defined by equation (9).

**The zero-sum game for deep generalized method of moments.** As the data-dimension grows, the function class $\mathcal{F}^i$ is replaced with a class of neural networks of a certain architecture, i.e. $\mathcal{F}^i = \{f^i(z, \tau) : \tau \in \mathcal{T}\}$ with varying weights $\tau$. Similarly, let $\mathcal{G}^i = \{g^i(x, \theta) : \theta \in \Theta\}$ be another class of neural networks with varying weights $\theta$. With that, define

$$U_{\tilde{\theta}}^i(\theta, \tau) := \frac{1}{n_i} \sum_{k=1}^{n_i} f^i(Z_k^i, \tau)\left(Y_k^i - g^i(X_k^i; \theta)\right) - \frac{1}{4n_i} \sum_{k=1}^{n_i} \left(f^i(Z_k^i, \tau)\right)^2 \left(Y_k^i - g^i(X_k^i; \theta)\right)^2 \tag{12}$$

Then for a client $i$, (11) is reformulated as the following

$$\theta^{\text{DGMM}} \in \underset{\theta \in \Theta}{\arg\min} \sup_{\tau \in \mathcal{T}} U_{\tilde{\theta}}^i(\theta, \tau). \tag{13}$$

Equation (13) forms a zero-sum game, whose equilibrium solution is shown to be a true estimator to $\theta_0$ under a set of standard assumptions; see Theorem 2 in (Bennett et al., 2019).

## 3 FEDERATED DEEP GMM VIA FEDERATED EQUILIBRIUM SOLUTIONS

### 3.1 FEDERATED DEEP GENERALIZED METHOD MOMENT (FEDDEEPGMM)

We need to find the global moment estimators for the causal response function to fit data on each client. Thus, the federated counterpart of equation (5) is given by

$$\psi(f; \theta) := \mathbb{E}_i[\mathbb{E}[f^i(Z^i)(Y_k^i - g^i(X^i; \theta))]] = 0, \tag{14}$$

where the expectation $\mathbb{E}_i$ is over the clients. In this work, we consider *full client participation*. Thus, for the empirical federated moment estimation, we formulate:

$$\psi_n(f; \theta) := \frac{1}{N} \sum_{i=1}^N \psi_{n_i}(f^i; \theta) = \frac{1}{N} \sum_{i=1}^N \frac{1}{n_i} \sum_{k=1}^{n_i} f^i(Z_k^i)(Y_k^i - g^i(X_k^i; \theta)) \tag{15}$$

With that, the federated problem for GMM following (11) is formulated as:

$$\theta^{\text{FedDeepGMM}} \in \underset{\theta \in \Theta}{\arg\min} \|\psi_n(f; \theta)\|_{\tilde{\theta}}^2, \tag{16}$$

where $\|w\|_{\tilde{\theta}} = w^\top \mathcal{C}_{\tilde{\theta}}^{-1} x$ is the previously defined weighted-norm with inverse covariance as weights. We propose FEDDEEPGMM, a "deep" reformulation of the federated optimization problem based on the neural networks of a given architecture shared among clients and is shown to have the same solution as the federated GMM problem formulated earlier.

**Lemma 1.** *Let $\mathcal{F} = span\{f_j^i \mid i \in [N],\ j \in [m]\}$. An equivalent objective function for the federated moment estimation optimization problem (16) is given by:*

$$\|\psi_N(f;\theta)\|_{\tilde{\theta}}^2 = \sup_{\substack{f^i \in \mathcal{F} \\ \forall i \in [N]}} \frac{1}{N} \sum_{i=1}^{N} \left( \psi_{n_i}(f^i;\theta) - \frac{1}{4} \mathcal{C}_{\tilde{\theta}}(f^i;f^i) \right),\ where$$

$$\psi_{n_i}(f^i;\theta) := \frac{1}{n_i} \sum_{k=1}^{n_i} f^i(Z_k^i)(Y_k^i - g^i(X_k^i;\theta)),\ and\ \mathcal{C}_{\tilde{\theta}}(f^i,f^i) := \frac{1}{n_i} \sum_{k=1}^{n_i} (f^i(Z_k^i))^2 (Y_k^i - g^i(X_k^i;\tilde{\theta}))^2.$$

The proof of Lemma 1 is given in Appendix B.1. The federated zero-sum game is then defined by:

$$\hat{\theta}^{\text{FedDeepGMM}} \in \arg\min_{\theta \in \Theta} \sup_{\tau \in \mathcal{T}} U_{\tilde{\theta}}(\theta,\tau) := \frac{1}{N} \sum_{i=1}^{N} U_{\tilde{\theta}}^i(\theta,\tau), \tag{17}$$

where $U_{\tilde{\theta}}^i(\theta,\tau)$ is defined in equation (12). The federated DEEPGMM formulation as a zero-sum game defined by a federated minimax optimization problem (17) provides a framework to recover the global estimator as a federated equilibrium solution.

Referring to (Dayan et al., 2021), $\hat{\theta}^{\text{FedDeepGMM}}$ is the federated GMM estimator that consistently estimates the moment conditions of the clients under an approximation error as described later in Definition 3. These moment conditions are then employed on each client to analyse the impact of the instrumental variable $Z^i$.

## 3.2 Federated Sequential Games and Their Equilibrium Solutions

As minimax is not equal to maximin in general for a non-convex-non-concave problem, it is important to model the federated game as a sequential game (Jin et al., 2020) whose outcome would depend on what move – maximization or minimization – is taken first. We start with the following assumptions:

**Assumption 1.** Client-local objective $U_{\tilde{\theta}}^i(\theta,\tau)\ \forall i \in [N]$ is twice continuously differentiable for both $\theta$ and $\tau$. Thus, the global objective $U_{\tilde{\theta}}(\theta,\tau)$ is also a twice continuously differentiable function.

**Assumption 2** (**Smoothness**). The gradient of each client's local objective, $\nabla U_{\tilde{\theta}}^i(\theta,\tau)$, is Lipschitz continuous with respect to both $\theta$ and $\tau$. For all $i \in [N]$, there exist constants $L > 0$ such that:

$$\|\nabla_\theta U_{\tilde{\theta}}^i(\theta_1,\tau_1) - \nabla_\theta U_{\tilde{\theta}}^i(\theta_2,\tau_2)\| \leq L\|(\theta_1,\tau_1) - (\theta_2,\tau_2)\|,\ and$$

$$\|\nabla_\tau U_{\tilde{\theta}}^i(\theta_1,\tau_1) - \nabla_\tau U_{\tilde{\theta}}^i(\theta_2,\tau_2)\| \leq L\|(\theta_1,\tau_1) - (\theta_2,\tau_2)\|,$$

$\forall (\theta_1,\tau_1), (\theta_2,\tau_2)$. Thus, $U_{\tilde{\theta}}(\theta,\tau)$ is $L$-Lipschitz smooth.

**Assumption 3** ( **Bounded Gradient Dissimilarity**). The heterogeneity of the local gradients with respect to (w.r.t.) $\theta$ and $\tau$ is bounded as follows:

$$\|\nabla_\theta U_{\tilde{\theta}}^i(\theta,\tau) - \nabla_\theta U_{\tilde{\theta}}(\theta,\tau)\| \leq \zeta_\theta^i \qquad \|\nabla_\tau U_{\tilde{\theta}}^i(\theta,\tau) - \nabla_\tau U_{\tilde{\theta}}(\theta,\tau)\| \leq \zeta_\tau^i,$$

where $\zeta_\theta^i,\ \zeta_\tau^i \geq 0$ are the bounds that quantify the degree of gradient dissimilarity at client $i \in [N]$.

**Assumption 4** (**Bounded Hessian Dissimilarity**). The heterogeneity in terms of hessian w.r.t. $\theta$ and $\tau$ is bounded as follows:

$$\|\nabla_{\theta\theta}^2 U_{\tilde{\theta}}^i(\theta,\tau) - \nabla_{\theta\theta}^2 U_{\tilde{\theta}}(\theta,\tau)\|_\sigma \leq \rho_\theta^i, \qquad \|\nabla_{\tau\tau}^2 U_{\tilde{\theta}}^i(\theta,\tau) - \nabla_{\tau\tau}^2 U_{\tilde{\theta}}(\theta,\tau)\|_\sigma \leq \rho_\tau^i,$$

$$\|\nabla_{\theta\tau}^2 U_{\tilde{\theta}}^i(\theta,\tau) - \nabla_{\theta\tau}^2 U_{\tilde{\theta}}(\theta,\tau)\|_\sigma \leq \rho_{\theta\tau}^i, \qquad \|\nabla_{\tau\theta}^2 U_{\tilde{\theta}}^i(\theta,\tau) - \nabla_{\tau\theta}^2 U_{\tilde{\theta}}(\theta,\tau)\|_\sigma \leq \rho_{\tau\theta}^i,$$

where $\rho_\theta^i,\ \rho_\tau^i,\ \rho_{\theta\tau}^i$, and $\rho_{\tau\theta}^i \geq 0$ quantify the degree of hessian dissimilarity at client $i \in [N]$ by spectral norm $\|.\|_\sigma$.

Assumptions 3 and 4 provide a measure of data heterogeneity across clients in a federated setting. In the special case, when $\zeta$ and $\rho$'s are all 0, then the data is homogeneous across clients.

We adopt the Stackelberg equilibrium for pure strategies (Jin et al., 2020) to characterize the solution of the minimax federated optimization problem for a non-convex non-concave function $U_{\tilde{\theta}}(\theta,\tau)$ for the sequential game where min-player goes first and the max-player goes second. *To avoid ambiguity between the adjectives of the terms global/local objective functions in federated learning and the global/local nature of minimax points in optimization, we refer to a global objective as the federated objective and a local objective as the client's objective.*

**Definition 1** (**Local minimax point**). *[Definition 14 of (Jin et al., 2020)] Let $U(\theta, \tau)$ be a function defined over $\Theta \times \mathcal{T}$ and let $h$ be a function satisfying $h(\delta) \to 0$ as $\delta \to 0$. There exists a $\delta_0$, such that for any $\delta \in (0, \delta_0]$, and any $(\theta, \tau)$ such that $\|\theta - \hat{\theta}\| \leq \delta$ and $\|\tau - \hat{\tau}\| \leq \delta$, then a point $(\hat{\theta}, \hat{\tau})$ is a local minimax point of $U$, if $\forall (\theta, \tau) \in \Theta \times \mathcal{T}$, it satisfies:*

$$U_{\tilde{\theta}}(\hat{\theta}, \tau) \leq U_{\tilde{\theta}}(\hat{\theta}, \hat{\tau}) \leq \max_{\tau\prime: \|\tau\prime - \hat{\tau}\| \leq h(\delta)} U_{\tilde{\theta}}(\theta, \tau\prime). \tag{18}$$

With that, the first-order & second-order necessary conditions for local minimax points are as below.

**Lemma 2** (Propositions 18, 19, 20 of (Jin et al., 2020)). *Under assumption 1, any local minimax point satisfies the following conditions:*

- ***First-order Necessary Condition:** A local minimax point $(\theta, \tau)$ satisfies: $\nabla_{\theta} U_{\tilde{\theta}}(\theta, \tau) = 0$ and $\nabla_{\tau} U_{\tilde{\theta}}(\theta, \tau) = 0$.*

- ***Second-order Necessary Condition:** A local minimax point $(\theta, \tau)$ satisfies: $\nabla^2_{\tau\tau} U_{\tilde{\theta}}(\theta, \tau) \preceq \mathbf{0}$. Moreover, if $\nabla^2_{\tau\tau} U_{\tilde{\theta}}(\theta, \tau) \prec 0$, then $\left[ \nabla^2_{\theta\theta} U_{\tilde{\theta}} - \nabla^2_{\theta\tau} U_{\tilde{\theta}} \left( \nabla^2_{\tau\tau} U_{\tilde{\theta}} \right)^{-1} \nabla^2_{\tau\theta} U_{\tilde{\theta}} \right] (\theta, \tau) \succeq 0$.*

- ***Second-order Sufficient Condition:** A stationary point $(\theta, \tau)$ that satisfies $\nabla^2_{\tau\tau} U_{\tilde{\theta}}(\theta, \tau) \prec \mathbf{0}$, and $\left[ \nabla^2_{\theta\theta} U_{\tilde{\theta}} - \nabla^2_{\theta\tau} U_{\tilde{\theta}} \left( \nabla^2_{\tau\tau} U_{\tilde{\theta}} \right)^{-1} \nabla^2_{\tau\theta} U_{\tilde{\theta}} \right] (\theta, \tau) \succ 0$ guarantees that $(\theta, \tau)$ is a strict local minimax.*

Now, in order to define the federated approximate equilibrium solutions, we first define an approximate local minimax point.

**Definition 2** (**Approximate Local minimax point**). *[An adaptation of definition 34 of (Jin et al., 2020)] Let $U(\theta, \tau)$ be a function defined over $\Theta \times \mathcal{T}$ and let $h$ be a function satisfying $h(\delta) \to 0$ as $\delta \to 0$. There exists a $\delta_0$, such that for any $\delta \in (0, \delta_0]$, and any $(\theta, \tau)$ such that $\|\theta - \hat{\theta}\| \leq \delta$ and $\|\tau - \hat{\tau}\| \leq \delta$, then a point $(\hat{\theta}, \hat{\tau})$ is an $\varepsilon$-approximate local minimax point of $U$, if it satisfies:*

$$U_{\tilde{\theta}}(\hat{\theta}, \tau) - \varepsilon \leq U_{\tilde{\theta}}(\hat{\theta}, \hat{\tau}) \leq \max_{\tau\prime: \|\tau\prime - \hat{\tau}\| \leq h(\delta)} U_{\tilde{\theta}}(\theta, \tau\prime) + \varepsilon, \tag{19}$$

We aim to achieve approximate local minimax points for every client as a solution of the federated minimax optimization. With that, we characterize the federated solution as the following.

**Definition 3** (**$\mathcal{E}$-Approximate Federated Equilibrium Solutions**). *Let $\mathcal{E} = \{\varepsilon^i\}_{i=1}^{N}$ be the approximation error vector for clients $i \in [N]$. Let $U_{\tilde{\theta}}^i(\theta, \tau)$ be a function defined over $\Theta \times \mathcal{T}$ for a client $i \in [N]$ and $U_{\tilde{\theta}}(\theta, \tau) := \frac{1}{N} \sum_{i=1}^{N} U_{\tilde{\theta}}^i(\theta, \tau)$. An $\mathcal{E}$-approximate federated equilibrium point $(\hat{\theta}, \hat{\tau})$ (that is an $\varepsilon^i$-approximate local minimax point for each client's objective $U_{\tilde{\theta}}^i$), must follow the conditions below:*

1. *$\varepsilon^i$- **First-order Necessary Condition:** The point $(\hat{\theta}, \hat{\tau})$ must be an $\varepsilon^i$ stationary point for every client $i \in [N]$, i.e., $\|\nabla_{\theta} U_{\tilde{\theta}}^i(\hat{\theta}, \hat{\tau})\| \leq \varepsilon^i, \quad \text{and} \quad \|\nabla_{\tau} U_{\tilde{\theta}}^i(\hat{\theta}, \hat{\tau})\| \leq \varepsilon^i$.*

2. ***Second-Order $\varepsilon^i$ Necessary Condition:** The point $(\hat{\theta}, \hat{\tau})$ must satisfy the second-order conditions: $\nabla^2_{\tau\tau} U_{\tilde{\theta}}^i(\hat{\theta}, \hat{\tau}) \preceq -\varepsilon^i I, \quad \text{and} \quad \left[ \nabla^2_{\theta\theta} U_{\tilde{\theta}}^i - \nabla^2_{\theta\tau} U_{\tilde{\theta}}^i \left( \nabla^2_{\tau\tau} U_{\tilde{\theta}} \right)^{-1} \nabla^2_{\tau\theta} U_{\tilde{\theta}}^i \right] (\hat{\theta}, \hat{\tau}) \succeq \varepsilon^i I$.*

3. ***Second-Order $\varepsilon^i$ Sufficient Condition:** An $\varepsilon^i$ stationary point $(\theta, \tau)$ that satisfies $\nabla^2_{\tau\tau} U_{\tilde{\theta}}^i(\hat{\theta}, \hat{\tau}) \prec -\varepsilon^i I$, and $\left[ \nabla^2_{\theta\theta} U_{\tilde{\theta}}^i - \nabla^2_{\theta\tau} U_{\tilde{\theta}}^i \left( \nabla^2_{\tau\tau} U_{\tilde{\theta}}^i \right)^{-1} \nabla^2_{\tau\theta} U_{\tilde{\theta}}^i \right] (\hat{\theta}, \hat{\tau}) \succ \varepsilon^i I$ guarantees that $(\hat{\theta}, \hat{\tau})$ is a strict local minimax point $\forall i \in [N]$ that satisfies $\varepsilon^i$ approximate equilibrium as in definition 2.*

We now state the main theoretical result of our work in this theorem.

**Theorem 1.** *Under assumptions 1, 2, 3 and 4, a minimax solution $(\hat{\theta}, \hat{\tau})$ of federated optimization problem (17) that satisfies the equilibrium condition as in definition 1: $U_{\tilde{\theta}}(\hat{\theta}, \tau) \leq U_{\tilde{\theta}}(\hat{\theta}, \hat{\tau}) \leq \max_{\tau\prime: \|\tau\prime - \hat{\tau}\| \leq h(\delta)} U_{\tilde{\theta}}(\theta, \tau\prime)$, is an $\mathcal{E}$-approximate federated equilibrium solution as defined in 3, where the approximation error $\varepsilon^i$ for each client $i \in [N]$ lies in: $\max\{\zeta_{\theta}^i, \zeta_{\tau}^i\} \leq \varepsilon^i \leq \min\{\alpha - \rho_{\tau}^i, \beta - B^i\}$ for $\rho_{\tau}^i < \alpha$ and $B^i > \beta$, such that $\alpha := \left| \lambda_{\max} \left( \nabla^2_{\tau\tau} U_{\tilde{\theta}}(\hat{\theta}, \hat{\tau}) \right) \right|$, $\beta := \lambda_{\min} \left( \left[ \nabla^2_{\theta\theta} U_{\tilde{\theta}} - \nabla^2_{\theta\tau} U_{\tilde{\theta}} \left( \nabla^2_{\tau\tau} U_{\tilde{\theta}} \right)^{-1} \nabla^2_{\tau\theta} U_{\tilde{\theta}} \right] (\hat{\theta}, \hat{\tau}) \right)$ and $B^i := \rho_{\theta}^i + L\rho_{\theta\tau}^i \frac{1}{|\lambda_{\max}(\nabla^2_{\tau\tau} U_{\tilde{\theta}}^i)|} + L\rho_{\tau\theta}^i \frac{1}{|\lambda_{\max}(\nabla^2_{\tau\tau} U_{\tilde{\theta}}^i)|} + L^2 \rho_{\tau}^i \frac{1}{|\lambda_{\max}(\nabla^2_{\tau\tau} U_{\tilde{\theta}}^i) \cdot \lambda_{\max}(\nabla^2_{\tau\tau} U_{\tilde{\theta}})|}$.*

The proof of Theorem 1 is given in Appendix B.2. Note that when data is homogeneous (i.e., for each client $i$, $\zeta_\theta^i, \zeta_\tau^i, \rho_\tau^i$ and $B^i$ are all zeroes), each client satisfies an exact local minimax equilibrium.

**Remark 1.** In Theorem 1, note that if the interval $[\max\{\zeta_\theta^i, \zeta_\tau^i\}, \min\{\alpha - \rho_\tau^i, \beta - B^i\}]$ is empty, i.e. $\max\{\zeta_\theta^i, \zeta_\tau^i\} > \min\{\alpha - \rho_\tau^i, \beta - B^i\}$, then no such $\varepsilon^i$ exists and $(\hat\theta, \hat\tau)$ fails to be a local $\varepsilon^i$ approximate equilibrium point for that clients. It may happen in two cases:

1. The gradient dissimilarity $\zeta_\theta^i, \zeta_\tau^i$ is too large, indicating high heterogeneity, then $(\hat\theta, \hat\tau)$- the solution to the federated objective would fail to become an approximate equilibrium point for the clients. It is a practical consideration for a federated convergence facing difficulty against high heterogeneity.
2. If $\alpha \approx \rho_\tau^i$ or $\beta \approx B^i$, this indicates that the client's local curvature structure significantly differs from the global curvature. In this case, the client's objective may be flatter or even oppositely curved compared to the global model, reflecting high heterogeneity.

Now we state the result on the per-client consistency of the FEDGMM estimator.

**Theorem 2 (Consistency).** *[Adaptation of Theorem 2 of (Bennett et al., 2019)] Let $\tilde\theta_n$ be a data-dependent choice for the federated objective that has a limit in probability. Let $h$ be a function satisfying $h(\delta) \to 0$ as $\delta \to 0$. For each client $i \in [N]$, define $m^i(\theta, \tau, \tilde\theta) := f^i(Z^i; \tau)(Y^i - g(X^i; \theta)) - \frac{1}{4}f^i(Z^i; \tau)^2(Y^i - g(X^i; \tilde\theta))^2$, $M^i(\theta) = \sup_{\tau \in \mathcal{T}} \mathbb{E}[m^i(\theta, \tau, \tilde\theta)]$ and $\eta^i(\epsilon) := \inf_{d(\theta, \theta_0) \geq \epsilon} M^i(\theta) - M^i(\theta_0)$ for every $\epsilon > 0$. Fix some $\delta_0$, for any $\delta \in (0, \delta_0]$ and any $(\theta, \tau)$ such that $\|\theta - \hat\theta\| \leq \delta$ and $\|\tau - \hat\tau\| \leq \delta$, let $(\hat\theta_n, \hat\tau_n)$ be a solution that satisfies the approximate equilibrium for each of the client $i \in [N]$ as*

$$\sup_{\tau \in \mathcal{T}} U_{\hat\theta}^i(\hat\theta_n, \tau) - \varepsilon^i - o_p(1) \leq U_{\hat\theta}^i(\hat\theta_n, \hat\tau_n) \leq \inf_{\theta \in \Theta} \max_{\tau\prime:\|\tau\prime - \hat\tau_n\| \leq h(\delta)} U_{\hat\theta}^i(\theta, \tau\prime) + \varepsilon^i + o_p(1).$$

*Then, under similar assumptions as in Assumptions 1 to 5 of (Bennett et al., 2019), the global solution $\hat\theta_n$ is a consistent estimator to the true parameter $\theta_0$, i.e. $\hat\theta_n \xrightarrow{p} \theta_0$ when the approximate error $\varepsilon^i < \frac{\eta^i(\epsilon)}{2}$ for every $\epsilon > 0$ for each client $i \in [N]$.*

The assumptions and the proof of Theorem 2 are included in Appendix B.3.

**Remark 2.** Theorem 2 formalizes a tradeoff between data heterogeneity and the consistency of the global estimator in federated learning for each client. If the approximation error $\varepsilon^i$ is large for a client $i \in [N]$, then the solution $\hat\theta_n$ may fail to consistently estimate the true parameter of client $i$. In contrast, when data across clients have similar distribution (i.e., case for low heterogeneity), the federated optimal model $\hat\theta_n$ is consistent across clients.

### 3.3 FEDERATED GRADIENT DESCENT ASCENT ALGORITHM AND IT'S LIMIT POINTS

Bennett et al. (2019) used Optimistic Adam (OADAM), a variant of Adam (Kingma, 2015) based stochastic gradient descent ascent (SGDA) algorithm (Daskalakis et al., 2018). However, it is known that a well-tuned SGD outperforms Adam in overparametrized settings (Wilson et al., 2017). As our experiments show in Section (4), that gradient descent ascent updates are competitive to OADAM for minimax optimization in centralized setting. Considering this, we employ an adaptation of the standard gradient descent ascent algorithm to federated (FEDGDA) setting.

FEDGDA is well-explored in the literature: (Deng & Mahdavi, 2021; Sharma et al., 2022; Shen et al., 2024; Wu et al., 2024). The clients run the gradient descent ascent algorithm for several local updates and then the orchestrating server synchronizes them by collecting the model states, averaging them, and broadcasting it to the clients. A detailed description is included as a pseudocode in Appendix A.

Similar to (Bennett et al., 2019), we note that the federated minimax optimization problem (17) is not convex-concave on $(\theta, \tau)$. The convergence results of variants of FEDGDA (Sharma et al., 2022; Shen et al., 2024; Wu et al., 2024) assume that $U_{\tilde\theta}(\theta, \tau)$ is non-convex on $\theta$ and satisfies a $\mu-$Polyak Łojasiewicz (PL) inequality on $\tau$, see assumption 4 in (Sharma et al., 2022). PL condition is known to be satisfied by over-parametrized neural networks (Charles & Papailiopoulos, 2018; Liu et al., 2022). The convergence results of FEDGDA will follow (Sharma et al., 2022). We include a formal statement in Appendix A. However, beyond convergence, we primarily aim to show that an optimal solution will consistently estimate the moment conditions of the clients, which we do next.

For Algorithm 1 in Appendix A, let $\alpha_1 = \frac{\eta}{\gamma}, \alpha_2 = \eta$ be the learning rates for gradient updates to $\theta$ and $\tau$, respectively. Without loss of generality the FEDGDA updates are:

$$\theta_{t+1} = \theta_t - \eta \frac{1}{\gamma} \frac{1}{N} \sum_{i \in [N]} \sum_{r=1}^{R} \nabla_\theta U_{\tilde{\theta}}^i(\theta_{t,r}^i, \tau_{t,r}^i) \text{ and } \tau_{t+1} = \tau_t + \eta \frac{1}{N} \sum_{i \in [N]} \sum_{r=1}^{R} \nabla_\tau U_{\tilde{\theta}}^i(\theta_{t,r}^i, \tau_{t,r}^i)$$

We call it $\gamma$-FEDGDA, where $\gamma$ is the ratio of $\alpha_1$ to $\alpha_2$. As $\eta \to 0$ corresponds to FEDGDA-flow, under the smoothness of $U_{\tilde{\theta}}^i$, bounded gradient heterogeneity (assumption 3) and for fixed local rounds $R$, FEDGDA-flow becomes:

$$\frac{d\theta}{dt} = -\frac{1}{\gamma} R \nabla_\theta U_{\tilde{\theta}}(\theta, \tau) + \mathcal{O}\left(\frac{R}{\gamma}\zeta_\theta\right), \text{ and } \frac{d\tau}{dt} = R \nabla_\tau U_{\tilde{\theta}}(\theta, \tau) + \mathcal{O}(R\zeta_\tau).$$

We further elaborate on FEDGDA-flow in Appendix C.1. We aim to find out the relationship between stable equilibrium and local minimax points of the federated optimization problem. For that, we now define a strictly linearly stable equilibrium of the $\gamma$−FEDGDA flow.

**Proposition 1.** *Given the Jacobian matrix for $\gamma$−FEDGDA flow as $\mathbf{J} = \begin{pmatrix} -\frac{1}{\gamma} R \nabla_{\theta\theta}^2 U_{\tilde{\theta}}(\theta, \tau) & -\frac{1}{\gamma} R \nabla_{\theta\tau}^2 U_{\tilde{\theta}}(\theta, \tau) \\ R \nabla_{\tau\theta}^2 U_{\tilde{\theta}}(\theta, \tau) & R \nabla_{\tau\tau}^2 U_{\tilde{\theta}}(\theta, \tau) \end{pmatrix}$, a point $(\theta, \tau)$ is a strictly linearly stable equilibrium of the $\gamma$−FEDGDA flow if and only if the real parts of all eigenvalues of $\mathbf{J}$ are negative, i.e., $\mathrm{Re}(\Lambda_j) < 0$ for all $j$.*

The proof follows a strategy similar to (Jin et al., 2020).

Let $\gamma$-$\mathcal{FGDA}$ be the set of strictly linearly stable points of the $\gamma$-FEDGDA flow, and $\mathcal{LocMinimax}$ be the set of local minimax points of the federated zero-sum game. Define

$$\overline{\infty - \mathcal{FGDA}} := \limsup_{\gamma \to \infty} \gamma - \mathcal{FGDA} := \cap_{\gamma_0 > 0} \cup_{\gamma > \gamma_0} \gamma - \mathcal{FGDA}, \text{ and}$$

$$\underline{\infty - \mathcal{FGDA}} := \liminf_{\gamma \to \infty} \gamma - \mathcal{FGDA} := \cup_{\gamma_0 > 0} \cap_{\gamma > \gamma_0} \gamma - \mathcal{FGDA}.$$

We now state the theorem that establishes that the stable limit points of $\infty$-$\mathcal{FGDA}$ are the local minimax points, up to some degenerate cases.

**Theorem 3.** *Under Assumption 1, $\mathcal{LocMinimax} \subset \underline{\infty - \mathcal{FGDA}} \subset \overline{\infty - \mathcal{FGDA}} \subset \mathcal{LocMinimax} \cup \mathcal{A}$, where $\mathcal{A} := \{(\theta, \tau) | (\theta, \tau) \text{ is stationary and } \nabla_{\tau\tau}^2 U_{\tilde{\theta}}(\theta, \tau) \text{ is degenerate}\}$. Moreover, if the hessian $\nabla_{\tau\tau}^2 U_{\tilde{\theta}}(\theta, \tau)$ is smooth, then $\mathcal{A}$ has measure zero in $\Theta \times \mathcal{T} \subset \mathbb{R}^d \times \mathbb{R}^k$.*

Essentially, Theorem 3 states that the limit points of FEDGDA are the local minimax solutions, and thereby the equilibrium solution of the federated zero-sum game, up to some degenerate case. The proof of Theorem 3 is included in Appendix C.2. Theorems 1, 2, and 3 together complete the theoretical foundation of the pipeline in our work.

# 4 EXPERIMENTS

We extend the experimental evaluations of DEEPGMM (Bennett et al., 2019) to a federated setting. We further discuss this benchmark structure in Appendix E. More specifically, we evaluate the ability of FEDDEEPGMM to fit low- and high- dimensional data to demonstrate its convergence. Similar to DEEPGMM, we assess two scenarios in regards to $((X, Y), Z)$:

(a) **The instrumental and treatment variables $Z$ and $X$ are both low-dimensional.** In this case, we use 1-dimensional synthetic datasets corresponding to the following functions: (a) **Absolute**: $g_0(x) = |x|$, (b) **Step**: $g_0(x) = 1_{\{x \geq 0\}}$, (c) **Linear**: $g_0(x) = x$.
To generate the synthetic data, similar to (Bennett et al., 2019; Lewis & Syrgkanis, 2018) we apply the following generation process:

$$Y = g_0(X) + e + \delta \qquad \text{and } X = Z^{(1)} + Z^{(2)} + e + \gamma \qquad (20)$$

$$(Z^{(1)}, Z^{(2)}) \sim \text{Uniform}([-3, 3]^2) \qquad \text{and } e \sim \mathcal{N}(0, 1), \quad \gamma, \delta \sim \mathcal{N}(0, 0.1) \qquad (21)$$

(b) $Z$ **and** $X$ **are low-dimensional or high-dimensional or both.** First, $Z$ and $X$ are generated as in (20,21). Then for high-dimensional data, we map $Z$ and $X$ to an image using the mapping:

$$\text{Image}(x) = \text{Dataset}\left(\text{round}\left(\min\left(\max(1.5x + 5, 0), 9\right)\right)\right),$$

where $(\text{round}(\min(\max(1.5x+5,0),9)))$ returns an integer between 0 and 9. Essentially, the function Dataset $(.)$ randomly selects an image following its index. We use datasets FEMNIST (Federated Extended MNIST) and CIFAR10 (Caldas et al., 2018) for images of size $28 \times 28$ and $3 \times 32 \times 32$, respectively. Thus, we have the following cases: (a) **Dataset$_\mathbf{z}$**: $X = X^{\text{low}}, Z = \text{Image}(Z^{\text{low}})$, (b) **Dataset$_\mathbf{x}$**: $Z = Z^{\text{low}}, X = \text{Image}(X^{\text{low}})$, and (c) **Dataset$_\mathbf{x,z}$**: $Z = \text{Image}(Z^{\text{low}})$, $X = \text{Image}(X^{\text{low}})$, where **Dataset** takes values **FEMNIST** and **CIFAR10**.

We implemented and benchmarked FEDGDA and FEDSGDA to solve the FEDDEEPGMM problem. For reference, we implemented OADAM, GDA, and SGDA to solve the DEEPGMM in centralized setting. For high-dimensional scenarios, we implement a CNN architecture to process images, while for low-dimensional scenarios, we use a multilayer perceptron (MLP). Code is available at `https://anonymous.4open.science/r/FederatedDeepGMM-417C`.

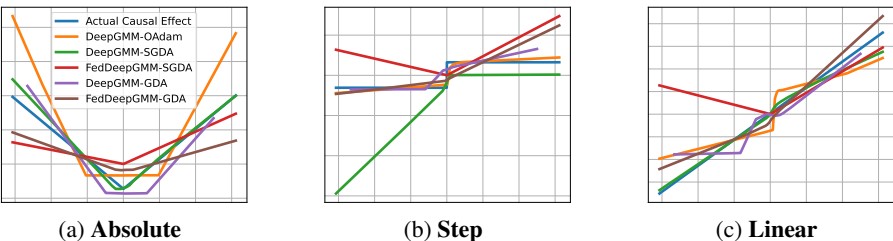

(a) **Absolute**    (b) **Step**    (c) **Linear**

Figure 1: Estimated $\hat{g}$ compared to true $g$ in low-dimensional scenarios

| Estimations | DEEPGMM-OAdam | DEEPGMM-GDA | FDEEPGMM-GDA | DEEPGMM-SGDA | FDEEPGMM-SGDA |
|---|---|---|---|---|---|
| **Absolute** | $0.03 \pm 0.01$ | $0.013 \pm .01$ | $0.4 \pm 0.01$ | $0.009 \pm 0.01$ | $0.2 \pm 0.00$ |
| **Step** | $0.3 \pm 0.00$ | $0.03 \pm 0.00$ | $0.04 \pm 0.01$ | $0.112 \pm 0.00$ | $0.23 \pm 0.01$ |
| **Linear** | $0.01 \pm 0.00$ | $0.02 \pm 0.00$ | $0.01 \pm 0.00$ | $0.03 \pm 0.00$ | $0.04 \pm 0.00$ |
| **FEMNIST$_\mathbf{x}$** | $0.50 \pm 0.00$ | $1.11 \pm 0.01$ | $0.21 \pm 0.02$ | $0.40 \pm 0.01$ | $0.19 \pm 0.01$ |
| **FEMNIST$_\mathbf{x,z}$** | $0.24 \pm 0.00$ | $0.46 \pm 0.09$ | $0.19 \pm 0.03$ | $0.14 \pm 0.02$ | $0.20 \pm 0.00$ |
| **FEMNIST$_\mathbf{z}$** | $0.10 \pm 0.00$ | $0.42 \pm 0.01$ | $0.24 \pm 0.01$ | $0.11 \pm 0.02$ | $0.23 \pm 0.01$ |
| **CIFAR10$_\mathbf{x}$** | $0.55 \pm 0.30$ | $0.19 \pm 0.01$ | $0.25 \pm 0.03$ | $0.20 \pm 0.08$ | $0.22 \pm 0.08$ |
| **CIFAR10$_\mathbf{x,z}$** | $0.40 \pm 0.11$ | $0.24 \pm 0.00$ | $0.24 \pm 0.03$ | $0.19 \pm 0.03$ | $0.22 \pm 0.02$ |
| **CIFAR10$_\mathbf{z}$** | $0.13 \pm 0.03$ | $0.13 \pm 0.01$ | $1.70 \pm 2.60$ | $0.24 \pm 0.01$ | $0.52 \pm 0.60$ |

Table 1: The averaged Test MSE with standard deviation on the low- and high-dimensional scenarios.

**Non-i.i.d. data.** To set up a non-i.i.d. distribution of data between clients, samples were divided amongst the clients using a Dirichlet distribution $Dir_S(\alpha)$ (Hsu et al., 2019), where $\alpha$ determines the degree of heterogeneity across $S$ clients. We used $Dir_S(\alpha) = 0.3$ for each train, test, and validation samples. Given the non-i.i.d. data, for the low-dimensional scenario, we sample $n = 20000$ points for each train, validation, and test set, while, for the high-dimensional scenario, we have $n = 20000$ for the train set and $n = 10000$ for the validation and test set.

**Hyperparameters.** We perform extensive grid-search to tune the learning rate. For FEDSGDA, we use a minibatch-size of 256. To avoid numerical instability, we standardize the observed $Y$ values by removing the mean and scaling to unit variance. We perform five runs of each experiment and present the mean and standard deviation of the results.

**Observations and Discussion.** In figure (1), we first observe that SGDA and GDA algorithms perform at par with OADAM to fit the DEEPGMM estimator. It establishes that hyperparameter tuning is effective. With that, we further observe that the federated algorithms efficiently fit the estimated function to the true data-generating process even though the data is decentralized and non-i.i.d. Thus, it shows that the federated algorithm converges effectively. In Table 1 we present the test mean squared error (MSE) values. In many cases, the federated MSE values are close or better than the centralized results, which sufficiently demonstrate that our federated implementation achieves a convergent dynamics. We include additional experimental results in Appendix E that investigate the effects of heterogeneity. These experiments establish the efficacy of our method.

## Existence of Federated Mixed-strategy Equilibrium and its Implications

In this work, we presented the equilibrium solutions of federated zero-sum games through federated local minimax solutions for non-convex non-concave minimax optimization problems. The translation of the federated equilibrium as an approximately consistent GMM estimator for the clients was obtained through the gradient and Hessian dissimilarities across the clients, see Theorem 1, Theorem 2, and Definition 3. We note that our minimax optimization solution provides a federated pure strategy equilibrium. However, a pure strategy equilibrium can correspond to only full gradients and a full client participation setting. To elaborate,

- Firstly, with stochastic gradients on the clients, there will be no guarantee of descent (correspondingly, ascent) at an optimization step, which is available only in expectation in this case. However, in a pure strategy zero-sum game, the minimizing player (correspondingly, the maximizing player) takes a step to minimize (correspondingly, maximize) the game objective at each step.

- Secondly, the path to the saddle-point of a player in a pure strategy game should be re-traceable/deterministic, which can not be possible with minimax optimization with stochastic gradients and/or partial client participation, considering the true random sampling.

Allowing for stochasticity, whether arising from stochastic gradients or client sampling for each communication round, would necessitate accommodating a distribution over multiple actions. Whereby the game ceases to be a pure strategy game, as the actions become non-deterministic, essentially, resulting in a mixed-strategy zero-sum game. It is well understood that, regardless of the analytical assumptions regarding the objective, mixed strategy solutions for zero-sum games exist (Jin et al., 2020).

However, for federated mixed strategy solutions, recovering a GMM estimator for a client is not immediate. To elaborate, there are no analogous necessary and sufficient conditions – the first-order and second-order necessary and sufficient conditions that we have for federated pure strategy solutions in Lemma 2 – for the mixed strategy solutions, which would correspond to a distribution over a set of the global model states synchronized across clients. Therefore, we can not directly apply Theorem 1. Still, we note here that a federated mixed strategy equilibrium will provide a robust federated GMM estimator compared to pure-strategy solutions, as it will output a probability distribution over a set of model states that accounts for the uncertainty across clients.

We leave the algorithm and characterization of a federated mixed strategy equilibrium solution for a robust federated GMM estimator as an open problem.

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

# APPENDIX

# A   FEDERATED GRADIENT DESCENT ASCENT ALGORITHM DESCRIPTION

---

**Algorithm 1** FEDGDA running on a federated learning server to solve the minimax problem (17)

---

**Server Input**: initial global estimate $\theta_1, \tau_1$; constant local learning rate $\alpha_1, \alpha_2$; total $N$ clients
**Output**: global model states $\theta_{T+1}, \tau_{T+1}$

 1: **for** synchronization round $t = 1, \ldots, T$ **do**
 2:     server sends $\theta_t, \tau_t$ to all clients
 3:     **for** each $i \in [N]$ in parallel **do**
 4:         $\theta_{t,1}^i \leftarrow \theta_t, \tau_{t,1}^i \leftarrow \tau_t$
 5:         **for** $r = 1, 2, \ldots, R$ **do**
 6:             $\theta_{t,r+1}^i = \theta_{t,r}^i - \alpha_1 \nabla_\theta U_{\tilde{\theta}}^i(\theta_{t,r}^i, \tau_{t,r}^i)$
 7:             $\tau_{t,r+1}^i = \tau_{t,r}^i + \alpha_2 \nabla_\tau U_{\tilde{\theta}}^i(\theta_{t,r}^i, \tau_{t,r}^i)$
 8:         **end for**
 9:         $(\Delta\theta_t^i, \Delta\tau_t) \leftarrow (\theta_{t,R+1}^i - \theta_t, \tau_{t,R+1}^i - \tau_t)$
10:     **end for**
11:     $(\Delta\theta_t, \Delta\tau_t) \leftarrow \frac{1}{N} \sum_{i \in [N]} (\Delta\theta_t^i, \Delta\tau_t^i)$
12:     $\theta_{t+1} \leftarrow (\theta_t + \Delta\theta_t), \tau_{t+1} \leftarrow (\tau_t + \Delta\tau_t)$
13: **end for**
14: **return** $\theta_{T+1}; \tau_{T+1}$

---

## A.1   CONVERGENCE OF FEDGDA

We adapt the proof of Theorem 1 in (Sharma et al., 2022) for the SGDA algorithm proposed in (Deng & Mahdavi, 2021) for the FEDGDA algorithm 1 for smooth non-convex- PL problems.

**Assumption 5** (Polyak Łojaisiewicz (PL) condition in $\tau$)**.** The function $U_{\tilde{\theta}}$ satisfyies $\mu - PL$ condition in $\tau$, $\mu > 0$, if for any fixed $\theta$, $\arg\max_{\tau'} U_{\tilde{\theta}}(\theta, \tau') \neq \phi$ and $\|\nabla_\tau U_{\tilde{\theta}}(\theta, \tau)\|^2 \geq 2\mu \left( \max_{\tau'} U_{\tilde{\theta}}(\theta, \tau') - U_{\tilde{\theta}}(\theta, \tau) \right)$.

We use the following result about the smoothness of $\Phi(\cdot)$.

**Lemma 3.** *(Nouiehed et al., 2019) If the function $U_{\tilde{\theta}}(\theta, \cdot)$ satisfies Assumptions 2, 5 (L-smoothness and $\mu$-PL condition in $\tau$), then $\Phi(\theta)$ is $L_\Phi$-smooth with $L_\Phi = \kappa L/2 + L$, where $\kappa = L/\mu$ is the condition number.*

**Lemma 4** (One-Step Envelope Descent)**.** *(Deng & Mahdavi, 2021) Suppose the local client loss functions $\{U_{\tilde{\theta}}^i(\theta, \tau)\}$ satisfy Assumptions 2, 5. Then the iterates generated by FEDGDA satisfy:*

$$\Phi(\theta_{t+1}) \leq \Phi(\theta_t) - \frac{\alpha_1}{2} \|\nabla\Phi(\theta_t)\|^2 - \frac{\alpha_1}{2} (1 - L_\Phi \alpha_1) \left\| \frac{1}{N} \sum_{i=1}^N \sum_{r=1}^R \nabla_\theta U_{\tilde{\theta}}^i(\theta_{t,r}^i, \tau_{t,r}^i) \right\|^2$$

$$+ \frac{2\alpha_1 L^2}{\mu} \left( \Phi(\theta_t) - U_{\tilde{\theta}}(\theta_t, \tau_t) \right) + 2\alpha_1 L^2 R \Delta_t^{\theta, \tau}$$

*where the synchronization error is defined as:*

$$\Delta_t^{\theta, \tau} := \frac{1}{N} \sum_{i=1}^N \sum_{r=1}^R \left( \|\theta_{t,r}^i - \theta_t\|^2 + \|\tau_{t,r}^i - \tau_t\|^2 \right).$$

*Proof.* Using Lemma 3, $\Phi(\cdot)$ is $L_\Phi = \kappa L/2 + L$-smooth, and together with the updating rule, we have:

$$\Phi(\theta_{t+1}) \leq \Phi(\theta_t) + \langle \nabla\Phi(\theta_t), \theta_{t+1} - \theta_t \rangle + \frac{L_\Phi}{2} \|\theta_{t+1} - \theta_t\|^2$$

$$\leq \Phi(\theta_t) - \alpha_1 \left\langle \nabla\Phi(\theta_t), \frac{1}{N} \sum_{i=1}^N \sum_{r=1}^R \nabla_\theta U_{\tilde{\theta}}^i(\theta_{t,r}^i, \tau_{t,r}^i) \right\rangle + \frac{L_\Phi}{2} \alpha_1^2 \left\| \frac{1}{N} \sum_{i=1}^N \sum_{r=1}^R \nabla_\theta U_{\tilde{\theta}}^i(\theta_{t,r}^i, \tau_{t,r}^i) \right\|^2$$

Using the identity $\langle \boldsymbol{a}, \boldsymbol{b} \rangle = -\frac{1}{2}\|\boldsymbol{a} - \boldsymbol{b}\|^2 + \frac{1}{2}\|\boldsymbol{a}\|^2 + \frac{1}{2}\|\boldsymbol{b}\|^2$, we have:

$$\Phi(\theta_{t+1}) - \Phi(\theta_t)$$

$$\leq -\frac{\alpha_1}{2}\|\nabla\Phi(\theta_t)\|^2 - \frac{\alpha_1}{2}\left\|\frac{1}{N}\sum_{i=1}^{N}\sum_{r=1}^{R}\nabla_\theta U_{\tilde{\theta}}^i(\theta_{t,r}^i, \tau_{t,r}^i)\right\|^2 + \alpha_1\left\|\nabla\Phi(\theta_t) - \frac{1}{N}\sum_{i=1}^{N}\nabla_\theta U_{\tilde{\theta}}^i(\theta_t, \tau_t)\right\|^2$$

$$+ \alpha_1\left\|\frac{1}{N}\sum_{i=1}^{N}\sum_{r=1}^{R}\nabla_\theta U_{\tilde{\theta}}^i(\theta_{t,r}^i, \tau_{t,r}^i) - \frac{1}{N}\sum_{i=1}^{N}\nabla_\theta U_{\tilde{\theta}}^i(\theta_t, \tau_t)\right\|^2 + \frac{L_\Phi}{2}\alpha_1^2\left\|\frac{1}{N}\sum_{i=1}^{N}\sum_{r=1}^{R}\nabla_\theta U_{\tilde{\theta}}^i(\theta_{t,r}^i, \tau_{t,r}^i)\right\|^2$$

$$\leq -\frac{\alpha_1}{2}\|\nabla\Phi(\theta_t)\|^2 - \frac{\alpha_1}{2}(1 - L_\Phi\alpha_1)\left\|\frac{1}{N}\sum_{i=1}^{N}\sum_{r=1}^{R}\nabla_\theta U_{\tilde{\theta}}^i(\theta_{t,r}^i, \tau_{t,r}^i)\right\|^2 + \alpha_1 L^2\|\phi(\theta_t) - \tau_t\|^2$$

$$+ \alpha_1 L^2\frac{R}{N}\sum_{i=1}^{N}\sum_{r=1}^{R}(2\left\|\theta_{t,r}^i - \theta_t\right\|^2 + 2\left\|\tau_{t,r}^i - \tau_t\right\|^2) \tag{22}$$

$$\leq -\frac{\alpha_1}{2}\|\nabla\Phi(\theta_t)\|^2 - \frac{\alpha_1}{2}(1 - L_\Phi\alpha_1)\left\|\frac{1}{N}\sum_{i=1}^{N}\sum_{r=1}^{R}\nabla_\theta U_{\tilde{\theta}}^i(\theta_{t,r}^i, \tau_{t,r}^i)\right\|^2$$

$$+ \alpha_1 L^2\|\phi(\theta_t) - \tau_t\|^2 + 2\alpha_1 L^2 R\Delta_t^{\theta,\tau} \tag{23}$$

Using the quadratic growth property of $\mu$-PL function $U_{\tilde{\theta}}(\theta, \cdot)$, i.e., $\frac{\mu}{2}\|\tau - \Phi(\theta)\|^2 \leq \max_{\tau'} U_{\tilde{\theta}}(\theta, \tau') - U_{\tilde{\theta}}(\theta, \tau)$, $\forall\, \theta, \tau$, where $\Phi(\theta) := \arg\max_{\tau'} U_{\tilde{\theta}}(\theta, \tau')$, we have

$$\Phi(\theta_{t+1}) - \Phi(\theta_t)$$

$$\leq -\frac{\alpha_1}{2}\|\nabla\Phi(\theta_t)\|^2 - \frac{\alpha_1}{2}(1 - L_\Phi\alpha_1)\left\|\frac{1}{N}\sum_{i=1}^{N}\sum_{r=1}^{R}\nabla_\theta U_{\tilde{\theta}}^i(\theta_{t,r}^i, \tau_{t,r}^i)\right\|^2$$

$$+ \frac{2\alpha_1 L^2}{\mu}\left(\Phi(\theta_t) - U_{\tilde{\theta}}(\theta_t, \tau_t)\right) + 2\alpha_1 L^2 R\Delta_t^{\theta,\tau} \tag{24}$$

$\square$

**Lemma 5.** *(Sharma et al., 2022) Suppose the local loss functions $\{U_{\tilde{\theta}}^i\}$ satisfy Assumptions 2 and 3. Further, in Algorithm 1, we choose step-sizes $\alpha_1, \alpha_2$ satisfying $\alpha_2 \leq 1/\mu$, $\frac{\alpha_1}{\alpha_2} \leq \frac{1}{8\kappa^2}$. Then the following inequality holds.*

$$\frac{1}{T}\sum_{t=1}^{T}\left(\Phi(\theta_t) - U_{\tilde{\theta}}(\theta_t, \tau_t)\right)$$

$$\leq \frac{2\left(\Phi(\theta_1) - U_{\tilde{\theta}}(\theta_1, \tau_1)\right)}{\alpha_2\mu T} + \frac{2L^2 R}{\mu\alpha_2}\left(2\alpha_1(1 - \alpha_2\mu) + \alpha_2\right)\frac{1}{T}\sum_{t=1}^{T}\Delta_t^{\theta,\tau} + (1 - \alpha_2\mu)\frac{\alpha_1}{\alpha_2\mu}\frac{1}{T}\sum_{t=1}^{T}\|\nabla\Phi(\theta_t)\|^2$$

$$+ \left[(1 - \alpha_2\mu)\frac{\alpha_1^2}{2}(L + L_\Phi) + \alpha_2 L^2\alpha_1^2\right]\frac{2}{\alpha_2\mu T}\sum_{t=1}^{T}\left\|\frac{1}{N}\sum_{i=1}^{N}\sum_{r=1}^{R}\nabla_\theta U_{\tilde{\theta}}^i(\theta_{t,r}^i, \tau_{t,r}^i)\right\|^2$$

*Proof.* Using $L$-smoothness of $U_{\tilde{\theta}}(\theta, \cdot)$

$$U_{\tilde{\theta}}(\theta_{t+1}, \tau_t) + \left\langle \nabla_\tau U_{\tilde{\theta}}(\theta_{t+1}, \tau_t), \tau_{t+1} - \tau_t \right\rangle - \frac{L}{2}\|\tau_{t+1} - \tau_t\|^2 \leq U_{\tilde{\theta}}(\theta_{t+1}, \tau_{t+1})$$

Using the update rule in Algorithm 1

$$U_{\tilde{\theta}}(\theta_{t+1}, \tau_t) \leq U_{\tilde{\theta}}(\theta_{t+1}, \tau_{t+1}) - \alpha_2 \left\langle \nabla_\tau U_{\tilde{\theta}}(\theta_{t+1}, \tau_t), \frac{1}{N} \sum_{i=1}^N \sum_{r=1}^R \nabla_\tau U_{\tilde{\theta}}^i(\theta_{t,r}^i, \tau_{t,r}^i) \right\rangle$$

$$+ \frac{\alpha_2^2 L}{2} \left\| \frac{1}{N} \sum_{i=1}^N \sum_{r=1}^R \nabla_\tau U_{\tilde{\theta}}^i(\theta_{t,r}^i, \tau_{t,r}^i) \right\|^2$$

$$= U_{\tilde{\theta}}(\theta_{t+1}, \tau_{t+1}) - \frac{\alpha_2}{2} \left\| \nabla_\tau U_{\tilde{\theta}}(\theta_{t+1}, \tau_t) \right\|^2 - \frac{\alpha_2}{2} (1 - \alpha_2 L) \left\| \frac{1}{N} \sum_{i=1}^N \sum_{r=1}^R \nabla_\tau U_{\tilde{\theta}}^i(\theta_{t,r}^i, \tau_{t,r}^i) \right\|^2$$

$$+ \frac{\alpha_2}{2} \left\| \nabla_\tau U_{\tilde{\theta}}(\theta_{t+1}, \tau_t) - \nabla_\tau U_{\tilde{\theta}}(\theta_t, \tau_t) + \nabla_\tau U_{\tilde{\theta}}(\theta_t, \tau_t) - \frac{1}{N} \sum_{i=1}^N \sum_{r=1}^R \nabla_\tau U_{\tilde{\theta}}^i(\theta_{t,r}^i, \tau_{t,r}^i) \right\|^2$$

$$\leq U_{\tilde{\theta}}(\theta_{t+1}, \tau_{t+1}) - \frac{\alpha_2}{2} \left\| \nabla_\tau U_{\tilde{\theta}}(\theta_{t+1}, \tau_t) \right\|^2 - \frac{\alpha_2}{2} (1 - \alpha_2 L) \left\| \frac{1}{N} \sum_{i=1}^N \sum_{r=1}^R \nabla_\tau U_{\tilde{\theta}}^i(\theta_{t,r}^i, \tau_{t,r}^i) \right\|^2$$

$$+ \alpha_2 L^2 \left\| \theta_{t+1} - \theta_t \right\|^2 + \alpha_2 L^2 R \Delta_t^{\theta, \tau}, \tag{25}$$

Note that

$$\left\| \theta_{t+1} - \theta_t \right\|^2 = \alpha_1^2 \left\| \frac{1}{N} \sum_{i=1}^N \sum_{r=1}^R \nabla_\theta U_{\tilde{\theta}}^i(\theta_{t,r}^i, \tau_{t,r}^i) \right\|^2. \tag{26}$$

Also, using Assumption 5,

$$\left\| \nabla_\tau U_{\tilde{\theta}}(\theta_{t+1}, \tau_t) \right\|^2 \geq 2\mu \left( \max_\tau U_{\tilde{\theta}}(\theta_{t+1}, \tau) - U_{\tilde{\theta}}(\theta_{t+1}, \tau_t) \right) = 2\mu \left( \Phi(\theta_{t+1}) - U_{\tilde{\theta}}(\theta_{t+1}, \tau_t) \right). \tag{27}$$

Substituting (26), (27) in (25), and rearranging the terms, we get

$$\alpha_2 \mu \left( \Phi(\theta_{t+1}) - U_{\tilde{\theta}}(\theta_{t+1}, \tau_t) \right)$$

$$\leq U_{\tilde{\theta}}(\theta_{t+1}, \tau_{t+1}) - U_{\tilde{\theta}}(\theta_{t+1}, \tau_t) - \frac{\alpha_2}{2} (1 - \alpha_2 L) \left\| \frac{1}{N} \sum_{i=1}^N \sum_{r=1}^R \nabla_\tau U_{\tilde{\theta}}^i(\theta_{t,r}^i, \tau_{t,r}^i) \right\|^2$$

$$+ \alpha_2 L^2 \left[ \alpha_1^2 \left\| \frac{1}{N} \sum_{i=1}^N \sum_{r=1}^R \nabla_\theta U_{\tilde{\theta}}^i(\theta_{t,r}^i, \tau_{t,r}^i) \right\|^2 \right] + \alpha_2 L^2 R \Delta_t^{\theta, \tau}$$

$$\Rightarrow \left( \Phi(\theta_{t+1}) - U_{\tilde{\theta}}(\theta_{t+1}, \tau_{t+1}) \right)$$

$$\leq (1 - \alpha_2 \mu) \left( \Phi(\theta_{t+1}) - U_{\tilde{\theta}}(\theta_{t+1}, \tau_t) \right) - \frac{\alpha_2}{2} (1 - \alpha_2 L) \left\| \frac{1}{N} \sum_{i=1}^N \sum_{r=1}^R \nabla_\tau U_{\tilde{\theta}}^i(\theta_{t,r}^i, \tau_{t,r}^i) \right\|^2$$

$$+ \alpha_1^2 \alpha_2 L^2 \left\| \frac{1}{N} \sum_{i=1}^N \sum_{r=1}^R \nabla_\theta U_{\tilde{\theta}}^i(\theta_{t,r}^i, \tau_{t,r}^i) \right\|^2 + \alpha_2 L^2 R \Delta_t^{\theta, \tau}. \tag{28}$$

Next, we bound $\left( \Phi(\theta_{t+1}) - U_{\tilde{\theta}}(\theta_{t+1}, \tau_t) \right)$.

$$\Phi(\theta_{t+1}) - U_{\tilde{\theta}}(\theta_{t+1}, \tau_t)$$
$$= \underbrace{\left( \Phi(\theta_{t+1}) - \Phi(\theta_t) \right)}_{T_1} + \left( \Phi(\theta_t) - U_{\tilde{\theta}}(\theta_t, \tau_t) \right) + \underbrace{\left( U_{\tilde{\theta}}(\theta_t, \tau_t) - U_{\tilde{\theta}}(\theta_{t+1}, \tau_t) \right)}_{T_2} \tag{29}$$

$T_1$ is bounded in Lemma 4. We next bound $T_2$. Using $L$-smoothness of $U_{\tilde{\theta}}(\cdot, \tau_t)$,

$$U_{\tilde{\theta}}(\theta_t, \tau_t) + \left\langle \nabla_\theta U_{\tilde{\theta}}(\theta_t, \tau_t), \theta_{t+1} - \theta_t \right\rangle - \frac{L}{2} \left\| \theta_{t+1} - \theta_t \right\|^2 \leq U_{\tilde{\theta}}(\theta_{t+1}, \tau_t)$$

$$\Rightarrow T_2 = \left( U_{\tilde{\theta}}(\theta_t, \tau_t) - U_{\tilde{\theta}}(\theta_{t+1}, \tau_t) \right)$$

$$\leq \alpha_1 \left\langle \nabla_\theta U_{\tilde{\theta}}(\theta_t, \tau_t), \frac{1}{N} \sum_{i=1}^{N} \sum_{r=1}^{R} \nabla_\theta U_{\tilde{\theta}}^i(\theta_{t,r}^i, \tau_{t,r}^i) \right\rangle + \frac{\alpha_1^2 L}{2} \left\| \frac{1}{N} \sum_{i=1}^{N} \sum_{r=1}^{R} \nabla_\theta U_{\tilde{\theta}}^i(\theta_{t,r}^i, \tau_{t,r}^i) \right\|^2$$

$$\leq \frac{\alpha_1}{2} \left( \left\| \nabla_\theta U_{\tilde{\theta}}(\theta_t, \tau_t) \right\|^2 + \left\| \frac{1}{N} \sum_{i=1}^{N} \sum_{r=1}^{R} \nabla_\theta U_{\tilde{\theta}}^i(\theta_{t,r}^i, \tau_{t,r}^i) \right\|^2 \right) + \frac{\alpha_1^2 L}{2} \left\| \frac{1}{N} \sum_{i=1}^{N} \sum_{r=1}^{R} \nabla_\theta U_{\tilde{\theta}}^i(\theta_{t,r}^i, \tau_{t,r}^i) \right\|^2$$

$$\leq \alpha_1 \left( \left\| \nabla \Phi(\theta_t) \right\|^2 + \left\| \nabla_\theta U_{\tilde{\theta}}(\theta_t, \tau_t) - \nabla \Phi(\theta_t) \right\|^2 \right) + \frac{\alpha_1}{2} \left( 1 + \alpha_1 L \right) \left\| \frac{1}{N} \sum_{i=1}^{N} \sum_{r=1}^{R} \nabla_\theta U_{\tilde{\theta}}^i(\theta_{t,r}^i, \tau_{t,r}^i) \right\|^2$$

$$\overset{(a)}{\leq} \alpha_1 \left\| \nabla \Phi(\theta_t) \right\|^2 + \alpha_1 L^2 \left\| \tau_t - \tau^*(\theta_t) \right\|^2 + \frac{\alpha_1}{2} \left( 1 + \alpha_1 L \right) \left\| \frac{1}{N} \sum_{i=1}^{N} \sum_{r=1}^{R} \nabla_\theta U_{\tilde{\theta}}^i(\theta_{t,r}^i, \tau_{t,r}^i) \right\|^2$$

$$\leq \alpha_1 \left\| \nabla \Phi(\theta_t) \right\|^2 + \frac{2\alpha_1 L^2}{\mu} \left( \Phi(\theta_t) - U_{\tilde{\theta}}(\theta_t, \tau_t) \right) + \frac{\alpha_1}{2} \left( 1 + \alpha_1 L \right) \left\| \frac{1}{N} \sum_{i=1}^{N} \nabla_\theta U_{\tilde{\theta}}^i(\theta_{t,r}^i, \tau_{t,r}^i) \right\|^2 .$$
$$(30)$$

where $(a)$ follows from Assumption 2 and Assumption 3. Also, recall that $\tau^*(\theta) \in \arg\max_{\tau'} U_{\tilde{\theta}}(\theta, \tau')$. (30) follows from the quadratic growth property of $\mu$-PL functions. Substituting the bounds on $T_1, T_2$ from Lemma 4 and (30) respectively, in (28), we get

$$\left( \Phi(\theta_{t+1}) - U_{\tilde{\theta}}(\theta_{t+1}, \tau_{t+1}) \right)$$

$$\leq (1 - \alpha_2 \mu) \left( 1 + \frac{4\alpha_1 L^2}{\mu} \right) \left( \Phi(\theta_t) - U_{\tilde{\theta}}(\theta_t, \tau_t) \right)$$

$$+ (1 - \alpha_2 \mu) \left[ -\frac{\alpha_1}{2} \left\| \nabla \Phi(\theta_t) \right\|^2 - \frac{\alpha_1}{2} \left( 1 - L_\Phi \alpha_1 \right) \left\| \frac{1}{N} \sum_{i=1}^{N} \sum_{r=1}^{R} \nabla_\theta U_{\tilde{\theta}}^i(\theta_{t,r}^i, \tau_{t,r}^i) \right\|^2 + 2\alpha_1 L^2 R \Delta_t^{\theta,\tau} \right]$$

$$+ (1 - \alpha_2 \mu) \left[ \alpha_1 \left\| \nabla \Phi(\theta_t) \right\|^2 + \frac{\alpha_1}{2} \left( 1 + \alpha_1 L \right) \left\| \frac{1}{N} \sum_{i=1}^{N} \sum_{r=1}^{R} \nabla_\theta U_{\tilde{\theta}}^i(\theta_{t,r}^i, \tau_{t,r}^i) \right\|^2 \right]$$

$$- \frac{\alpha_2}{2} \left( 1 - \alpha_2 L \right) \left\| \frac{1}{N} \sum_{i=1}^{N} \sum_{r=1}^{R} \nabla_\tau U_{\tilde{\theta}}^i(\theta_{t,r}^i, \tau_{t,r}^i) \right\|^2$$

$$+ \alpha_1^2 \alpha_2 L^2 \left\| \frac{1}{N} \sum_{i=1}^{N} \nabla_\theta U_{\tilde{\theta}}^i(\theta_{t,r}^i, \tau_{t,r}^i) \right\|^2 + \alpha_2 L^2 R \Delta_t^{\theta,\tau}$$

$$\leq \left( 1 - \frac{\alpha_2 \mu}{2} \right) \left( \Phi(\theta_t) - U_{\tilde{\theta}}(\theta_t, \tau_t) \right) + \alpha_2 L^2 R \Delta_t^{\theta,\tau}$$

$$+ \left[ (1 - \alpha_2 \mu) \frac{\alpha_1^2}{2} \left( L + L_\Phi \right) + \alpha_2 L^2 \alpha_1^2 \right] \left\| \frac{1}{N} \sum_{i=1}^{N} \sum_{r=1}^{R} \nabla_\theta U_{\tilde{\theta}}^i(\theta_{t,r}^i, \tau_{t,r}^i) \right\|^2$$

$$+ (1 - \alpha_2 \mu) \left[ \frac{\alpha_1}{2} \left\| \nabla \Phi(\theta_t) \right\|^2 + 2\alpha_1 L^2 R \Delta_t^{\theta,\tau} \right],$$
$$(31)$$

where we choose $\alpha_1$ such that $(1 - \alpha_2\mu)\left(1 + \frac{4\alpha_1 L^2}{\mu}\right) \le \left(1 - \frac{\alpha_2\mu}{2}\right)$. This holds if $\frac{4\alpha_1 L^2}{\mu} \le \frac{\alpha_2\mu}{2} \Rightarrow$
$\alpha_1 \le \frac{\alpha_2}{8\kappa^2}$. Summing (31) over $t = 1, \ldots, T$, and rearranging the terms, we get

$$\frac{1}{T} \sum_{t=1}^{T} \left( \Phi(\theta_{t+1}) - U_{\tilde{\theta}}(\theta_{t+1}, \tau_{t+1}) \right)$$

$$\le \left(1 - \frac{\alpha_2\mu}{2}\right) \frac{1}{T} \sum_{t=1}^{T} \left( \Phi(\theta_t) - U_{\tilde{\theta}}(\theta_t, \tau_t) \right) + L^2 R \left(2\alpha_1(1 - \alpha_2\mu) + \alpha_2\right) \frac{1}{T} \sum_{t=1}^{T} \Delta_t^{\theta,\tau}$$

$$+ \left[ (1 - \alpha_2\mu)\frac{\alpha_1^2}{2}(L + L_\Phi) + \alpha_2 L^2 \alpha_1^2 \right] \frac{1}{T} \sum_{t=1}^{T} \left\| \frac{1}{N} \sum_{i=1}^{N} \sum_{r=1}^{R} \nabla_\theta U_{\tilde{\theta}}^i(\theta_{t,r}^i, \tau_{t,r}^i) \right\|^2 + (1 - \alpha_2\mu)\frac{\alpha_1}{2} \frac{1}{T} \sum_{t=1}^{T} \|\nabla\Phi(\theta_t)\|^2$$

Rearranging the terms, we get

$$\frac{1}{T} \sum_{t=1}^{T} \left( \Phi(\theta_t) - U_{\tilde{\theta}}(\theta_t, \tau_t) \right)$$

$$\le \frac{2}{\alpha_2\mu} \left[ \frac{\Phi(\theta_1) - U_{\tilde{\theta}}(\theta_1, \tau_1)}{T} - \frac{\left( \Phi(\theta_{T+1}) - U_{\tilde{\theta}}(\theta_{T+1}, \tau_{T+1}) \right)}{T} \right] + \frac{2L^2 R}{\alpha_2\mu} \left(2\alpha_1(1 - \alpha_2\mu) + \alpha_2\right) \frac{1}{T} \sum_{t=1}^{T} \Delta_t^{\theta,\tau}$$

$$+ \left[ (1 - \alpha_2\mu)\frac{\alpha_1^2}{2}(L + L_\Phi) + \alpha_2 L^2 \alpha_1^2 \right] \frac{2}{\alpha_2\mu T} \sum_{t=1}^{T} \left\| \frac{1}{N} \sum_{i=1}^{N} \sum_{r=1}^{R} \nabla_\theta U_{\tilde{\theta}}^i(\theta_{t,r}^i, \tau_{t,r}^i) \right\|^2 + (1 - \alpha_2\mu)\frac{\alpha_1}{\alpha_2\mu T} \sum_{t=1}^{T} \|\nabla\Phi(\theta_t)\|^2$$

$$\le \frac{2\left( \Phi(\theta_1) - U_{\tilde{\theta}}(\theta_1, \tau_1) \right)}{\alpha_2\mu T} + \frac{2L^2 R}{\alpha_2\mu} \left(2\alpha_1(1 - \alpha_2\mu) + \alpha_2\right) \frac{1}{T} \sum_{t=1}^{T} \Delta_t^{\theta,\tau}$$

$$+ \left[ (1 - \alpha_2\mu)\frac{\alpha_1^2}{2}(L + L_\Phi) + \alpha_2 L^2 \alpha_1^2 \right] \frac{2}{\alpha_2\mu T} \sum_{t=1}^{T} \left\| \frac{1}{N} \sum_{i=1}^{N} \sum_{r=1}^{R} \nabla_\theta U_{\tilde{\theta}}^i(\theta_{t,r}^i, \tau_{t,r}^i) \right\|^2 + (1 - \alpha_2\mu)\frac{\alpha_1}{\alpha_2\mu T} \sum_{t=1}^{T} \|\nabla\Phi(\theta_t)\|^2$$

since $\Phi(\theta_T) := \arg\max_\tau U_{\tilde{\theta}}(\theta_T, \tau)$, which concludes the proof. $\qquad\square$

**Lemma 6.** *Suppose the local loss functions $\{U_{\tilde{\theta}}^i\}$ satisfy Assumptions 2 and 3. Further, in Algorithm 1, using bounded gradient assumption, i.e., $\|\nabla U_{\tilde{\theta}}^i(\theta, \tau)\| \le G$, we choose step-sizes $\alpha_1, \alpha_2 \le \frac{1}{8RL}$. Then, the iterates $\{\theta_t, \tau_t\}$ generated by Algorithm 1 satisfy*

$$\frac{1}{T} \sum_{t=1}^{T} \Delta_t^{\theta,\tau} := \frac{1}{T} \sum_{t=1}^{T} \frac{1}{N} \sum_{i=1}^{N} \sum_{r=1}^{R} \left( \left\| \theta_{t,r}^i - \theta_t \right\|^2 + \left\| \tau_{t,r}^i - \tau_t \right\|^2 \right)$$
$$\le 6R(R-1)^2 \left[ (\zeta_\theta^2 \alpha_1^2 + \zeta_\tau^2 \alpha_2^2) + (\alpha_1^2 + \alpha_2^2)G^2 \right].$$

*Proof of Lemma 6.* We define the separate synchronization errors for $\theta$ and $\tau$

$$\Delta_t^\theta := \frac{1}{N} \sum_{i=1}^{N} \sum_{r=1}^{R} \left\| \theta_{t,r}^i - \theta_t \right\|^2, \qquad \Delta_t^\tau := \frac{1}{N} \sum_{i=1}^{N} \sum_{r=1}^{R} \left\| \tau_{t,r}^i - \tau_t \right\|^2,$$

such that $\Delta_t^{\theta,\tau} = \Delta_t^\theta + \Delta_t^\tau$. We first bound the $\theta$- synchronization error $\Delta_t^\theta$. Then,

$$\Delta_t^\theta := \frac{1}{N} \sum_{i=1}^{N} \sum_{r=1}^{R} \left\| \theta_{t,r}^i - \theta_t \right\|^2$$

$$= \alpha_1^2 \frac{1}{N} \sum_{i=1}^{N} \sum_{r=1}^{R} \left\| \sum_{j=1}^{r} \nabla_\theta U_{\tilde{\theta}}^i(\theta_{t,j}^i, \tau_{t,j}^i) \right\|^2$$

$$\le \alpha_1^2 \frac{1}{N} \sum_{i=1}^{N} \sum_{r=1}^{R} (r-1) \sum_{j=1}^{r} \left\| \nabla_\theta U_{\tilde{\theta}}^i(\theta_{t,j}^i, \tau_{t,j}^i) \right\|^2.$$

This can be written as

$$\Delta_t^\theta \overset{(a)}{\leq} \alpha_1^2 \frac{1}{N} \sum_{i=1}^N \sum_{j=1}^{R-1} \left\| \nabla_\theta U_{\tilde{\theta}}^i(\theta_{t,j}^i, \tau_{t,j}^i) \right\|^2 \sum_{r=j+1}^R (r-1)$$

$$\overset{(b)}{\leq} \alpha_1^2 \frac{(R-1)^2}{2} \frac{1}{N} \sum_{i=1}^N \sum_{j=1}^{R-1} \left\| \nabla_\theta U_{\tilde{\theta}}^i(\theta_{t,j}^i, \tau_{t,j}^i) \right\|^2$$

$$\overset{(c)}{\leq} \alpha_1^2 \frac{(R-1)^2}{2} \frac{1}{N} \sum_{i=1}^N \sum_{j=1}^R \left\| \nabla_\theta U_{\tilde{\theta}}^i(\theta_{t,j}^i, \tau_{t,j}^i) \right\|^2,$$

where $(a)$ follows from rewriting the sum, $(b)$ follows since $\sum_{r=j+1}^R (r-1) \leq \frac{R^2}{2}$ and $(c)$ follows because a positive quantity is being added. Now,

$$\Delta_t^\theta \leq \alpha_1^2 \frac{(R-1)^2}{2} \frac{1}{N} \sum_{i=1}^N \sum_{r=1}^R \left\| \nabla_\theta U_{\tilde{\theta}}^i(\theta_{t,r}^i, \tau_{t,r}^i) - \nabla_\theta U_{\tilde{\theta}}^i(\theta_t, \tau_t) + \nabla_\theta U_{\tilde{\theta}}^i(\theta_t, \tau_t) - \nabla_\theta U_{\tilde{\theta}}(\theta_t, \tau_t) + \nabla_\theta U_{\tilde{\theta}}(\theta_t, \tau_t) \right\|^2$$

$$\leq \alpha_1^2 \frac{(R-1)^2}{2} \frac{1}{N} \sum_{i=1}^N \sum_{r=1}^R \left[ 3 \left\| \nabla_\theta U_{\tilde{\theta}}^i(\theta_{t,r}^i, \tau_{t,r}^i) - \nabla_\theta U_{\tilde{\theta}}^i(\theta_t, \tau_t) \right\|^2 + 3 \left\| \nabla_\theta U_{\tilde{\theta}}^i(\theta_t, \tau_t) - \nabla_\theta U_{\tilde{\theta}}(\theta_t, \tau_t) \right\|^2 \right]$$

$$+ 3\alpha_1^2 \frac{R(R-1)^2}{2} \left\| \nabla_\theta U_{\tilde{\theta}}(\theta_t, \tau_t) \right\|^2$$

$$\leq 6L^2 \alpha_1^2 \frac{(R-1)^2}{2} \Delta_t^{\theta,\tau} + 3\zeta_\theta^2 \alpha_1^2 \frac{R(R-1)^2}{2} + 3\alpha_1^2 \frac{R(R-1)^2}{2} \left\| \nabla_\theta U_{\tilde{\theta}}(\theta_t, \tau_t) \right\|^2. \qquad (32)$$

Similarly, for the synchronization error $\Delta_t^\tau$, we have

$$\Delta_t^\tau \leq 6L^2 \alpha_2^2 \frac{(R-1)^2}{2} \Delta_t^{\theta,\tau} + 3\zeta_\tau^2 \alpha_2^2 \frac{R(R-1)^2}{2} + 3\alpha_2^2 \frac{R(R-1)^2}{2} \left\| \nabla_\tau U_{\tilde{\theta}}(\theta_t, \tau_t) \right\|^2. \qquad (33)$$

Using bounded gradient assumption, i.e., $\left\| \nabla U_{\tilde{\theta}}^i(\theta, \tau) \right\| \leq G$, and adding (32) and (33), we obtain

$$\Delta_t^{\theta,\tau} \leq 6L^2(\alpha_1^2 + \alpha_2^2)(R-1)^2 \Delta_t^{\theta,\tau} + 3(\zeta_\theta^2 \alpha_1^2 + \zeta_\tau^2 \alpha_2^2)R(R-1)^2 + 3(\alpha_1^2 + \alpha_2^2)R(R-1)^2 G^2.$$

For our choice of $\alpha_1$ and $\alpha_2$, we have $6L^2(\alpha_1^2 + \alpha_2^2)(R-1)^2 \leq \frac{1}{2}$, thus

$$\Delta_t^{\theta,\tau} \leq 6R(R-1)^2 \left[ (\zeta_\theta^2 \alpha_1^2 + \zeta_\tau^2 \alpha_2^2) + (\alpha_1^2 + \alpha_2^2)G^2 \right].$$

Averaging across all the communication rounds $t = 1, \ldots, T$ proves the lemma.

$\square$

**Theorem 4** (Convergence of FEDGDA). *Suppose the local loss functions $\{U_{\tilde{\theta}}^i\}_i$ satisfy Assumptions 2,3 and have bounded gradients, and the global function $U_{\tilde{\theta}}$ satisfies 5. Suppose the step-sizes $\alpha_1, \alpha_2$ are chosen such that $\alpha_2 \leq \frac{1}{8LR}$, $\frac{\alpha_1}{\alpha_2} = \frac{1}{8\kappa^2}$, where $\kappa = \frac{L}{\mu}$ is the condition number. Then for the output $\bar{\theta}_T$ of Algorithm 1, the following holds.*

$$\left\| \nabla \Phi(\bar{\theta}_T) \right\|^2 = \frac{1}{T} \sum_{t=1}^T \left\| \nabla \Phi(\theta_t) \right\|^2$$

$$\leq \underbrace{\mathcal{O}\left( \kappa^2 \frac{\Delta_\Phi}{\alpha_2 T} \right)}_{\text{Error with full synchronization}} + \underbrace{\mathcal{O}\left( L^2 \kappa^2 R(R-1)^2 \left[ \alpha_2^2 \left( G^2 + \zeta_\tau^2 \right) + \alpha_1^2 \zeta_\theta^2 \right] \right)}_{\text{Error due to local updates}}, \quad (34)$$

*where $\Phi(\theta) := \max_\tau U_{\tilde{\theta}}(\theta, \tau)$ is the envelope function, $\Delta_\Phi := \Phi(\theta_1) - \min_\theta \Phi(\theta)$. Using $\alpha_2 = \sqrt{\frac{N}{LT}}$ and $\alpha_1 = \frac{1}{8\kappa^2} \sqrt{\frac{N}{LT}}$, we get*

$$\left\| \nabla \Phi(\bar{\theta}_T) \right\|^2 \leq \mathcal{O}\left( \frac{\kappa^2 \Delta_\Phi}{\sqrt{NT}} + \kappa^2 R(R-1)^2 \frac{N \left( \sigma^2 + \zeta_\theta^2 + \zeta_\tau^2 \right)}{T} \right).$$

*Proof.* We start by summing the expression in Lemma 4 over $t = 1, \ldots, T$.

$$\frac{1}{T}\sum_{t=1}^{T}\left(\Phi(\theta_{t+1}) - \Phi(\theta_t)\right) \leq -\frac{\alpha_1}{2}\frac{1}{T}\sum_{t=1}^{T}\|\nabla\Phi(\theta_t)\|^2 - \frac{\alpha_1}{2}\left(1 - L_\Phi\alpha_1\right)\frac{1}{T}\sum_{t=1}^{T}\left\|\frac{1}{N}\sum_{i=1}^{n}\sum_{r=1}^{R}\nabla_\theta U_{\tilde{\theta}}^i(\theta_{t,r}^i, \tau_{t,r}^i)\right\|^2$$

$$+ \frac{2\alpha_1 L^2}{\mu}\frac{1}{T}\sum_{t=1}^{T}\left[\Phi(\theta_t) - U_{\tilde{\theta}}(\theta_t, \tau_t)\right] + 2\alpha_1 L^2 R\frac{1}{T}\sum_{t=1}^{T}\Delta_t^{\theta,\tau}. \quad (35)$$

Substituting the bound on $\frac{1}{T}\sum_{t=1}^{T}\Delta_t^{\theta,\tau}$ from Lemma 6, and the bound on $\frac{1}{T}\sum_{t=1}^{T}\left(\Phi(\theta_t) - U_{\tilde{\theta}}(\theta_t, \tau_t)\right)$ from Lemma 5, and rearranging the terms in (35), we get

$$\frac{1}{T}\left(\Phi(\theta_T) - \Phi(\theta_1)\right)$$

$$\leq -\underbrace{\left(\frac{\alpha_1}{2} - (1 - \alpha_2\mu)\frac{2\alpha_1^2 L^2}{\alpha_2\mu^2}\right)}_{\geq \alpha_1/4}\frac{1}{T}\sum_{t=1}^{T}\|\nabla\Phi(\theta_t)\|^2$$

$$-\underbrace{\frac{\alpha_1}{2}\left(1 - L_\Phi\alpha_1 - \frac{8L^2}{\mu^2\alpha_2}\left[(1 - \alpha_2\mu)\frac{\alpha_1^2}{2}\left(L + L_\Phi\right) + \alpha_2 L^2\alpha_1^2\right]\right)}_{\geq 0}\frac{1}{T}\sum_{t=1}^{T}\left\|\frac{1}{N}\sum_{i=1}^{n}\nabla_\theta U_{\tilde{\theta}}^i(\theta_{t,r}^i, \tau_{t,r}^i)\right\|^2$$

$$+ \left[\frac{2\alpha_1 L^2}{\mu}\left(\frac{2L^2 R}{\mu} + \frac{4\alpha_1 R L^2(1 - \alpha_2\mu)}{\mu\alpha_2}\right) + 2\alpha_1 L^2 R\right]\frac{1}{T}\sum_{t=1}^{T}\Delta_t^{\theta,\tau}$$

$$+ \frac{4\alpha_1 L^2}{\mu}\frac{\left(\Phi(\theta_1) - U_{\tilde{\theta}}(\theta_1, \tau_1)\right)}{\alpha_2\mu T}. \quad (36)$$

Here, $\frac{\alpha_1}{2} - \frac{2\alpha_1^2(1 - \mu\alpha_2)L^2}{\mu^2\alpha_2} \geq \frac{\alpha_1}{4}$ holds since $\frac{\alpha_1}{\alpha_2} \leq \frac{1}{8\kappa^2}$. Also, $1 - L_\Phi\alpha_1 - \frac{8L^2}{\mu^2\alpha_2}\left[(1 - \alpha_2\mu)\frac{\alpha_1^2}{2}\left(L + L_\Phi\right) + \alpha_2 L^2\alpha_1^2\right] \geq 0$ follows from the bounds on $\alpha_1, \alpha_2$. Rearranging the terms in Equation (36) and using lemma (6), we get

$$\frac{1}{T}\sum_{t=1}^{T}\|\nabla\Phi(\theta_t)\|^2 \leq \frac{4\left(\Phi(\theta_1) - \Phi(\theta_T)\right)}{\alpha_1 T}$$

$$+ \frac{4}{\alpha_1}2\alpha_1 L^2 R\left[1 + 2\kappa^2 + 4\kappa^2\frac{\alpha_1}{\alpha_2}\right]6R(R - 1)^2\left[\left(\alpha_1^2 + \alpha_2^2\right)G^2 + \left(\alpha_1^2\zeta_\theta^2 + \alpha_2^2\zeta_\tau^2\right)\right]$$

$$+ \frac{4}{\alpha_1}\frac{4\alpha_1\kappa^2}{\alpha_2}\frac{\left(\Phi(\theta_1) - U_{\tilde{\theta}}(\theta_1, \tau_1)\right)}{T}$$

$$\overset{(a)}{\leq} \frac{4\Delta_\Phi}{\alpha_1 T} + 8L^2 R\left[2 + 2\kappa^2\right]6R(R - 1)^2\left[\left(\alpha_1^2 + \alpha_2^2\right)G^2 + \left(\alpha_1^2\zeta_\theta^2 + \alpha_2^2\zeta_\tau^2\right)\right] + \frac{16\kappa^2\Delta_\Phi}{\alpha_2 T}$$

$$\overset{(b)}{\leq} \frac{4\Delta_\Phi}{\alpha_1 T} + 192L^2\kappa^2 R(R - 1)^2\left[\left(\alpha_1^2 + \alpha_2^2\right)G^2 + \alpha_1^2\zeta_\theta^2 + \alpha_2^2\zeta_\tau^2\right] + \frac{16\kappa^2\Delta_\Phi}{\alpha_2 T}$$

$$= \mathcal{O}\left(\frac{\Delta_\Phi}{\alpha_1 T} + \kappa^2\frac{\Delta_\Phi}{\alpha_2 T} + L^2\kappa^2 R(R - 1)^2\left[\left(\alpha_1^2 + \alpha_2^2\right)G^2 + \alpha_1^2\zeta_\theta^2 + \alpha_2^2\zeta_\tau^2\right]\right).$$

$$= \underbrace{\mathcal{O}\left(\kappa^2\frac{\Delta_\Phi}{\alpha_2 T}\right)}_{\text{Error with full synchronization}} + \underbrace{\mathcal{O}\left(L^2\kappa^2 R(R - 1)^2\left[\alpha_2^2\left(G^2 + \zeta_\tau^2\right) + \alpha_1^2\zeta_\theta^2\right]\right)}_{\text{Error due to local updates}}. \quad \text{(since } \kappa \geq 1\text{)}$$

where, we denote $\Delta_\Phi := \Phi(\theta_1) - \min_\theta \Phi(\theta)$. $(a)$ follows from $\frac{\alpha_1}{\alpha_2} \leq \frac{1}{8\kappa^2}$; $(b)$ follows since $\kappa \geq 1$ and $L_\Phi \geq L$. Therefore, $\frac{8\kappa^2\alpha_1}{\alpha_2}\frac{\alpha_1\sigma^2}{N}(L + L_\Phi) \leq \frac{\alpha_1\sigma^2}{N}(L + L_\Phi) \leq \frac{2L_\Phi\alpha_1\sigma^2}{N}$, which results in Equation (34).

Using $\alpha_2 = \sqrt{\frac{N}{LT}}$ and $\alpha_1 = \frac{1}{8\kappa^2}\sqrt{\frac{N}{LT}} \le \frac{\alpha_2}{8\kappa^2}$, and since $\kappa \ge 1$, we get

$$\frac{1}{T}\sum_{t=1}^{T}\|\nabla\Phi(\theta_t)\|^2 \le \mathcal{O}\left(\frac{\kappa^2\Delta_\Phi}{\sqrt{NT}} + \kappa^2 R(R-1)^2\frac{N}{T}\left[G^2 + \frac{\zeta_\theta^2}{\kappa^4} + \zeta_\tau^2\right]\right).$$

$\square$

# B  PROOFS

## B.1  PROOF OF LEMMA 1

**Lemma 7** (Restatement of Lemma 1). *Let $\mathcal{F} = span\{f_j^i \mid i \in [N],\ j \in [m]\}$. An equivalent objective function for the federated moment estimation optimization problem (16) is given by:*

$$\|\psi_N(f;\theta)\|_{\tilde{\theta}}^2 = \sup_{\substack{f^i \in \mathcal{F} \\ \forall i \in [N]}} \frac{1}{N}\sum_{i=1}^{N}\left(\psi_{n_i}(f^i;\theta) - \frac{1}{4}\mathcal{C}_{\tilde{\theta}}(f^i;f^i)\right),\ \textit{where} \tag{37}$$

$$\psi_{n_i}(f^i;\theta) := \frac{1}{n_i}\sum_{k=1}^{n_i}f^i(Z_k^i)(Y_k^i - g^i(X_k^i;\theta)),\ \textit{and}\ \mathcal{C}_{\tilde{\theta}}(f^i,f^i) := \frac{1}{n_i}\sum_{k=1}^{n_i}(f^i(Z_k^i))^2(Y_k^i - g^i(X_k^i;\tilde{\theta}))^2.$$

*Proof.* Let $\psi = (\frac{1}{N}\sum_{i=1}^{N}\psi_{n_i}(f_1^i;\theta), \frac{1}{N}\sum_{i=1}^{N}\psi_{n_i}(f_2^i;\theta), \ldots, \frac{1}{N}\sum_{i=1}^{N}\psi_{n_i}(f_m^i;\theta))$.

We know that $\|v\|^2 = v^\top C_{\tilde{\theta}}^{-1} v$ and the associated dual norm is obtained as $\|v\|_*^2 = \sup_{\|v\|\le 1} v^\top v = v^\top C_{\tilde{\theta}} v$.

Using the definition of the dual norm,

$$\|\psi\| = \sup_{\|v\|_* \le 1} v^\top \psi$$
$$\|\psi\|^2 = \sup_{\|v\|_* \le \|\psi\|} v^\top \psi$$
$$\|\psi\|^2 = \sup_{v^\top C_{\tilde{\theta}} v \le \|\psi\|^2} v^\top \psi. \tag{38}$$

We now find the equivalent dual optimization problem for (38).

The Lagrangian of the constrained maximization problem (38) is given as

$$\mathcal{L}(v,\lambda) = v^\top \psi + \lambda(v^\top C_{\tilde{\theta}} v - \|\psi\|^2),\ \text{where}\ \lambda \le 0.$$

To maximize $\mathcal{L}(v,\lambda)$ w.r.t. $v$, put $\frac{\partial \mathcal{L}}{\partial v} = \psi + 2\lambda C_{\tilde{\theta}} v = 0$ to obtain $v = \frac{-1}{2\lambda}C_{\tilde{\theta}}^{-1}\psi$.

When $\|\psi\| > 0$, $v = 0$ satisfies Slater's condition as a strictly feasible interior point of the constraint $v^\top C_{\tilde{\theta}} v - \|\psi\|^2 \le 0$. Since $C_{\tilde{\theta}} \succeq 0$, the quadratic form $v^\top C_{\tilde{\theta}} v$ is convex in $v$, and the objective $v^\top \psi$ is linear. Hence, for this convex optimization problem, the Slater's condition applies. Thus, strong duality holds. Substituting $v = \frac{-1}{2\lambda}C_{\tilde{\theta}}^{-1}\psi$ in the Lagrangian gives

$$\mathcal{L}^*(\lambda) = \frac{-1}{2\lambda}\psi^\top C_{\tilde{\theta}}^{-1}\psi + \frac{1}{4\lambda}\psi^\top C_{\tilde{\theta}}^{-1}\psi - \lambda\|\psi\|^2$$
$$= -\frac{\|\psi\|^2}{4\lambda} - \lambda\|\psi\|^2.$$

Hence, the dual becomes $\|\psi\|^2 = inf_{\lambda < 0}\{\mathcal{L}^*(\lambda)\}$. Thus, the equivalent dual optimization problem for (38) is given as

$$\|\psi\|^2 = \inf_{\lambda < 0}\left\{-\frac{\|\psi\|^2}{4\lambda} - \lambda\|\psi\|^2\right\}. \tag{39}$$

Putting $\frac{\partial \mathcal{L}}{\partial \lambda} = \frac{\|\psi\|^2}{4\lambda^2} - \|\psi\|^2 = 0$ gives $\lambda = \frac{-1}{2}$. Thus, due to strong duality $\|\psi\|^2 = \sup_v \mathcal{L}(v, \frac{-1}{2}) = \sup_v v^\top \psi - \frac{1}{2}(v^\top C_{\tilde\theta} v - \|\psi\|^2)$.

Rewriting it $\frac{1}{2}\|\psi\|^2 = \sup_v v^\top \psi - \frac{1}{2}v^\top C_{\tilde\theta} v$ and substituting $u = 2v$

$$\|\psi\|^2 = \sup_u u^\top \psi - \frac{1}{4}u^\top C_{\tilde\theta} u.$$

Using the change of variables $u \to v$

$$\|\psi\|^2 = \sup_v v^\top \psi - \frac{1}{4}v^\top C_{\tilde\theta} v.$$

Now, we want to find a function form for the optimization problem mentioned above.

Consider a finite-dimensional functional spaces $\mathcal{F}^i = \mathrm{span}\{f_1^i, f_2^i, \ldots, f_m^i\}$ for each client $i$. Hence, for $f^i \in \mathcal{F}^i$

$$f^i = \sum_{j=1}^{m} v_j f_j^i.$$

Since all the clients share the same neural network architecture, we define a global functional space $\mathcal{F}$ as

$$\mathcal{F} = \mathrm{span}\{f_j^i \mid i \in [N],\ j \in [m]\}.$$

Therefore, $v$ corresponds to $f^i$ such that

$$f^i = \sum_{c=1}^{N} \sum_{j=1}^{m} v_j^i f_j^c, \text{ where } v_j^i = \begin{cases} v_j & \text{if } c = i \\ 0 & \text{if } c \neq i \end{cases}$$

Hence,

$$v^\top \psi = \frac{1}{N} \sum_{i=1}^{N} \sum_{j=1}^{m} v_j \psi_{n_i}(f_j^i; \theta)$$

$$= \frac{1}{N} \sum_{i=1}^{N} \frac{1}{n_i} \sum_{k=1}^{n_i} f^i(Z_k^i)(Y_k^i - g^i(X_k^i; \theta)).$$

Similarly,

$$v^\top C_{\tilde\theta} v = \sum_{p=1}^{m} \sum_{q=1}^{m} v_p v_q [C_{\tilde\theta}]pq$$

$$= \sum_{p=1}^{m} \sum_{q=1}^{m} v_p v_q \frac{1}{N} \sum_{i=1}^{N} \frac{1}{n_i} \sum_{k=1}^{n_i} f_p^i(Z_k^i) f_q^i(Z_k^i)(Y_k^i - g^i(X_k^i; \tilde\theta))$$

$$= \frac{1}{N} \sum_{i=1}^{N} \frac{1}{n_i} \sum_{k=1}^{n_i} \sum_{p=1}^{m} v_p f_p^i(Z_k^i) \sum_{q=1}^{m} v_q f_q^i(Z_k^i)(Y_k^i - g^i(X_k^i; \tilde\theta))^2$$

$$= \frac{1}{N} \sum_{i=1}^{N} \frac{1}{n_i} \sum_{k=1}^{n_i} (f^i(Z_k^i))^2 (Y_k^i - g^i(X_k^i; \tilde\theta))^2$$

$$= \frac{1}{N} \sum_{i=1}^{N} \mathcal{C}_{\tilde\theta}(f^i, f^i).$$

Thus, using the linear isomorphism between $\mathbb{R}^m$ and $\mathrm{span}\{f_1^i, f_2^i, \ldots, f_m^i\}$, using $v^\top \psi = \frac{1}{N} \sum_{i=1}^{N} \psi_{n_i}(f^i; \theta)$ and $v^\top C_{\tilde\theta} v = \frac{1}{N} \sum_{i=1}^{N} \mathcal{C}_{\tilde\theta}(f^i, f^i)$, we can write the objective in functional form as

$$\|\psi\|^2 = \sup_{\substack{f^i \in \mathcal{F} \\ \forall i \in [N]}} \frac{1}{N} \sum_{i=1}^{N} \left( \psi_{n_i}(f^i; \theta) - \frac{1}{4}\mathcal{C}_{\tilde\theta}(f^i, f^i) \right).$$

This gives us the desired result.

$\square$

## B.2 PROOF OF THEOREM 1

**Theorem 5** (Restatement of Theorem 1). *Under assumptions 1, 2, 3 and 4, a minimax solution* $(\hat{\theta}, \hat{\tau})$ *of federated optimization problem (17) that satisfies the equilibrium condition as in definition 1:* $U_{\tilde{\theta}}(\hat{\theta}, \tau) \leq U_{\tilde{\theta}}(\hat{\theta}, \hat{\tau}) \leq \max_{\tau\prime:\|\tau\prime-\hat{\tau}\|\leq h(\delta)} U_{\tilde{\theta}}(\theta, \tau\prime)$, *is an $\mathcal{E}$-approximate federated equilibrium solution as defined in 3, where the approximation error $\varepsilon^i$ for each client $i \in [N]$ lies in:* $\max\{\zeta_{\theta}^i, \zeta_{\tau}^i\} \leq \varepsilon^i \leq \min\{\alpha - \rho_{\tau}^i, \beta - B^i\}$ *for $\rho_{\tau}^i < \alpha$ and $B^i > \beta$, such that* $\alpha := \left|\lambda_{\max}\left(\nabla_{\tau\tau}^2 U_{\tilde{\theta}}(\hat{\theta}, \hat{\tau})\right)\right|$, $\beta := \lambda_{\min}\left(\left[\nabla_{\theta\theta}^2 U_{\tilde{\theta}} - \nabla_{\theta\tau}^2 U_{\tilde{\theta}}\left(\nabla_{\tau\tau}^2 U_{\tilde{\theta}}\right)^{-1}\nabla_{\tau\theta}^2 U_{\tilde{\theta}}\right](\hat{\theta}, \hat{\tau})\right)$ *and* $B^i := \rho_{\theta}^i + L\rho_{\theta\tau}^i \frac{1}{|\lambda_{\max}(\nabla_{\tau\tau}^2 U_{\tilde{\theta}}^i)|} + L\rho_{\tau\theta}^i \frac{1}{|\lambda_{\max}(\nabla_{\tau\tau}^2 U_{\tilde{\theta}}^i)|} + L^2\rho_{\tau}^i \frac{1}{|\lambda_{\max}(\nabla_{\tau\tau}^2 U_{\tilde{\theta}}^i)\cdot\lambda_{\max}(\nabla_{\tau\tau}^2 U_{\tilde{\theta}})|}$.

*Proof.* The pure-strategy Stackelberg equilibrium for the federated objective is:

$$U_{\tilde{\theta}}(\hat{\theta}, \tau) \leq U_{\tilde{\theta}}(\hat{\theta}, \hat{\tau}) \leq \max_{\tau\prime:\|\tau\prime-\tau^*\|\leq h(\delta)} U_{\tilde{\theta}}(\theta, \tau\prime), \tag{40}$$

We want to show that the $\epsilon^i$- approximate equilibrium for each client's objective $U_{\tilde{\theta}}^i$ also hold individually.

The first-order necessary condition for (40) to hold is $\nabla_\theta U_{\tilde{\theta}}(\hat{\theta}, \hat{\tau}) = 0$ and $\nabla_\tau U_{\tilde{\theta}}(\hat{\theta}, \hat{\tau}) = 0$. Thus, $\left\|\nabla_\theta U_{\tilde{\theta}}(\hat{\theta}, \hat{\tau})\right\|^2 = 0$.

Consider

$$\left\|\nabla_\theta U_{\tilde{\theta}}(\hat{\theta}, \hat{\tau})\right\|^2 = \left\|\nabla_\theta U_{\tilde{\theta}}(\hat{\theta}, \hat{\tau}) - \nabla_\theta U_{\tilde{\theta}}^i(\hat{\theta}, \hat{\tau}) + \nabla_\theta U_{\tilde{\theta}}^i(\hat{\theta}, \hat{\tau})\right\|^2$$

$$= \left\|\nabla_\theta U_{\tilde{\theta}}(\hat{\theta}, \hat{\tau}) - \nabla_\theta U_{\tilde{\theta}}^i(\hat{\theta}, \hat{\tau})\right\|^2 + \left\|\nabla_\theta U_{\tilde{\theta}}^i(\hat{\theta}, \hat{\tau})\right\|^2$$

$$+ 2\left(\nabla_\theta U_{\tilde{\theta}}(\hat{\theta}, \hat{\tau}) - \nabla_\theta U_{\tilde{\theta}}^i(\hat{\theta}, \hat{\tau})\right)^\top\left(\nabla_\theta U_{\tilde{\theta}}^i(\hat{\theta}, \hat{\tau})\right)$$

Rearranging

$$2\left(\nabla_\theta U_{\tilde{\theta}}^i(\hat{\theta}, \hat{\tau}) - \nabla_\theta U_{\tilde{\theta}}(\hat{\theta}, \hat{\tau})\right)^\top\left(\nabla_\theta U_{\tilde{\theta}}^i(\hat{\theta}, \hat{\tau})\right) - \left\|\nabla_\theta U_{\tilde{\theta}}^i(\hat{\theta}, \hat{\tau})\right\|^2 = \left\|\nabla_\theta U_{\tilde{\theta}}(\hat{\theta}, \hat{\tau}) - \nabla_\theta U_{\tilde{\theta}}^i(\hat{\theta}, \hat{\tau})\right\|^2$$

$$2\left\|\nabla_\theta U_{\tilde{\theta}}^i(\hat{\theta}, \hat{\tau})\right\|^2 - \left\|\nabla_\theta U_{\tilde{\theta}}^i(\hat{\theta}, \hat{\tau})\right\|^2 = \left\|\nabla_\theta U_{\tilde{\theta}}(\hat{\theta}, \hat{\tau}) - \nabla_\theta U_{\tilde{\theta}}^i(\hat{\theta}, \hat{\tau})\right\|^2$$

Using gradient heterogeneity assumption (3) on R.H.S.

$$\left\|\nabla_\theta U_{\tilde{\theta}}(\hat{\theta}, \hat{\tau}) - \nabla_\theta U_{\tilde{\theta}}^i(\hat{\theta}, \hat{\tau})\right\|^2 \leq (\zeta_{\theta}^i)^2$$

Thus, we obtain $\left\|\nabla_\theta U_{\tilde{\theta}}^i(\hat{\theta}, \hat{\tau})\right\| \leq \zeta_{\theta}^i$. Similarly, $\left\|\nabla_\tau U_{\tilde{\theta}}^i(\hat{\theta}, \hat{\tau})\right\| \leq \zeta_{\tau}^i$.

In the special case, when $\zeta_{\theta}^i = 0$ and $\zeta_{\tau}^i = 0$, thus we will have $\left\|\nabla_\theta U_{\tilde{\theta}}^i(\hat{\theta}, \hat{\tau})\right\|^2 = \left\|\nabla_\tau U_{\tilde{\theta}}^i(\hat{\theta}, \hat{\tau})\right\|^2 = 0$ for all $i \in [N]$, which gives $\nabla_\theta U_{\tilde{\theta}}^i(\hat{\theta}, \hat{\tau}) = \nabla_\tau U_{\tilde{\theta}}^i(\hat{\theta}, \hat{\tau}) = 0$ for all clients $i$.

Next, we prove that each client satisfies the second-order necessary condition approximately. Since $(\hat{\theta}, \hat{\tau})$ satisfy the equilibrium condition (40), the second-order necessary condition holds for the global function $U_{\tilde{\theta}}$, i.e. $\nabla_{\tau\tau}^2 U_{\tilde{\theta}}(\hat{\theta}, \hat{\tau}) \preceq \mathbf{0}$. We now prove that $\nabla_{\tau\tau}^2 U_{\tilde{\theta}}^i(\hat{\theta}, \hat{\tau}) \preceq \mathbf{0}$.

Using assumption 1, the hess ian is symmetric. Thus, $\nabla_{\tau\tau}^2 U_{\tilde{\theta}}(\hat{\theta}, \hat{\tau}) \preceq \mathbf{0}$ implies $\lambda_{\max}(\nabla_{\tau\tau}^2 U_{\tilde{\theta}}(\hat{\theta}, \hat{\tau})) \leq 0$, where $\lambda_{\max}$ is the largest eigenvalue of the hessian. Suppose, $\lambda_{\max}(\nabla_{\tau\tau}^2 U_{\tilde{\theta}}(\hat{\theta}, \hat{\tau})) = -\alpha$, for some $\alpha \geq 0$.

We can write $\nabla_{\tau\tau}^2 U_{\tilde{\theta}}^i(\hat{\theta}, \hat{\tau}) = \nabla_{\tau\tau}^2 U_{\tilde{\theta}}^i(\hat{\theta}, \hat{\tau}) - \nabla_{\tau\tau}^2 U_{\tilde{\theta}}(\hat{\theta}, \hat{\tau}) + \nabla_{\tau\tau}^2 U_{\tilde{\theta}}(\hat{\theta}, \hat{\tau})$.

Using a corollary of Weyl's theorem (Horn & Johnson, 2012) for real symmetric matrices $A$ and $B$, $\lambda_{\max}(A + B) \leq \lambda_{\max}(A) + \lambda_{\max}(B)$. Hence,

$$\lambda_{\max}(\nabla_{\tau\tau}^2 U_{\tilde{\theta}}^i(\hat{\theta}, \hat{\tau})) \leq \lambda_{\max}(\nabla_{\tau\tau}^2 U_{\tilde{\theta}}^i(\hat{\theta}, \hat{\tau}) - \nabla_{\tau\tau}^2 U_{\tilde{\theta}}(\hat{\theta}, \hat{\tau})) + \lambda_{\max}(\nabla_{\tau\tau}^2 U_{\tilde{\theta}}(\hat{\theta}, \hat{\tau})).$$

Thus, $\lambda_{\max}(\nabla^2_{\tau\tau} U^i_{\hat\theta}(\hat\theta, \hat\tau)) \leq \lambda_{\max}(\nabla^2_{\tau\tau} U^i_{\hat\theta}(\hat\theta, \hat\tau) - \nabla^2_{\tau\tau} U_{\tilde\theta}(\hat\theta, \hat\tau)) - \alpha$.

Since the spectral norm of a real symmetric matrix A is given as $\|A\|_\sigma = \max\{|\lambda_{\max}(A)|, |\lambda_{\min}(A)|\}$.

Under hessian heterogeneity assumption 4

$$\|\nabla^2_{\tau\tau} U^i_{\hat\theta}(\hat\theta, \hat\tau) - \nabla^2_{\tau\tau} U_{\tilde\theta}(\hat\theta, \hat\tau)\|_\sigma = \max\big\{\big|\lambda_{\max}(\nabla^2_{\tau\tau} U^i_{\hat\theta}(\theta, \tau) - \nabla^2_{\tau\tau} U_{\tilde\theta}(\theta, \tau))\big|,$$
$$\big|\lambda_{\min}(\nabla^2_{\tau\tau} U^i_{\hat\theta}(\theta, \tau) - \nabla^2_{\tau\tau} U_{\tilde\theta}(\theta, \tau))\big|\big\}$$
$$\leq \rho^i_\tau.$$

Thus, we have

$$\lambda_{max}(\nabla^2_{\tau\tau} U^i_{\hat\theta}(\hat\theta, \hat\tau) - \nabla^2_{\tau\tau} U_{\tilde\theta}(\hat\theta, \hat\tau)) \leq \max\big\{\big|\lambda_{\max}(\nabla^2_{\tau\tau} U^i_{\hat\theta}(\hat\theta, \hat\tau) - \nabla^2_{\tau\tau} U_{\tilde\theta}(\hat\theta, \hat\tau))\big|,$$
$$\big|\lambda_{\min}(\nabla^2_{\tau\tau} U^i_{\hat\theta}(\hat\theta, \hat\tau) - \nabla^2_{\tau\tau} U_{\tilde\theta}(\hat\theta, \hat\tau))\big|\big\}$$
$$\leq \rho^i_\tau.$$

Thus, $\lambda_{\max}(\nabla^2_{\tau\tau} U^i_{\hat\theta}(\hat\theta, \hat\tau)) \leq \lambda_{\max}(\nabla^2_{\tau\tau} U^i_{\hat\theta}(\hat\theta, \hat\tau) - \nabla^2_{\tau\tau} U_{\tilde\theta}(\hat\theta, \hat\tau)) - \alpha \leq \rho^i_\tau - \alpha$, where $\rho^i_\tau \geq 0$. Hence,

$$\nabla^2_{\tau\tau} U^i_{\hat\theta}(\hat\theta, \hat\tau) \preceq (\rho^i_\tau - \alpha)\mathbf{I}.$$

When $\rho^i_\tau \leq \alpha$, then $\nabla^2_{\tau\tau} U^i_{\hat\theta}(\hat\theta, \hat\tau) \preceq 0$.

Now, since $(\hat\theta, \hat\tau)$ satisfy the equilibrium condition (40), thus $\nabla^2_{\tau\tau} U_{\tilde\theta}(\hat\theta, \hat\tau) \prec 0$ and the Schur complement of $\nabla^2_{\tau\tau} U_{\tilde\theta}(\hat\theta, \hat\tau)$ is positive semi-definite. Now when $\rho^i_\tau < \alpha$, it follows from above that $\nabla^2_{\tau\tau} U^i_{\hat\theta}(\hat\theta, \hat\tau) \prec 0$, hence $\left(\nabla^2_{\tau\tau} U^i_{\hat\theta}(\hat\theta, \hat\tau)\right)^{-1}$ exists. Now, we need to show that Schur complement of $\nabla^2_{\tau\tau} U^i_{\hat\theta}(\hat\theta, \hat\tau)$ is positive semi-definite.

Since, $S(\hat\theta, \hat\tau) := \left[\nabla^2_{\theta\theta} U_{\tilde\theta} - \nabla^2_{\theta\tau} U_{\tilde\theta} \left(\nabla^2_{\tau\tau} U_{\tilde\theta}\right)^{-1} \nabla^2_{\tau\theta} U_{\tilde\theta}\right](\hat\theta, \hat\tau) \succ 0$.

Define $S^i := \left[\nabla^2_{\theta\theta} U^i_{\hat\theta} - \nabla^2_{\theta\tau} U^i_{\hat\theta} \left(\nabla^2_{\tau\tau} U^i_{\hat\theta}\right)^{-1} \nabla^2_{\tau\theta} U^i_{\hat\theta}\right]$. We aim to prove $\lambda_{\min}(S^i) \geq 0$ to show $S^i$ is positive semidefinite (PSD).

Analogous to the above part, using corollary to Weyl's theorem, we have

$$\lambda_{\min}(S^i - S) + \lambda_{\min}(S) \leq \lambda_{\min}(S^i).$$

Let $\lambda_{\min}(S) = \beta$, where $\beta \geq 0$. Moreover, $\|S^i - S\|_\sigma = \max\big\{\big|\lambda_{\max}(S^i - S)\big|, \big|\lambda_{\min}(S^i - S)\big|\big\}$, thus $\lambda_{\min}(S^i - S) \geq -\|S^i - S\|_\sigma$.

Thus, we have

$$-\|(S^i - S)\|_\sigma + \beta \leq \lambda_{\min}(S^i).$$

We can write $S^i - S$ as

$$S^i - S = (\nabla^2_{\theta\theta} U^i_{\hat\theta} - \nabla^2_{\theta\theta} U_{\tilde\theta}) - \Big[(\nabla^2_{\theta\tau} U^i_{\hat\theta} - \nabla^2_{\theta\tau} U_{\tilde\theta})(\nabla^2_{\tau\tau} U^i_{\hat\theta})^{-1} \nabla^2_{\tau\theta} U^i_{\hat\theta}$$
$$+ \nabla^2_{\theta\tau} U_{\tilde\theta}(\nabla^2_{\tau\tau} U^i_{\hat\theta})^{-1}(\nabla^2_{\tau\theta} U^i_{\hat\theta} - \nabla^2_{\tau\theta} U_{\tilde\theta}) + \nabla^2_{\theta\tau} U_{\tilde\theta}\Big((\nabla^2_{\tau\tau} U^i_{\hat\theta})^{-1} - (\nabla^2_{\tau\tau} U_{\tilde\theta})^{-1}\Big)\nabla^2_{\tau\theta} U_{\tilde\theta}\Big].$$

Hence,

$$\|S^i - S\|_\sigma \leq \|\nabla^2_{\theta\theta} U^i_{\hat\theta} - \nabla^2_{\theta\theta} U_{\tilde\theta}\|_\sigma + \underbrace{\|(\nabla^2_{\theta\tau} U^i_{\hat\theta} - \nabla^2_{\theta\tau} U_{\tilde\theta})(\nabla^2_{\tau\tau} U^i_{\hat\theta})^{-1}\nabla^2_{\tau\theta} U^i_{\hat\theta}\|_\sigma}_{T_1}$$

$$+ \underbrace{\|\nabla^2_{\theta\tau} U_{\tilde\theta}(\nabla^2_{\tau\tau} U^i_{\hat\theta})^{-1}(\nabla^2_{\tau\theta} U^i_{\hat\theta} - \nabla^2_{\tau\theta} U_{\tilde\theta})\|_\sigma}_{T_2}$$

$$+ \underbrace{\|\nabla^2_{\theta\tau} U_{\tilde\theta}\Big((\nabla^2_{\tau\tau} U^i_{\hat\theta})^{-1} - (\nabla^2_{\tau\tau} U_{\tilde\theta})^{-1}\Big)\nabla^2_{\tau\theta} U_{\tilde\theta}\|_\sigma}_{T_3}.$$

Note that the eigenvalue of $(\nabla^2_{\tau\tau}U^i_{\hat\theta})^{-1}$ is $\lambda\left((\nabla^2_{\tau\tau}U^i_{\hat\theta})^{-1}\right) = \frac{1}{\lambda(\nabla^2_{\tau\tau}U^i_{\hat\theta})}$, hence $\|(\nabla^2_{\tau\tau}U^i_{\hat\theta})^{-1}\|_\sigma = \frac{1}{|\lambda_{\max}(\nabla^2_{\tau\tau}U^i_{\hat\theta})|}$ as $\nabla^2_{\tau\tau}U^i_{\hat\theta}$ is negative definite. By Assumption 2, each client's function $U^i$ is $L$-Lipschitz thus $\|\nabla^2 U^i_{\hat\theta}\|_\sigma \le L$. Since the Hessian $\nabla^2 U^i_{\hat\theta}$ is a block matrix of the form:

$$\nabla^2 U^i_{\hat\theta} = \begin{bmatrix} \nabla^2_{\theta\theta}U^i_{\hat\theta} & \nabla^2_{\theta\tau}U^i_{\hat\theta} \\ \nabla^2_{\tau\theta}U^i_{\hat\theta} & \nabla^2_{\tau\tau}U^i_{\hat\theta} \end{bmatrix},$$

The norm of Hessian is at least the norm of one of its components

$$\|\nabla^2_{\theta\theta}U^i_{\hat\theta}\|_\sigma \le L, \quad \|\nabla^2_{\theta\tau}U^i_{\hat\theta}\|_\sigma \le L, \quad \|\nabla^2_{\tau\theta}U^i_{\hat\theta}\|_\sigma \le L, \quad \|\nabla^2_{\tau\tau}U^i_{\hat\theta}\|_\sigma \le L.$$

Thus, each Hessian block is individually bounded by $L$. Additionally, $U$ is $L$-Lipschitz too. Using Assumption 4, bounding $T_1$

$$\begin{aligned} T_1 &= \|(\nabla^2_{\theta\tau}U^i_{\hat\theta} - \nabla^2_{\theta\tau}U_{\tilde\theta})(\nabla^2_{\tau\tau}U^i_{\hat\theta})^{-1}\nabla^2_{\tau\theta}U^i_{\hat\theta}\|_\sigma \\ &\le \|(\nabla^2_{\theta\tau}U^i_{\hat\theta} - \nabla^2_{\theta\tau}U_{\tilde\theta})\|_\sigma \cdot \|(\nabla^2_{\tau\tau}U^i_{\hat\theta})^{-1}\|_\sigma \cdot \|\nabla^2_{\tau\theta}U^i_{\hat\theta}\|_\sigma \\ &\le L\rho^i_{\theta\tau}\frac{1}{|\lambda_{\max}(\nabla^2_{\tau\tau}U^i_{\hat\theta})|} \end{aligned}$$

Similarly, bounding $T_2$

$$\begin{aligned} T_2 &= \|\nabla^2_{\theta\tau}U_{\tilde\theta}(\nabla^2_{\tau\tau}U^i_{\hat\theta})^{-1}(\nabla^2_{\tau\theta}U^i_{\hat\theta} - \nabla^2_{\tau\theta}U_{\tilde\theta})\|_\sigma \\ &\le \|\nabla^2_{\theta\tau}U_{\tilde\theta}\|_\sigma \cdot \|(\nabla^2_{\tau\tau}U^i_{\hat\theta})^{-1}\|_\sigma \cdot \|(\nabla^2_{\tau\theta}U^i_{\hat\theta} - \nabla^2_{\tau\theta}U_{\tilde\theta})\|_\sigma \\ &\le L\rho^i_{\tau\theta}\frac{1}{|\lambda_{\max}(\nabla^2_{\tau\tau}U^i_{\hat\theta})|} \end{aligned}$$

Lastly we bound $T_3$, it is easy to verify that $\boldsymbol{A}^{-1} - \boldsymbol{B}^{-1} = \boldsymbol{A}^{-1}(\boldsymbol{B} - \boldsymbol{A})\boldsymbol{B}^{-1}$

$$\begin{aligned} T_3 &= \|\nabla^2_{\theta\tau}U_{\tilde\theta}\left((\nabla^2_{\tau\tau}U^i_{\hat\theta})^{-1} - (\nabla^2_{\tau\tau}U_{\tilde\theta})^{-1}\right)\nabla^2_{\tau\theta}U_{\tilde\theta}\|_\sigma \\ &\le \|\nabla^2_{\theta\tau}U_{\tilde\theta}\|_\sigma \cdot \|(\nabla^2_{\tau\tau}U^i_{\hat\theta})^{-1} - (\nabla^2_{\tau\tau}U_{\tilde\theta})^{-1}\|_\sigma \cdot \|\nabla^2_{\tau\theta}U_{\tilde\theta}\|_\sigma \\ &= \|\nabla^2_{\theta\tau}U_{\tilde\theta}\|_\sigma \cdot \|(\nabla^2_{\tau\tau}U^i_{\hat\theta})^{-1}(\nabla^2_{\tau\tau}U_{\tilde\theta} - \nabla^2_{\tau\tau}U^i_{\hat\theta})(\nabla^2_{\tau\tau}U_{\tilde\theta})^{-1}\|_\sigma \cdot \|\nabla^2_{\tau\theta}U_{\tilde\theta}\|_\sigma \\ &\le \|\nabla^2_{\theta\tau}U_{\tilde\theta}\|_\sigma \cdot \|(\nabla^2_{\tau\tau}U^i_{\hat\theta})^{-1}\|_\sigma \cdot \|\nabla^2_{\tau\tau}U_{\tilde\theta} - \nabla^2_{\tau\tau}U^i_{\hat\theta}\|_\sigma \cdot \|(\nabla^2_{\tau\tau}U_{\tilde\theta})^{-1}\|_\sigma \cdot \|\nabla^2_{\tau\theta}U_{\tilde\theta}\|_\sigma \\ &\le L^2\rho^i_\tau\frac{1}{|\lambda_{\max}(\nabla^2_{\tau\tau}U^i_{\hat\theta}) \cdot \lambda_{\max}(\nabla^2_{\tau\tau}U_{\tilde\theta})|} \end{aligned}$$

Using bounds for $T_1$, $T_2$ and $T_3$, we can obtain a bound on $\|S^i - S\|_\sigma \le B^i$, where $B^i = \rho^i_\theta + L\rho^i_{\theta\tau}\frac{1}{|\lambda_{\max}(\nabla^2_{\tau\tau}U^i_{\hat\theta})|} + L\rho^i_{\tau\theta}\frac{1}{|\lambda_{\max}(\nabla^2_{\tau\tau}U^i_{\hat\theta})|} + L^2\rho^i_\tau\frac{1}{|\lambda_{\max}(\nabla^2_{\tau\tau}U^i_{\hat\theta}) \cdot \lambda_{\max}(\nabla^2_{\tau\tau}U_{\tilde\theta})|}$. Consider $\rho^i = \max\{\rho^i_\theta, \rho^i_{\tau\theta}, \rho^i_{\theta\tau}, \rho^i_\tau\}$. Hence, $B^i \le \rho^i\left(1 + \frac{L}{\lambda_{\max}(\nabla^2_{\tau\tau}U^i_{\hat\theta})}\left(2 + \frac{1}{\lambda_{\max}(\nabla^2_{\tau\tau}U_{\tilde\theta})}\right)\right)$. Hence, we obtain

$$\lambda_{\min}(S^i) \ge -B^i + \beta,$$

where $\lambda_{\max}(S) = \beta$ such that $\beta \ge 0$. Hence, we obtain $\left[\nabla^2_{\theta\theta}U^i_{\hat\theta} - \nabla^2_{\theta\tau}U^i_{\hat\theta}\left(\nabla^2_{\tau\tau}U^i_{\hat\theta}\right)^{-1}\nabla^2_{\tau\theta}U^i_{\hat\theta}\right](\hat\theta, \hat\tau) \succeq (\beta - B^i)I$. When $\beta \ge B^i$, then $S^i$ is positive semi-definite. When $B^i = 0$, hence $\left[\nabla^2_{\theta\theta}U^i_{\hat\theta} - \nabla^2_{\theta\tau}U^i_{\hat\theta}\left(\nabla^2_{\tau\tau}U^i_{\hat\theta}\right)^{-1}\nabla^2_{\tau\theta}U^i_{\hat\theta}\right](\hat\theta, \hat\tau) \succeq \beta I$, thus it will be positive semidefinite. When $\rho^i_\tau < \alpha$ and $\beta > B^i$, then the suuficient condition for $\varepsilon^i$-approximate equilibrium is satisfied. And we obtain the result.

Thus, for each client $i$, any approximation error $\varepsilon^i$ that satisfies:

$$\max\{\zeta^i_\theta, \zeta^i_\tau\} \le \varepsilon^i \le \min\{\alpha - \rho^i_\tau, \beta - B^i\}.$$

for $\rho^i_\tau < \alpha$ and $B^i > \beta$, then $(\hat\theta, \hat\tau)$ is an $\varepsilon^i$-approximate local equilibrium point for client $i$. $\square$

## B.3 CONSISTENCY

### B.3.1 ASSUMPTIONS

We first state the assumptions that are necessary to establish the consistency of the estimated parameter.

**Assumption 6** (Identification). $\theta_0$ is the unique $\theta \in \Theta$ such that $\psi(f^i; \theta) = 0$ for all $f^i \in \mathcal{F}$, where $i \in [n]$.

**Assumption 7** (Absolutely Star Shaped). For every $f^i \in \mathcal{F}^i$ and $|c| \leq 1$, we have $cf^i \in \mathcal{F}^i$.

**Assumption 8** (Continuity). For any $x$, $g^i(x; \theta)$, $f^i(x; \tau)$ are continuous in $\theta$ and $\tau$, respectively for all $i \in [N]$.

**Assumption 9** (Boundedness). $Y^i$, $\sup_{\theta \in \Theta} |g^i(X; \theta)|$, $\sup_{\tau \in \mathcal{T}} |f^i(Z; \tau)|$ are bounded random variables for all $i \in [N]$.

**Assumption 10** (Bounded Complexity). $\mathcal{F}^i$ and $\mathcal{G}^i$ have bounded Rademacher complexities:

$$\frac{1}{2^{n_i}} \sum_{\xi_i \in \{-1,+1\}^{n_i}} \mathbb{E} \sup_{\tau \in \mathcal{T}} \frac{1}{n_i} \sum_{k=1}^{n_i} \xi_i f^i(Z_k; \tau) \to 0, \quad \frac{1}{2^{n_i}} \sum_{\xi_i \in \{-1,+1\}^{n_i}} \mathbb{E} \sup_{\theta \in \Theta} \frac{1}{n_i} \sum_{k=1}^{n_i} \xi_i g^i(X_k; \theta) \to 0.$$

### B.3.2 PROOF OF THEOREM 2

**Theorem 6** (Restatement of Theorem 2). *Let $\tilde{\theta}_n$ be a data-dependent choice for the federated objective that has a limit in probability. For each client $i \in [N]$, define $m^i(\theta, \tau, \tilde{\theta}) := f^i(Z^i; \tau)(Y^i - g(X^i; \theta)) - \frac{1}{4} f^i(Z^i; \tau)^2(Y^i - g(X^i; \tilde{\theta}))^2$, $M^i(\theta) = \sup_{\tau \in \mathcal{T}} \mathbb{E}[m^i(\theta, \tau, \tilde{\theta})]$ and $\eta^i(\epsilon) := \inf_{d(\theta, \theta_0) \geq \epsilon} M^i(\theta) - M^i(\theta_0)$ for every $\epsilon > 0$. Let $(\hat{\theta}_n, \hat{\tau}_n)$ be a solution that satisfies the approximate equilibrium for each of the client $i \in [N]$ as*

$$\sup_{\tau \in \mathcal{T}} U_{\hat{\theta}}^i(\hat{\theta}_n, \tau) - \varepsilon^i - o_p(1) \leq U_{\hat{\theta}}^i(\hat{\theta}_n, \hat{\tau}_n) \leq \inf_{\theta \in \Theta} \max_{\tau\prime: \|\tau\prime - \hat{\tau}_n\| \leq h(\delta)} U_{\hat{\theta}}^i(\theta, \tau\prime) + \varepsilon^i + o_p(1),$$

*for some $\delta_0$, such that for any $\delta \in (0, \delta_0]$, and any $\theta, \tau$ such that $\|\theta - \hat{\theta}\| \leq \delta$ and $\|\tau - \hat{\tau}\| \leq \delta$ and a function $h(\delta) \to 0$ as $\delta \to 0$. Then, under similar assumptions as in Assumptions 1 to 5 of (Bennett et al., 2019), the global solution $\hat{\theta}_n$ is a consistent estimator to the true parameter $\theta_0$, i.e. $\hat{\theta}_n \xrightarrow{p} \theta_0$ when the approximate error $\varepsilon^i < \frac{\eta^i(\epsilon)}{2}$ for every $\epsilon > 0$ for each client $i \in [N]$.*

*Proof.* The proof follows from the result of Bennett et al. (2019) that established the consistency of the DEEPGMM estimator.

First, we define the following terms for the ease of analysis:

$$m^i(\theta, \tau, \tilde{\theta}) = f^i(Z^i; \tau)(Y^i - g(X^i; \theta)) - \frac{1}{4} f^i(Z^i; \tau)^2(Y^i - g(X^i; \tilde{\theta}))^2$$

$$M^i(\theta) = \sup_{\tau \in \mathcal{T}} \mathbb{E}[m^i(\theta, \tau, \tilde{\theta})]$$

$$M_{n_i}(\theta) = \sup_{\tau \in \mathcal{T}} \mathbb{E}_{n_i}[m^i(\theta, \tau, \tilde{\theta}_n)]$$

Note that $\tilde{\theta}_n$ is a data-dependent sequence for the global model. Practically, the previous global iterate is used as $\tilde{\theta}$. Thus, we can define for the federated setting $\tilde{\theta}_n = \frac{1}{N} \sum_{i=1}^{N} \tilde{\theta}_{n_i}$. Let's assume $\tilde{\theta}_n \xrightarrow{p} \tilde{\theta}$.

**Claim 1:** $\sup_\theta |M_{n_i}(\theta) - M^i(\theta)| \xrightarrow{p} 0$.

$$
\sup_\theta |M_{n_i}(\theta) - M^i(\theta)| = \sup_\theta \left| \sup_{\tau \in \mathcal{T}} \mathbb{E}_{n_i}[m^i(\theta, \tau, \tilde{\theta}_n)] - \sup_{\tau \in \mathcal{T}} \mathbb{E}[m^i(\theta, \tau, \tilde{\theta})] \right|
$$

$$
\leq \sup_{\theta, \tau} \left| \mathbb{E}_{n_i}[m^i(\theta, \tau, \tilde{\theta}_n)] - \mathbb{E}[m^i(\theta, \tau, \tilde{\theta})] \right|
$$

$$
\leq \sup_{\theta, \tau} \left| \mathbb{E}_{n_i}[m^i(\theta, \tau, \tilde{\theta}_n)] - \mathbb{E}[m^i(\theta, \tau, \tilde{\theta}_n)] \right| + \sup_{\theta, \tau} \left| \mathbb{E}[m^i(\theta, \tau, \tilde{\theta}_n)] - \mathbb{E}[m^i(\theta, \tau, \tilde{\theta})] \right|
$$

$$
\leq \sup_{\theta_1, \theta_2, \tau} \left| \mathbb{E}_{n_i}[m^i(\theta_1, \tau, \theta_2)] - \mathbb{E}[m^i(\theta_1, \tau, \theta_2)] \right| + \sup_{\theta, \tau} \left| \mathbb{E}[m^i(\theta, \tau, \tilde{\theta}_n)] - \mathbb{E}[m^i(\theta, \tau, \tilde{\theta})] \right|
$$

We will now handle the two terms in the above equation separately.

We will take the first term and call it $B_1$. For $m^i(\theta, \tau, \tilde{\theta}_n)$, we constitute its empirical counterpart $m_k^i(\theta, \tau, \tilde{\theta}_n) = f^i(Z_k^i; \tau)(Y_k^i - g^i(X_k^i; \theta)) - \frac{1}{4} f^i(Z_k^i; \tau)^2 (Y_k^i - g^i(X_k^i; \tilde{\theta}))^2$ and using $m_k^{i\,\prime}(\theta, \tau, \tilde{\theta}_n')$ with ghost variables $\tilde{\theta}_n'$ for symmetrization and $\epsilon_k$ as $k$ i.i.d. Rademacher random variables , we obtain

$$
\mathbb{E}[B_1] = \mathbb{E}\left[ \sup_{\theta_1, \theta_2, \tau} \left| \frac{1}{n_i} \sum_{k=1}^{n_i} m_k^i(\theta_1, \tau, \theta_2) - \mathbb{E}\left[ m_k^{i\,\prime}(\theta_1, \tau, \theta_2') \right] \right| \right]
$$

$$
\leq \mathbb{E}\left[ \sup_{\theta_1, \theta_2, \tau} \left| \frac{1}{n_i} \sum_{k=1}^{n_i} \left( m_k^i(\theta_1, \tau, \theta_2) - m_k^{i\,\prime}(\theta_1, \tau, \theta_2') \right) \right| \right]
$$

$$
\leq \mathbb{E}\left[ \sup_{\theta_1, \theta_2, \tau} \left| \frac{1}{n_i} \sum_{k=1}^{n_i} \epsilon_k \left( m_k^i(\theta_1, \tau, \theta_2) - m_k^{i\,\prime}(\theta_1, \tau, \theta_2') \right) \right| \right]
$$

$$
\leq 2\mathbb{E}\left[ \sup_{\theta_1, \theta_2, \tau} \left| \frac{1}{n_i} \sum_{k=1}^{n_i} \epsilon_k m_k^i(\theta_1, \tau, \theta_2) \right| \right]
$$

$$
\leq 2\mathbb{E}\left[ \sup_{\theta, \tau} \left| \frac{1}{n_i} \sum_{k=1}^{n_i} \epsilon_k f^i(Z_k^i; \tau)(Y_k^i - g^i(X_k^i; \theta)) \right| \right]
$$

$$
+ \frac{1}{2}\mathbb{E}\left[ \sup_{\theta, \tau} \left| \frac{1}{n_i} \sum_{k=1}^{n_i} \epsilon_k f^i(Z_k^i; \tau)^2 (Y_k^i - g^i(X_k^i; \tilde{\theta}))^2 \right| \right]
$$

$$
\leq 2\mathbb{E}\left[ \sup_{\theta, \tau} \left| \frac{1}{n_i} \sum_{k=1}^{n_i} \epsilon_k \left( \frac{1}{2} f^i(Z_k^i; \tau)^2 + \frac{1}{2}(Y_k^i - g^i(X_k^i; \theta))^2 \right) \right| \right]
$$

$$
+ \frac{1}{2}\mathbb{E}\left[ \sup_{\theta, \tau} \left| \frac{1}{n_i} \sum_{k=1}^{n_i} \epsilon_k \left( \frac{1}{2} f^i(Z_k^i; \tau)^4 + \frac{1}{2}(Y_k^i - g^i(X_k^i; \tilde{\theta}))^4 \right) \right| \right]
$$

$$
\leq \mathbb{E}\left[ \sup_{\theta, \tau} \left| \frac{1}{n_i} \sum_{k=1}^{n_i} \epsilon_k f^i(Z_k^i; \tau)^2 \right| \right] + \mathbb{E}\left[ \sup_{\theta, \tau} \left| \frac{1}{n_i} \sum_{k=1}^{n_i} \epsilon_k (Y_k^i - g^i(X_k^i; \theta))^2 \right| \right]
$$

$$
+ \frac{1}{4}\mathbb{E}\left[ \sup_{\theta, \tau} \left| \frac{1}{n_i} \sum_{k=1}^{n_i} \epsilon_k f^i(Z_k^i; \tau)^4 \right| \right] + \frac{1}{4}\mathbb{E}\left[ \sup_{\theta, \tau} \left| \frac{1}{n_i} \sum_{k=1}^{n_i} \epsilon_k (Y_k^i - g^i(X_k^i; \tilde{\theta}))^4 \right| \right]
$$

Using boundedness assumption 9, we consider the mapping from $f^i(Z_k^i; \tau)$ and $g^i(X_k^i; \tilde{\theta})$ to the summation terms in the last inequality as Lipschitz functions, hence for any functional class $\mathcal{F}^i$ and $L$- Lipschitz function $\phi$, $\mathcal{R}_{n_i}(\phi \circ f^i) \leq L\mathcal{R}_{n_i}(\mathcal{F}^i)$, where $\mathcal{R}_{n_i}(\mathcal{F}^i)$ is the Rademacher complexity of class $\mathcal{F}^i$. Hence, $\mathbb{E}[B_1] \leq L(\mathcal{R}_{n_i}(\mathcal{G}^i) + \mathcal{R}_{n_i}(\mathcal{F}^i))$. Using assumption 10, $\mathbb{E}[B_1] \to 0$. Let $B_1'$ be a modified value of $B$, after changing the $j$-th value of $X^i, Z^i$ and $Y^i$ values, using assumption 9 on

boundedness, we obtain the bounded difference inequality:

$$\sup_{X_{1:n_i},Z_{1:n_i},Y_{1:n_i},X'_j,Z'_j,Y'_j} |B_1 - B'_1| \le \sup_{\theta_1,\theta_2,\tau,X_{1:n_i},Z_{1:n_i},Y_{1:n_i},X'_j,Z'_j,Y'_j} |\frac{1}{n_i}\left(m^i_j(\theta_1,\tau,\theta_2) - m^{i\prime}_j(\theta_1,\tau,\theta_2)\right)|$$

$$\le \frac{b}{n_i},$$

where $b$ is some constant. Using McDiarmid's Inequality, we have $P(|B_1 - \mathbb{E}[B_1]| \ge \epsilon_0) \le 2\exp\left(\frac{-2n_i\epsilon_0^2}{c^2}\right)$. And $\mathbb{E}[B_1] \to 0$, we have $B_1 \xrightarrow{p} 0$.

Now, we will handle $B_2$. For that

$$B_2 = \sup_{\theta,\tau}\left|\mathbb{E}\left[m^i(\theta,\tau,\tilde{\theta}_n)\right] - \mathbb{E}\left[m^i(\theta,\tau,\tilde{\theta})\right]\right|$$

$$= \sup_{\theta,\tau}\left|\mathbb{E}\left[f^i(Z^i;\tau)(Y^i - g(X^i;\theta)) - \frac{1}{4}f^i(Z^i;\tau)^2(Y^i - g(X^i;\tilde{\theta}_n))^2\right]\right.$$

$$\left. - \mathbb{E}\left[f^i(Z^i;\tau)(Y^i - g(X^i;\theta)) - \frac{1}{4}f^i(Z^i;\tau)^2(Y^i - g(X^i;\tilde{\theta}))^2\right]\right|$$

$$= \sup_{\theta,\tau}\frac{1}{4}\left|\mathbb{E}\left[f^i(Z^i;\tau)^2(Y^i - g(X^i;\tilde{\theta}_n))^2\right] - \mathbb{E}\left[f^i(Z^i;\tau)^2(Y^i - g(X^i;\tilde{\theta}))^2\right]\right|$$

$$= \sup_{\theta,\tau}\frac{1}{4}\left|\mathbb{E}\left[f^i(Z^i;\tau)^2(Y^i - g(X^i;\tilde{\theta}_n))^2\right] + \mathbb{E}\left[f^i(Z^i;\tau)^2(Y^i - g(X^i;\tilde{\theta}))^2\right]\right.$$

$$\left. - \mathbb{E}\left[f^i(Z^i;\tau)^2(Y^i - g(X^i;\tilde{\theta}))^2\right] - \mathbb{E}\left[f^i(Z^i;\tau)^2(Y^i - g(X^i;\tilde{\theta}))^2\right]\right|$$

$$\le \frac{1}{4}\sup_{\tau}\left|\mathbb{E}\left[f^i(Z^i;\tau)^2\omega_n\right]\right|$$

Here, $\omega_n = \left|(Y^i - g(X^i;\tilde{\theta}_n))^2 - (Y^i - g(X^i;\tilde{\theta}))^2\right|$. Due to our assumption, $\tilde{\theta}_n \xrightarrow{p} \tilde{\theta}$, thus $\omega_n \xrightarrow{p} 0$ due to Slutsky's and continuous mapping theorem. Since, $f^i(Z;\tau)$ is uniformly bounded, thus for some constant $b' > 0$, we have

$$B_2 \le \frac{b'}{4}\sup_{\tau}\frac{1}{N}\sum_{i=1}^{N}|\mathbb{E}[\omega_n]|$$

$$\le \frac{b'}{4}\sup_{\tau}\frac{1}{N}\sum_{i=1}^{N}\mathbb{E}[|\omega_n|]$$

Based on the boundedness assumption, we can verify that $\omega_n$ is bounded, hence using Lebesgue Dominated Convergence Theorem, we can conclude that $\mathbb{E}[|\omega_n|] \to 0$.

Thus, using the convergence of $B_1$ and $B_2$, we have $\sup_{\theta}|M_{n_i}(\theta) - M^i(\theta)| \xrightarrow{p} 0$ for each $i \in [N]$.

**Claim 2:** for every $\epsilon > 0$, we have $\inf_{d(\theta,\theta_0)\ge\epsilon} M^i(\theta) > M^i(\theta_0)$.

$M^i(\theta_0)$ is the unique minimizer of $M^i(\theta)$. By assumption (6) and (7), $\theta_0$ is the unique minimizer of $\sup_{\tau}\mathbb{E}[f^i(Z^i;\tau)(Y^i - g^i(X;\theta))]$ such that $\sup_{\tau}\mathbb{E}[f^i(Z^i;\tau)(Y^i - g^i(X;\theta))] = 0$. Thus, any other value of $\theta$ will have at least one $\tau$ such that this expectation is strictly positive. $M(\theta_0) = 0$ and $M(\theta_0) = \sup_{\tau} -\frac{1}{4}f^i(Z^i;\tau)^2(Y^i - g^i(X;\theta_0))^2$, the function whose supremum is being evaluated is non-positive but can be set to zero by assumption (7) by taking the zero function of $f^i$. Let for any other $\theta' \ne \theta_0$, let $f^{i\prime}$ be a function in $\mathcal{F}^i$ such that $\mathbb{E}[f^i(Z)(Y^i - g^i(X;\theta'))] > 0$. If we have $\mathbb{E}[f^{i\prime}(Z)^2(Y^i - g^i(X;\tilde{\theta}))^2] = 0$, then $M^i(\theta') > 0$. Else, consider $cf^{i\prime}$ for any $c \in (0,1)$. Using assumption (7), $cf^{i\prime} \in \mathcal{F}^i$, thus

$$M^i(\theta') = \sup_{f^i\in\mathcal{F}^i}\mathbb{E}\left[f^i(Z^i)(Y^i - g(X^i;\theta')) - \frac{1}{4}f^i(Z^i)^2(Y^i - g(X^i;\tilde{\theta}))^2\right]$$

$$\le c\mathbb{E}\left[f^{i\prime}(Z^i)(Y^i - g(X^i;\theta'))\right] - \frac{c^2}{4}\mathbb{E}\left[f^{i\prime}(Z^i)^2(Y^i - g(X^i;\tilde{\theta}))^2\right]$$

This is quadratic in $c$ and is positive when $c$ is sufficiently small, thus $M^i(\theta') > 0$.

We now prove claim 2 using contradiction. Let us assume claim 2 is false, i.e. for some $\epsilon > 0$, we have $\inf_{\theta \in B(\theta_0, \epsilon)} M^i(\theta) = M^i(\theta_0)$, where $B(\theta_0, \epsilon)^c = \{\theta \mid d(\theta, \theta_0) \geq \epsilon\}$., since $\theta_0$ is the unique minimizer of $M^i(\theta)$ by assumption (6). Thus, there must exist some sequence $(\theta_1, \theta_2, \dots)$ in $B(\theta_0, \epsilon)^c$ such that $M^i(\theta_n) \to M^i(\theta_0)$. By construction, $B(\theta_0, \epsilon)^c$ is closed and the corresponding limit parameters $\theta^* = \lim_{n \to \infty} \theta_n \in B(\theta_0, \epsilon)^c$ must satisfy $M^i(\theta^*) = M^i(\theta_0)$ using assumption (8). But $d(\theta^*, \theta_0) \geq \epsilon > 0$, thus $\theta^* \neq \theta_0$. This contradicts that $\theta_0$ is the unique minimizer of $M^i(\theta)$; hence, claim 2 is true.

**Claim 3:** For the third part, we know that $\hat{\theta}_n$ satisfies the $\varepsilon^i$- approximate equilibrium condition, given as:

$$\mathbb{E}_{n_i}[m^i(\hat{\theta}_n, \tau, \tilde{\theta}_n)] - \varepsilon^i \leq \mathbb{E}_{n_i}[m^i(\hat{\theta}_n, \hat{\tau}_n, \tilde{\theta}_n)] \leq \max_{\tau\prime : \|\tau\prime - \hat{\tau}_n\| \leq h(\delta)} \mathbb{E}_{n_i}[m^i(\theta, \tau\prime, \tilde{\theta}_n)] + \varepsilon^i,$$

for a function $h(\delta) \to 0$ as $\delta \to 0$ and some $\delta_0$, such that for any $\delta \in (0, \delta_0]$, and any $\theta, \tau$ such that $\|\theta - \hat{\theta}\| \leq \delta$ and $\|\tau - \hat{\tau}\| \leq \delta$. Assume that this is true with $o_p(1)$, hence

$$\sup_\tau \mathbb{E}_{n_i}[m^i(\hat{\theta}_n, \tau, \tilde{\theta}_n)] - \varepsilon^i - o_p(1) \leq \mathbb{E}_{n_i}[m^i(\hat{\theta}_n, \hat{\tau}_n, \tilde{\theta}_n)] \leq \inf_\theta \max_{\tau\prime : \|\tau\prime - \hat{\tau}_n\| \leq h(\delta)} \mathbb{E}_{n_i}[m^i(\theta, \tau\prime, \tilde{\theta}_n)] + \varepsilon^i + o_p(1),$$

.

Now, since $M_{n_i}(\hat{\theta}_n) = \sup_\tau \mathbb{E}_{n_i}[m^i(\hat{\theta}_n, \tau, \tilde{\theta}_n)]$. Hence,

$$inf_\theta \max_{\tau\prime : \|\tau\prime - \hat{\tau}_n\| \leq h(\delta)} \mathbb{E}_{n_i}[m^i(\theta, \tau\prime, \tilde{\theta}_n) \leq inf_\theta \sup_\tau \mathbb{E}_{n_i}[m^i(\theta, \tau\prime, \tilde{\theta}_n)] = inf_\theta M_{n_i}(\theta) \leq M_{n_i}(\theta_0)$$

Thus, we have

$$M_{n_i}(\hat{\theta}_n) - \varepsilon^i - o_p(1) \leq \mathbb{E}_{n_i}[m^i(\hat{\theta}_n, \hat{\tau}_n, \tilde{\theta}_n)] \leq M_{n_i}(\theta_0) + \varepsilon^i + o_p(1).$$

We have proven all three conditions until now. From the first and second condition, since $|M_{n_i}(\theta_0) - M^i(\theta_0)| \xrightarrow{p} 0$, hence $M_{n_i}(\hat{\theta}_n) \leq M^i(\theta_0) + 2\varepsilon^i + o_p(1)$. Hence, we obtain

$$M^i(\hat{\theta}_n) - M^i(\theta_0) \leq M^i(\hat{\theta}_n) - M_{n_i}(\hat{\theta}_n) + 2\varepsilon^i + o_p(1)$$
$$\leq \sup_\theta |M^i(\hat{\theta}) - M_{n_i}(\hat{\theta})| + 2\varepsilon^i + o_p(1)$$
$$\leq 2\varepsilon^i + o_p(1)$$

Hence, we obtain

$$M^i(\hat{\theta}_n) - M^i(\theta_0) - 2\varepsilon^i \leq M^i(\hat{\theta}_n) - M_{n_i}(\hat{\theta}_n) + o_p(1)$$
$$\leq \sup_\theta |M^i(\hat{\theta}) - M_{n_i}(\hat{\theta})| + o_p(1)$$
$$\leq o_p(1)$$

Since, let $\eta^i(\epsilon) := inf_{d(\theta, \theta_0) \geq \epsilon} M^i(\theta) - M^i(\theta_0)$. Hence, whenever $d(\hat{\theta}_n, \theta_0) \geq \epsilon$, we have $M^i(\hat{\theta}_n) - M^i(\theta_0) \geq \eta^i(\epsilon)$. Thus, $\mathbb{P}[d(\hat{\theta}_n, \theta_0) \geq \epsilon] \leq \mathbb{P}[M^i(\hat{\theta}_n) - M^i(\theta_0) \geq \eta^i(\epsilon)] = \mathbb{P}[M^i(\hat{\theta}_n) - M^i(\theta_0) - 2\varepsilon^i \geq \eta^i(\epsilon) - 2\varepsilon^i]$. For every $\epsilon > 0$, we have $\eta^i(\epsilon) > 0$ from claim 2, and $M^i(\hat{\theta}_n) - M^i(\theta_0) - 2\varepsilon^i = o_p(1)$. Thus, $\eta^i(\epsilon) - 2\varepsilon^i > 0$ when $\varepsilon^i < \frac{\eta^i(\epsilon)}{2}$. We have that for every $\epsilon > 0$ and $\varepsilon^i < \frac{\eta^i(\epsilon)}{2}$, the RHS probability converges to 0, thus $d(\hat{\theta}_n, \theta_0) = o_p(1)$, hence $\hat{\theta}_n$ converges in probability to $\theta_0$ for each client $i \in [N]$.

$\square$

## C   LIMIT POINTS OF FEDGDA

We first discuss the $\gamma$- FEDGDA flow.

## C.1   FEDGDA FLOW

The FEDGDA updates can be written as

$$\theta_{t+1} = \theta_t - \eta\frac{1}{\gamma}\frac{1}{N}\sum_{i\in[N]}\sum_{r=1}^{R}\left(\nabla_\theta U_{\tilde{\theta}}(\theta_t, \tau_t) + (\nabla_\theta U_{\tilde{\theta}}^i(\theta_{t,r}^i, \tau_{t,r}^i) - \nabla_\theta U_{\tilde{\theta}}^i(\theta_t, \tau_t))\right.$$

$$\left. + (\nabla_\theta U_{\tilde{\theta}}^i(\theta_t, \tau_t) - \nabla_\theta U_{\tilde{\theta}}(\theta_t, \tau_t)))\right.$$

$$\tau_{t+1} = \tau_t + \eta\frac{1}{N}\sum_{i\in[N]}\sum_{r=1}^{R}\left(\nabla_\tau U_{\tilde{\theta}}(\theta_t, \tau_t) + (\nabla_\tau U_{\tilde{\theta}}^i(\theta_{t,r}^i, \tau_{t,r}^i) - \nabla_\tau U_{\tilde{\theta}}^i(\theta_t, \tau_t))\right.$$

$$\left. + (\nabla_\tau U_{\tilde{\theta}}^i(\theta_t, \tau_t) - \nabla_\tau U_{\tilde{\theta}}(\theta_t, \tau_t)))\right.$$

Rearranging the terms and taking the continuous-time limit as $\eta \to 0$

$$\lim_{\eta\to 0}\frac{\theta_{t+1} - \theta_t}{\eta} = \lim_{\eta\to 0} -\frac{1}{\gamma}\frac{1}{N}\sum_{i\in[N]}\sum_{r=1}^{R}\left(\nabla_\theta U_{\tilde{\theta}}(\theta_t, \tau_t) + (\nabla_\theta U_{\tilde{\theta}}^i(\theta_{t,r}^i, \tau_{t,r}^i) - \nabla_\theta U_{\tilde{\theta}}^i(\theta_t, \tau_t))\right.$$

$$\left. + (\nabla_\theta U_{\tilde{\theta}}^i(\theta_t, \tau_t) - \nabla_\theta U_{\tilde{\theta}}(\theta_t, \tau_t)))\right.$$

$$\lim_{\eta\to 0}\frac{\tau_{t+1} - \tau_t}{\eta} = \lim_{\eta\to 0}\frac{1}{N}\sum_{i\in[N]}\sum_{r=1}^{R}\left(\nabla_\tau U_{\tilde{\theta}}(\theta_t, \tau_t) + (\nabla_\tau U_{\tilde{\theta}}^i(\theta_{t,r}^i, \tau_{t,r}^i) - \nabla_\tau U_{\tilde{\theta}}^i(\theta_t, \tau_t))\right.$$

$$\left. + (\nabla_\tau U_{\tilde{\theta}}^i(\theta_t, \tau_t) - \nabla_\tau U_{\tilde{\theta}}(\theta_t, \tau_t)))\right.$$

We obtain the gradient flow equations as

$$\frac{d\theta}{dt} = -\frac{R}{\gamma}\frac{1}{N}\sum_{i\in[N]}\left(\nabla_\theta U_{\tilde{\theta}}(\theta(t), \tau(t))\right) - \frac{R}{\gamma}\frac{1}{N}\sum_{i\in[N]}\left(\nabla_\theta U_{\tilde{\theta}}^i(\theta^i(t), \tau^i(t)) - \nabla_\theta U_{\tilde{\theta}}^i(\theta(t), \tau(t))\right)$$

$$-\frac{R}{\gamma}\frac{1}{N}\sum_{i\in[N]}\left(\nabla_\theta U_{\tilde{\theta}}^i(\theta(t), \tau(t)) - \nabla_\theta U_{\tilde{\theta}}(\theta(t), \tau(t))\right), \tag{41}$$

$$\frac{d\tau}{dt} = R\frac{1}{N}\sum_{i\in[N]}\left(\nabla_\tau U_{\tilde{\theta}}(\theta(t), \tau(t))\right) + R\frac{1}{N}\sum_{i\in[N]}\left(\nabla_\tau U_{\tilde{\theta}}^i(\theta^i(t), \tau^i(t)) - \nabla_\tau U_{\tilde{\theta}}^i(\theta(t), \tau(t))\right)$$

$$+ R\frac{1}{N}\sum_{i\in[N]}\left(\nabla_\tau U_{\tilde{\theta}}^i(\theta(t), \tau(t)) - \nabla_\tau U_{\tilde{\theta}}(\theta(t), \tau(t))\right). \tag{42}$$

Using Assumption 3

$$\left\|\frac{R}{\gamma}\frac{1}{N}\sum_{i\in[N]}(\nabla_\theta U_{\tilde{\theta}}^i(\theta(t), \tau(t)) - \nabla_\theta U_{\tilde{\theta}}(\theta(t), \tau(t)))\right\| \le \frac{R}{\gamma}\zeta_\theta$$

$$\left\|R\frac{1}{N}\sum_{i\in[N]}(\nabla_\tau U_{\tilde{\theta}}^i(\theta(t), \tau(t)) - \nabla_\tau U_{\tilde{\theta}}(\theta(t), \tau(t)))\right\| \le R\zeta_\tau$$

Thus,

$$\frac{R}{\gamma}\frac{1}{N}\sum_{i\in[N]}(\nabla_\theta U_{\tilde{\theta}}^i(\theta(t), \tau(t)) - \nabla_\theta U_{\tilde{\theta}}(\theta(t), \tau(t))) = \mathcal{O}\left(\frac{R}{\gamma}\zeta_\theta\right)$$

$$R\frac{1}{N}\sum_{i\in[N]}(\nabla_\tau U_{\tilde{\theta}}^i(\theta(t), \tau(t)) - \nabla_\tau U_{\tilde{\theta}}(\theta(t), \tau(t))) = \mathcal{O}(R\zeta_\tau)$$

Since $U_{\tilde{\theta}}^i$ is Lipschitz smooth by assumption 2, we have

$$\left\| \frac{R}{\gamma} \frac{1}{N} \sum_{i \in [N]} \left( \nabla_\theta U_{\tilde{\theta}}^i(\theta^i(t), \tau^i(t)) - \nabla_\theta U_{\tilde{\theta}}^i(\theta(t), \tau(t)) \right) \right\| \leq L \frac{R}{\gamma} \frac{1}{N} \sum_{i \in [N]} \|(\theta^i(t), \tau^i(t)) - (\theta(t), \tau(t)\|,$$

$$\left\| R \frac{1}{N} \sum_{i \in [N]} \left( \nabla_\tau U_{\tilde{\theta}}^i(\theta^i(t), \tau^i(t)) - \nabla_\tau U_{\tilde{\theta}}^i(\theta(t), \tau(t)) \right) \right\| \leq L R \frac{1}{N} \sum_{i \in [N]} \|(\theta^i(t), \tau^i(t)) - (\theta(t), \tau(t))\|.$$

Substituting these bounds into Equations (41) and (42), we obtain

$$\frac{R}{\gamma} \frac{1}{N} \sum_{i \in [N]} \left( \nabla_\theta U_{\tilde{\theta}}^i(\theta^i(t), \tau^i(t)) - \nabla_\theta U_{\tilde{\theta}}^i(\theta, \tau) \right) = \mathcal{O} \left( L \frac{R}{\gamma} \frac{1}{N} \sum_{i \in [N]} \|(\theta^i(t), \tau^i(t)) - (\theta(t), \tau(t)\| \right),$$

$$R \frac{1}{N} \sum_{i \in [N]} \left( \nabla_\tau U_{\tilde{\theta}}^i(\theta^i(t), \tau^i(t)) - \nabla_\tau U_{\tilde{\theta}}^i(\theta, \tau) \right) = \mathcal{O} \left( L R \frac{1}{N} \sum_{i \in [N]} \|(\theta^i(t), \tau^i(t)) - (\theta(t), \tau(t))\| \right).$$

Since the local update follows

$$\theta^i(t) = \theta(t) - \frac{\eta}{\gamma} \sum_{j=1}^R \nabla_\theta U_{\tilde{\theta}}^i(\theta_j^i(t), \tau_j^i(t)),$$

$$\tau^i(t) = \tau(t) + \eta \sum_{j=1}^R \nabla_\tau U_{\tilde{\theta}}^i(\theta_j^i(t), \tau_j^i(t)),$$

Using bounded gradient assumption, i.e. $\|\nabla_\theta U_{\tilde{\theta}}^i(\theta, \tau))\|^2 \leq G_\theta$ and $\|\nabla_\tau U_{\tilde{\theta}}^i(\theta, \tau))\|^2 \leq G_\tau$ for all $i$, as $\eta \to 0$ and $R$ is fixed and finite, the deviation $\|(\theta^i(t), \tau^i(t)) - (\theta(t), \tau(t))\|$ vanish, leading to

$$\frac{d\theta}{dt} = -\frac{1}{\gamma} R \nabla_\theta U_{\tilde{\theta}}(\theta(t), \tau(t)) + \mathcal{O} \left( \frac{R}{\gamma} \zeta_\theta \right),$$

$$\frac{d\tau}{dt} = R \nabla_\tau U_{\tilde{\theta}}(\theta(t), \tau(t)) + \mathcal{O}(R \zeta_\tau).$$

### C.1.1 PROOF OF PROPOSITION 1

**Proposition** (Restatement of Proposition 1). *Given the Jacobian matrix for $\gamma-$FEDGDA flow as*

$$\boldsymbol{J} = \begin{pmatrix} -\frac{1}{\gamma} R \nabla_{\theta\theta}^2 U_{\tilde{\theta}}(\theta, \tau) & -\frac{1}{\gamma} R \nabla_{\theta\tau}^2 U_{\tilde{\theta}}(\theta, \tau) \\ R \nabla_{\tau\theta}^2 U_{\tilde{\theta}}(\theta, \tau) & R \nabla_{\tau\tau}^2 U_{\tilde{\theta}}(\theta, \tau) \end{pmatrix},$$

*a point $(\theta, \tau)$ is a strictly linearly stable equilibrium of the $\gamma-$FEDGDA flow if and only if the real parts of all eigenvalues of $\boldsymbol{J}$ are negative, i.e., $\mathrm{Re}(\Lambda_j) < 0$ for all $j$.*

*Proof.* Considering the FEDGDA dynamics with step size $\eta$, the Jacobian matrix of this dynamic system is $\boldsymbol{I} + \eta \boldsymbol{J}$. The eigenvalues of $\boldsymbol{J}$ are $\Lambda_j$, thus the eigenvalues of $\boldsymbol{I} + \eta \boldsymbol{J}$ are $\{1 + \eta \Lambda_j\}$.

By definition, a fixed point $\boldsymbol{z}^\star$ of a dynamical system $\boldsymbol{w}$, such that $\boldsymbol{z}^\star = \boldsymbol{w}(\boldsymbol{z}^\star)$, is a strict linearly stable point if the spectral radius $\rho(\boldsymbol{J}(\boldsymbol{z}^\star)) < 1$, where $\boldsymbol{J}$ is the Jacobian matrix of $\boldsymbol{w}$. Therefore, $(\theta, \tau)$ is a strict linearly stable point if and only if $\rho(\boldsymbol{I} + \eta \boldsymbol{J}) < 1$, that is $|1 + \eta \Lambda_j| < 1$ for all $j$. When taking $\eta \to 0$, this is equivalent to $\mathrm{Re}(\Lambda_j) < 0$ for all $j$. $\qquad \square$

## C.2 Proof of Theorem 3

*Proof.* Let $\boldsymbol{A} = \nabla^2_{\theta\theta} U_{\tilde{\theta}}(\theta, \tau), \boldsymbol{B} = \nabla^2_{\tau\tau} U_{\tilde{\theta}}(\theta, \tau)$ and $\boldsymbol{C} = \nabla^2_{\theta\tau} U_{\tilde{\theta}}(\theta, \tau)$. Consider $\epsilon = \frac{1}{\gamma}$, thus for sufficiently small $\epsilon$ (hence a large $\gamma$), the Jacobian $\boldsymbol{J}$ of FEDGDA for a point $(\theta, \tau)$ is given as:

$$\boldsymbol{J}_\epsilon = R \begin{pmatrix} -\epsilon\boldsymbol{A} & -\epsilon\boldsymbol{C} \\ \boldsymbol{C}^\top & \boldsymbol{B} \end{pmatrix}.$$

Using Lemma 9, $\boldsymbol{J}_\epsilon$ has $d_1 + d_2$ complex eigenvalues $\{\Lambda_j\}_{j=1}^{d_1+d_2}$ such that

$$\begin{aligned} |\Lambda_j + \epsilon\mu_j| &= o(\epsilon) & 1 \leq j \leq d_1 \\ |\Lambda_{j+d_1} - \nu_j| &= o(1), & 1 \leq j \leq d_2, \end{aligned} \tag{43}$$

where $\{\mu_j\}_{j=1}^{d_1}$ and $\{\nu_j\}_{j=1}^{d_2}$ are the eigenvalues of matrices $R(\boldsymbol{A} - \boldsymbol{C}\boldsymbol{B}^{-1}\boldsymbol{C}^\top)$ and $R\boldsymbol{B}$ respectively.

We now prove the theorem statement:

$$\mathcal{L}oc\mathcal{M}inimax \quad \subset \quad \underline{\infty - \mathcal{F}\mathcal{G}\mathcal{D}\mathcal{A}} \quad \subset \quad \overline{\infty - \mathcal{F}\mathcal{G}\mathcal{D}\mathcal{A}} \quad \subset \quad \mathcal{L}oc\mathcal{M}inimax \quad \cup$$
$$\{(\theta, \tau) | (\theta, \tau) \text{ is stationary and } \nabla^2_{\tau\tau} U_{\tilde{\theta}}(\theta, \tau) \text{ is degenerate}\}.$$

By definition of $\lim\sup$ and $\lim\inf$, we know that $\underline{\infty - \mathcal{F}\mathcal{G}\mathcal{D}\mathcal{A}} \subset \overline{\infty - \mathcal{F}\mathcal{G}\mathcal{D}\mathcal{A}}$.

Now we show $\mathcal{L}oc\mathcal{M}inimax \subset \underline{\infty - \mathcal{F}\mathcal{G}\mathcal{D}\mathcal{A}}$. Consider a strict local minimax point $(\theta, \tau)$, then by sufficient condition it follows that:

$$\boldsymbol{B} \prec 0, \quad \text{and} \quad \boldsymbol{A} - \boldsymbol{C}\boldsymbol{B}^{-1}\boldsymbol{C}^\top \succ 0.$$

Thus, $R\boldsymbol{B} \prec 0$, and $R(\boldsymbol{A} - \boldsymbol{C}\boldsymbol{B}^{-1}\boldsymbol{C}^\top) \succ 0$, where $R$ is always positive. Hence, $\{\nu_j\}_{j=1}^{d_1} < 0$ and $\{\mu_j\}_{j=1}^{d_2} < 0$. Using equations 43, for some small $\epsilon_0 < \epsilon$, $\text{Re}(\Lambda_j) < 0$ for all $j$. Thus, $(\theta, \tau)$ is a strict linearly stable point of $\frac{1}{\epsilon}$-FEDGDA.

Now, we show $\overline{\infty - \mathcal{F}\mathcal{G}\mathcal{D}\mathcal{A}} \subset \mathcal{L}oc\mathcal{M}inimax \cup \{(\theta, \tau) | (\theta, \tau) \text{ is stationary and } \nabla^2_{\tau\tau} U_{\tilde{\theta}}(\theta, \tau) \text{ is degenerate}\}$. Consider $(\theta, \tau)$ a strict linearly stable point of $\frac{1}{\epsilon}$-FEDGDA, such that for some small $\epsilon$, $\text{Re}(\Lambda_j) < 0$ for all $j$. By equation 43, assuming $B^{-1}$ exists

$$R\boldsymbol{B} \prec 0, \quad \text{and} \quad R(\boldsymbol{A} - \boldsymbol{C}\boldsymbol{B}^{-1}\boldsymbol{C}^\top) \succeq 0.$$

Since, $R$ is positive, thus $\boldsymbol{B} \prec 0$, and $\boldsymbol{A} - \boldsymbol{C}\boldsymbol{B}^{-1}\boldsymbol{C}^\top \succeq 0$. Let's assume $\boldsymbol{A} - \boldsymbol{C}\boldsymbol{B}^{-1}\boldsymbol{C}^\top$ has 0 as an eigenvalue. Thus, there exists a unit eigenvector $\boldsymbol{w}$ such that $\boldsymbol{A} - \boldsymbol{C}\boldsymbol{B}^{-1}\boldsymbol{C}^\top \boldsymbol{w} = 0$. Then,

$$\boldsymbol{J}_\epsilon \cdot (\boldsymbol{w}, -B^{-1}C^\top \boldsymbol{w})^\top = R \begin{pmatrix} -\epsilon\boldsymbol{A} & -\epsilon\boldsymbol{C} \\ \boldsymbol{C}^\top & \boldsymbol{B} \end{pmatrix} \cdot \begin{pmatrix} \boldsymbol{w} \\ -B^{-1}C^\top \boldsymbol{w} \end{pmatrix} = \boldsymbol{0}.$$

Thus, $\boldsymbol{J}_\epsilon$ has 0 as its eigenvalue, which is a contradiction because for strict linearly stable point $\text{Re}(\Lambda_j) < 0$ for all $j$. Thus, $\boldsymbol{A} - \boldsymbol{C}\boldsymbol{B}^{-1}\boldsymbol{C}^\top \succ 0$. Hence, $(\theta, \tau)$ is a strict local minimax point.

Let $G : \mathbb{R}^d \times \mathbb{R}^k \to \mathbb{R}$ be the function defined as: $G(\theta, \tau) = \det(\nabla^2_{\tau\tau} U_{\tilde{\theta}}(\theta, \tau))$. Let's assume that $\nabla^2_{\tau\tau} U_{\tilde{\theta}}(\theta, \tau)$ is smooth, thus the determinant function is a polynomial in the entries of the Hessian, which implies that $G$ is a smooth function. Since $\nabla^2_{\tau\tau} U_{\tilde{\theta}}(\theta, \tau) = 0$ implies at least one eigenvalue of $\nabla^2_{\tau\tau} U_{\tilde{\theta}}(\theta, \tau)$ is zero, thus $\det(\nabla^2_{\tau\tau} U_{\tilde{\theta}}(\theta, \tau)) = 0$.

We aim to show that the set

$$\mathcal{A} = \{(\theta, \tau) \mid (\theta, \tau) \text{ is stationary and } \det(\nabla^2_{\tau\tau} U_{\tilde{\theta}}(\theta, \tau)) = 0\}$$

has measure zero in $\mathbb{R}^d \times \mathbb{R}^k$.

A point $q \in \mathbb{R}^d \times \mathbb{R}^k$ is a *regular value* of $G$ if for every $(\theta, \tau) \in G^{-1}(q)$, the differential $dG(\theta, \tau)$ is surjective. Otherwise, $q$ is a *critical value*.

The differential of $G$ is given by: $\nabla G(\theta, \tau) = \text{Tr}\left(\text{Adj}(\nabla^2_{\tau\tau} U_{\tilde{\theta}}) \cdot \nabla(\nabla^2_{\tau\tau} U_{\tilde{\theta}})\right)$. If $\det(\nabla^2_{\tau\tau} U_{\tilde{\theta}}(\theta, \tau)) = 0$, then the Hessian $\nabla^2_{\tau\tau} U_{\tilde{\theta}}$ is singular. This causes its adjugate matrix to lose rank, leading to a degeneracy in $\nabla G(\theta, \tau)$, making $dG(\theta, \tau)$ *not surjective*.

Thus, every $(\theta, \tau)$ satisfying $G(\theta, \tau) = 0$ is a critical point of $G$, meaning that $0$ is a *critical value* of $G$.

By Sard's theorem, the set of critical values of a smooth function has measure zero in the codomain. Since $G$ is smooth, the set of critical values of $G$ in $\mathbb{R}$ has measure zero. In particular, since $0$ is a critical value of $G$, the set: $G^{-1}(0) = \{(\theta, \tau) \mid \det(\nabla^2_{\tau\tau} U_{\tilde{\theta}}(\theta, \tau)) = 0\}$ has measure zero in $\mathbb{R}^{d+k}$.

Since the set of degenerate $\nabla^2_{\tau\tau} U_{\tilde{\theta}}(\theta, \tau)$ is precisely $G^{-1}(0)$, we conclude that Lebesgue measure$(\mathcal{A}) = 0$. Thus, the set of stationary points where the Hessian $\nabla^2_{\tau\tau} U_{\tilde{\theta}}(\theta, \tau)$ is singular has measure zero in $\mathbb{R}^d \times \mathbb{R}^k$. $\qquad\square$

**Lemma 8.** *(Zedek, 1965) Given a polynomial $p_n(z) := \sum_{k=0}^n a_k z^k$, where $a_n \neq 0$, an integer $m \geq n$ and a number $\epsilon > 0$, there exists a number $\delta > 0$ such that whenever the $m + 1$ complex numbers $b_k$, $0 \leq k \leq m$, satisfy the inequalities*

$$|b_k - a_k| < \delta \quad \text{for } 0 \leq k \leq n, \quad \text{and} \quad |b_k| < \delta \quad \text{for } n + 1 \leq k \leq m,$$

*then the roots $\beta_k$, $1 \leq k \leq m$, of the polynomial $q_m(z) := \sum_{k=0}^m b_k z^k$ can be labeled in such a way as to satisfy, with respect to the zeros $\alpha_k$, $1 \leq k \leq n$, of $p_n(z)$, the inequalities*

$$|\beta_k - \alpha_k| < \epsilon \quad \text{for } 1 \leq k \leq n, \quad \text{and} \quad |\beta_k| > 1/\epsilon \quad \text{for } n + 1 \leq k \leq m.$$

**Lemma 9.** *For any symmetric matrix $\boldsymbol{A} \in \mathbb{R}^{d_1 \times d_1}$, $\boldsymbol{B} \in \mathbb{R}^{d_2 \times d_2}$, any rectangular matrix $\boldsymbol{C} \in \mathbb{R}^{d_1 \times d_2}$ and a scalar $R$, assume that $\boldsymbol{B}$ is non-degenerate. Then, matrix*

$$R \begin{pmatrix} -\epsilon\boldsymbol{A} & -\epsilon\boldsymbol{C} \\ \boldsymbol{C}^\top & \boldsymbol{B} \end{pmatrix}$$

*has $d_1 + d_2$ complex eigenvalues $\{\Lambda_j\}_{j=1}^{d_1+d_2}$ with following form for sufficiently small $\epsilon$:*

$$|\Lambda_j + \epsilon\mu_j| = o(\epsilon) \qquad 1 \leq j \leq d_1$$
$$|\Lambda_{j+d_1} - \nu_j| = o(1), \qquad 1 \leq j \leq d_2,$$

*where $\{\frac{1}{R}\mu_j\}_{j=1}^{d_1}$ and $\{\frac{1}{R}\nu_j\}_{j=1}^{d_2}$ are the eigenvalues of matrices $\boldsymbol{A} - \boldsymbol{C}\boldsymbol{B}^{-1}\boldsymbol{C}^\top$ and $\boldsymbol{B}$ respectively.*

The proof follows from Lemma 8 by a similar argument as in (Jin et al., 2020) with $\{\mu_j\}_{j=1}^{d_1}$ and $\{\nu_j\}_{j=1}^{d_2}$ as the eigenvalues of matrices $R(\boldsymbol{A} - \boldsymbol{C}\boldsymbol{B}^{-1}\boldsymbol{C}^\top)$ and $R\boldsymbol{B}$, respectively, and is thus omitted.

## D   RELATED WORK

The federated supervised learning has received algorithmic advancements guided by factors such as tackling the system and statistical heterogeneities, better sample and communication complexities, model personalization, differential privacy, etc. An incomplete list includes FEDPROX (Li et al., 2020), SCAFFOLD (Karimireddy et al., 2020), FEDOPT (Reddi et al., 2020), LPP-SGD (Chatterjee et al., 2024), PFEDME (T Dinh et al., 2020), DP-SCAFFOLD (Noble et al., 2022), and others.

By contrast, federated learning with confounders in a causal learning setting is a relatively under-explored research area. Vo et al. (2022a) presented a method to learn the similarities among the data sources translating a structural causal model (Pearl, 2009) to federated setting. They transform the loss function by utilizing Random Fourier Features into components associated with the clients. Thereby they compute individual treatment effects (ITE) and average treatment effects (ATE) by a federated maximization of evidence lower bound (ELBO). Vo et al. (2022b) presented another federated Bayesian method to estimate the posterior distributions of the ITE and ATE using a non-parametric approach.

Xiong et al. (2023) presented maximum likelihood estimator (MLE) computation in a federated setting for ATE estimation. They showed that the federated MLE consistently estimates the ATE parameters considering the combined data across clients. However, it is not clear if this approach is applicable to consistent local moment conditions estimation for the participating clients. Almodóvar et al. (2024) applied FedAvg to variational autoencoder (Kingma et al., 2019) based treatment effect estimation TEDVAE (Zhang et al., 2021). However, their work mainly focused on comparing the

performance of vanilla FedAvg with a propensity score-weighted FedAvg in the context of federated implementation of TEDVAE.

Our work differs from the above related works in the following:

(a) we introduce IV analysis in federated setting, and, we introduce federated GMM estimators, which has applications for various empirical research (Wooldridge, 2001),

(b) specifically, we adopt a non-Bayesian approach based on a federated zero-sum game, wherein we focus on analysing the dynamics of the federated minimax optimization and characterize the global equilibria as a consistent estimator of the clients' moment conditions.

Our work also differs from federated minimax optimization algorithms: Sharma et al. (2022); Shen et al. (2024); Wu et al. (2024); Zhu et al. (2024), where the motivation is to analyse and improve the non-asymptotic convergence under various analytical assumptions on the objective functions. We primarily focus on deriving the equilibrium via the limit points of the federated GDA algorithm.

# E    BENCHMARK CONSIDERATIONS AND ADDITIONAL EXPERIMENTS

## E.1    THE EXPERIMENTAL BENCHMARK DESIGN

As stated, our experiments take the Bennett et al. (2019)'s experiments as a centralized-setting baseline. Therefore, we have used the same synthetic dataset as DEEPGMM, which they use in their experiments to benchmarks against the baselines therein such as DEEPIV (Hartford et al., 2017). It is standard to perform experimental analysis on synthetic datasets for unavailability of ground truth for causal inference; for example see Section 4.1.1 of Vo et al. (2022b). As the learning process essentially involves estimating the true parameter $\theta_0$ by $\hat{\theta}$, to measure the performance of the learning procedure, we use the MSE of the estimate $\hat{g} := g(., \hat{\theta})$ against the true $g_0$ averaged over the clients. Nonetheless, an experimental comparison of our work with recent works on federated Bayesian methods for causal effect estimations does not apply directly. We discuss that below.

The two works in the domain of federated Bayesian methods for causal effect estimations are CAUSALRFF (Vo et al., 2022a) and FEDCI (Vo et al., 2022b). The aim of CAUSALRFF (Vo et al., 2022a) is to estimate the conditional average treatment effect (CATE) and average treatment effect (ATE), whereas FEDCI (Vo et al., 2022b) aims to estimate individual treatment effect (ITE) and ATE. For this, (Vo et al., 2022a) consider a setting of $Y$, $W$, and $X$ to be random variables denoting the outcome, treatment, and proxy variable, respectively. Along with that, they also consider a confounding variable $Z$. However, their causal dependency builds on the dependence of each of $Y$, $W$, and $X$ on $Z$ besides dependency of $Y$ on $W$. Consequently, to compute CATE and ATE, they need to estimate the conditional probabilities $p(w^i|x^i)$, $p(y^i|x^i, w^i)$, $p(z^i|x^i, y^i, w^i)$, $p(y^i|w^i, z^i)$, where the superscript $i$ represents a client. Their experiments compare the estimates of CATE and ATE with the Bayesian baselines (Hill, 2011), (Shalit et al., 2017), (Louizos et al., 2017), etc. in a centralized setting without any consideration of data decentralization or heterogeneity native to federated learning. Further, they compare against the same baselines in a *one-shot federated* setting, where at the end of training on separate data sources independently, the predicted treatment effects are averaged. Similar is the experimental evaluation of (Vo et al., 2022b). Similarly, Xiong et al. (2023) address a fundamentally different causal setting from ours, as they target ATE/ATT estimation in observational studies under the unconfoundedness assumption ($\{Y(0), Y(1)\} \perp W \mid X$), which implies that treatment assignment $W$ is exogenous given observed covariates $X$.

By contrast, the setting of IV analysis as in our work does not consider dependency of the outcome variable $Y$ on the confounder $Z$, though the treatment variable $X$ could be endogenous and depend on $Z$. In our synthetic data generation, an unobserved confounder explicitly enters both the treatment and outcome equations, inducing correlation between $(X)$ and the residual $(Y - g_0(X))$ and therefore violating unconfoundedness by construction. For us, computing the treatment effects and thereby comparing it against these works is not direct. Furthermore, it is unclear, if the approach of (Vo et al., 2022a) and (Vo et al., 2022b), where the predicted inference over a number of datasets is averaged as the final result, would be comparable to our approach where the problem is solved using a federated maximin optimization with multiple synchronization rounds among the clients. For us, the federated optimization subsumes the experimental of comparing the average predicted values after independent training with the predicted value over the entire data. This is the reason that

our centralized counterpart i.e. DEEPGMM (Bennett et al., 2019), do not experimentally compare against the baselines of (Vo et al., 2022a) and (Vo et al., 2022b). In summary, for us the experimental benchmarks were guided by showing the efficient fit of the GMM estimator in a federated setting.

### E.2 ADDITIONAL EXPERIMENTS

| Estimations | $Dir_S(\alpha) = 0.1$ | | $Dir_S(\alpha) = 1.0$ | |
|---|---|---|---|---|
| | FDEEPGMM-GDA | FDEEPGMM-SGDA | FDEEPGMM-GDA | FDEEPGMM-SGDA |
| **FEMNIST$_\mathbf{x}$** | $0.27 \pm 0.04$ | $0.23 \pm 0.02$ | $0.17 \pm 0.01$ | $0.19 \pm 0.03$ |
| **FEMNIST$_\mathbf{x,z}$** | $0.21 \pm 0.01$ | $0.24 \pm 0.04$ | $0.16 \pm 0.03$ | $0.18 \pm 0.02$ |
| **FEMNIST$_\mathbf{z}$** | $0.29 \pm 0.02$ | $0.25 \pm 0.03$ | $0.20 \pm 0.04$ | $0.23 \pm 0.01$ |
| **CIFAR10$_\mathbf{x}$** | $0.26 \pm 0.01$ | $0.27 \pm 0.01$ | $0.18 \pm 0.01$ | $0.15 \pm 0.02$ |
| **CIFAR10$_\mathbf{x,z}$** | $0.29 \pm 0.02$ | $0.30 \pm 0.01$ | $0.21 \pm 0.02$ | $0.13 \pm 0.01$ |
| **CIFAR10$_\mathbf{z}$** | $1.73 \pm 0.01$ | $0.67 \pm 0.02$ | $0.37 \pm 0.05$ | $0.35 \pm 0.02$ |

Table 2: The averaged Test MSE with standard deviation in the high-dimensional scenarios with varying levels of heterogeneity.

The experimental results included in Section 4 were conducted setting $Dir_S(\alpha) = 0.3$, which corresponds to the case wherein a dataset with 10 classes, such as MNIST and CIFAR10, samples of 3 classes on average will be distributed to each client (Hsu et al., 2019). To further investigate the effect of heterogeneity on the performance of FEDDEEPGMM, we conducted experiments with $Dir_S(\alpha) = 0.1$ and $Dir_S(\alpha) = 1$. $Dir_S(\alpha) = 0.1$ would correspond to the case when every client would have samples from one class on average from a dataset with 10 classes, which represents a high heterogeneity setting. Whereas, setting $Dir_S(\alpha) = 1$, the data distribution across clients with regards to samples from different classes becomes roughly uniform representing a near homogeneous scenario. The experimental results are presented in Table 2.

The results presented in Table 2 indicate that on decreasing $Dir_S(\alpha)$ from 0.3 to 0.1, i.e. increasing heterogeneity, the Test MSE achieved increases marginally. Whereas, on increasing $Dir_S(\alpha)$ from 0.3 to 1.0, i.e. decreasing heterogeneity, the Test MSE achieved decreases. This set of observations corroborate our theoretical insight that the consistency of the GMM estimator depends on the heterogeneity bias. The change in the MSE values being only marginal can be attributed to the overparametrized setting offered by the CNN on a small-sized data on each client as well as hyperparameter tuning.

