# OpenReview forum: "Federated Equilibrium Solutions for Generalized Method of Moments applied to Instrumental Variable Analysis"
_ICLR.cc/2026/Conference — ICLR 2026 Conference Withdrawn Submission_

### Official Review · Reviewer_G6pz · 2025-10-29

**Soundness:** 2
**Presentation:** 1
**Contribution:** 3
**Rating:** 2
**Confidence:** 2

**Summary:**

The authors introduce FedIV, a framework for federated instrumental variable (IV) analysis via a federated version of generalized method of moments (FedGMM). They also introduce FedDeepGMM to solve FedIV, which is a federated adaptation of DeepGMM from Bennett et al. (2019). The authors formulate the federated GMM problem as a non-convex, non-concave minimax optimization, modeled as a federated zero-sum game. They characterize equilibrium solutions using Stackelberg equilibria. Experiments on synthetic and image datasets (FeMNIST, CIFAR10) demonstrate convergence and consistency of the proposed method, even under non-i.i.d. client data distributions.

**Strengths:**

* The paper proposes federated GMM and federated IV analysis.
* The paper provides a theoretical framework that connects federated minimax optimization with Stackelberg equilibria.
* Extends classical GMM consistency results (Hansen, 1982; Bennett et al., 2019) to the federated setting, quantifying how heterogeneity affects consistency.
* Experiments demonstrate empirical convergence and estimation quality comparable to centralized DeepGMM, matching theoretical claims.

**Weaknesses:**

* While experiments confirm convergence, they are mostly illustrative. The scope of the experiments is quite limited, and there is no comparison with other FL causal methods such as the cited work of Xiong et al. (2023).
* The notation is not clear and even confusing sometimes, especially the probabilistic parts of the theory.
* Some gaps exist in the theoretical results. Especially, the proofs of Lemma 1 and Theorem 2 which are directly adapted from Bennett et al. (2019) to the federated setting.
* Justifications for some steps in several proofs are not explicitly given.

**Questions:**

1. Why didn't the authors compare their method with other existing FL causal methods such as Xiong et al. (2023)? At least under the conditions where these methods apply.
2. The paper doesn't specify which random variables the expected values are taken over, either in the main paper or in the proofs in the appendix. For example in Equation 4, is the expected value taken over $\epsilon_i$? What is $\mathbb{E}_{n_i}$ in Equation 6, and how is it equal to the empirical average?
3. In Lemma 1, the inverse of $C_{\tilde{\theta}}$ is taken without properly discussing that it is possible to do so. Also, defining the dual norm $\lVert \cdot \rVert_{*}$ based on $C_{\tilde{\theta}}^{-1}$ would require $C_{\tilde{\theta}}$ to be positive semi-definite. I couldn't find in the paper the proof of that or any assumption about it.
4. In the proof of theorem 1, line 911, does $\lambda_{\mathrm{max}}(\cdot)$ mean the largest eigenvalue in magnitude or largest eigenvalue as a real number? If it is the latter, then the equation in 911 does not necessarily hold unless the matrix is positive semi-definite. Am I missing something or misunderstanding the notation?
5. In the proof of Theorem 2, why does the inequality at line 1088 hold when you replace the expected value with the empirical average? Also, why does the inequality at line 1091 hold when you introduce the $\epsilon_k$? It could be much clearer if each step in the proof is clearly justified.
6. There is no proof of proposition 1 and the authors only cite Jin et al. (2020), but it should be stated how the eigenvalues of the Jacobian behave asymptotically ($\gamma \rightarrow \infty$).
7. Theorem 3 analyzes the deterministic continuous-time FedGDA flow but Algorithm 1 is based on discrete steps. The flow equation is obtained from the discrete equations ($\eta \rightarrow 0$) but not the other way around. The paper glosses over this.

I believe that the presentation needs serious enhancement overall, as my comments and questions imply. It would be easier if all equations are properly numbered so that the authors or even the reviewers can easily refer to them.

---

> ### Author Response · Authors · 2025-11-19
> **Response to concerns on experimental settings, notational clearity, etc.**
>
> We sincerely thank the reviewer for taking the time to thoroughly read our submission. We address the concerns raised.
>
> >While experiments confirm convergence.
>
> We indeed guessed that a reader/reviewer would ask this question. Considering this, we thoroughly discussed the choice of algorithms in our experimental benchmark in Appendix E1. Summarily, our method of federated causal inference via IV analysis fundamentally differs from a Bayesian method in that IV analysis does not offer a way to evaluate treatment effects in the same manner as Bayesian methods do, while still being applicable to many causal settings, as discussed in our introduction.
>
> >The notation is not clear and even confusing sometimes, especially the probabilistic parts of the theory. Some gaps exist in the theoretical results. Especially, the proofs of Lemma 1 and Theorem 2, which are directly adapted from Bennett et al. (2019) to the federated setting. Justifications for some steps in several proofs are not explicitly given.
>
> We appreciate your careful reading of the proofs. We will be happy and prompt to rectify and update if you could share specific points, without which, unfortunately, we are unsure how to address this concern. In any case, we will thoroughly proofread the theoretical results and will rectify every possible oversight in the revised version.
>
> >The paper doesn't specify which random variables the expected values are taken over, either in the main paper or in the proofs in the appendix. For example, in Equation 4, is the expected value taken over $\epsilon_i$? What is $\mathbb{E}_{n_i}$ in Equation 6, and how is it equal to the empirical average?
>
> Section 2 of our paper summarises the standard formulation of IV analysis, where the treatment variable $X$, the outcome variable $Y$, the instrumental variables $Z$, and the error term $\epsilon$ are all random variables. So the expectation in equation (4) is taken with respect to the joint distribution of $Z^i$ and $\epsilon^i$. $\mathbb{E}_{n_i}$ is the shorthand we used to define empirical mean. We acknowledge your concern here; although already mentioned, we will use the assignment operators for definitions in the revised version.
>
> > In Lemma 1, the inverse of $C_{\tilde\theta}$ ....
>
> As discussed in lines 150-154, the inverse of the empirical covariance matrix is used as the optimal weighting matrix for the weighted norm, which minimizes the asymptotic variance, as shown by Hansen (1982). The inverse of the empirical covariance matrix is the classic choice for optimally weighted GMM. The matrix $C_{\tilde{\theta}}$, as defined in equation (9), is an **empirical covariance matrix** of the moment functions:
> $$C_{\tilde{\theta}} = \frac{1}{n_i} \sum_{k=1}^{n_i}  f^i(Z_k^i) f^i(Z_k^i)\left(Y_k^i - g^i(X_k^i; \tilde{\theta})\right)^2.$$
> $C_{\tilde\theta}$ is positive semi-definite by its construction and thus $C_{\tilde\theta}^{-1}$ is positive semi-definite, where $\tilde\theta$ is a consistent estimator of $\theta_0$. In cases where the inverse does not exist, we take the pseudoinverse, such as the Moore-Penrose Pseudoinverse.
>
> > In the proof of Theorem 1... misunderstanding something?
>
> $\lambda_{max}(.)$ refers to the largest eigenvalue as a real number. The equation in line 911 holds regardless of the matrix being positive or negative (semi)-definite.
>
> Consider the matrix, $A:=\nabla_{\tau\tau}^2U_{\tilde\theta}^i(\hat\theta,\hat\tau)-\nabla_{\tau\tau}^2U_{\tilde\theta}(\hat\theta,\hat\tau)$, the referred inequality is: $\lambda_{\max}(A)\le\max\bigl(|\lambda_{\max}(A)|,|\lambda_{\min}(A)|\bigr)$. Now, the following cases hold:
>
> a) If $\lambda_{max}(A)\ge 0$ and $\lambda_{min}(A)\ge 0$, then $\lambda_{max}(A)\le\lambda_{max}(A)$ holds trivially.
>
> b) Else if $\lambda_{max}(A)\ge 0$ and $\lambda_{min}(A)< 0$, then for $|\lambda_{max}(A)|\ge|\lambda_{min}(A)|$, it holds trivially. Otherwise for $|\lambda_{max}(A)|<|\lambda_{min}(A)|$, then RHS=$|\lambda_{min}(A)|$, thus the inequality holds.
>
> c)  Else for $\lambda_{max}(A)<0$, $\lambda_{min}(A)<\lambda_{max}(A)<0$, thus it holds that $\lambda_{max}(A)\le \vert\lambda_{min}(A)\vert$.
>
> Hence, the proof of the referred inequality is correct.

---

> ### Author Response · Authors · 2025-11-19
> **Response continued.**
>
> >In the proof of Theorem 2, why does the inequality at line 1088 hold when you replace the expected value with the empirical average? Also, why does the inequality at line 1091 hold when you introduce the $\xi_k$? It could be much clearer if each step in the proof were clearly justified.
>
> **1. How is the second inequality obtained?**: The second inequality uses a **standard symmetrization argument** from statistical learning theory; see Theorem 1 of Duchi (2017). Since the expectation of the function is not computable directly, it is replaced with an **empirical average over an independent "ghost" sample**. This lets us bound the deviation between the empirical average and the true expectation by comparing two independent samples — the original and the ghost sample. This step transforms the problem into one where we now compare two empirical quantities instead of an empirical and a theoretical one, making the analysis tractable.
>
> **2. How is the third inequality obtained?**: The third inequality introduces **Rademacher random variables** $\xi_k$, where $\xi_k\in\{1,-1\}$ are i.i.d. for each client i, with probability $\frac{1}{2}$ for both, thus $\mathbb{E}[\xi_k]=0$. This trick is illustrated in the proof of Theorem 1 in Duchi (2017). The key idea is: instead of directly analyzing the difference between the two functions, we multiply their difference by random ±1 signs and take the expectation over those signs. This doesn't change the expectation but makes it easier to bound using standard tools from learning theory.
>
> (Duchi 2017) John Duchi, Lecture notes on CG Classes, Symmetrization, Subgaussian Processes and Chaining, Stats 300b: Theory of Statistics of Stanford University, 2/14/2017
>
> > There is no proof of proposition 1 and the authors only cite Jin et al. (2020),..
>
> We have included the derivation of the Jacobian matrix for the FedGDA-flow in Appendix C.1. The proof of Proposition 1 thus follows exactly the same steps as those given in Proposition 25 by Jin et al. (2020). We did not include it due to redundancy and directed an interested reader to see the proof in Jin et al. (2020).
>
> >Theorem 3 analyzes the deterministic continuous-time FedGDA flow, but Algorithm 1... glosses over this.
>
> Theorem 3 establishes the connection between the limit point of the FedGDA flow and the local minimax points. For limit-point derivations, an asymptotic analysis is a standard approach. In our case, this translates to the continuous-time fedGDA flow from the discrete FedGDA algorithm, which is sufficiently elaborated in Appendix C.1. We followed the techniques used by Jin et al. (2020). We are unsure what you mean by the other way around.
>
>
> We are hopeful that we have addressed your concerns. We will be glad to respond promptly should you have further queries. If you have no further queries, we kindly request that you review your score.

---

> ### Comment · Reviewer_G6pz · 2025-11-20
>
> Thank you for your response. I will only review my score when I see a satisfactory revision of the manuscript, and when I see how the other reviewers respond to your rebuttal. You have not addressed all my original concerns. I'll restate my concerns in the original order as in the Questions section of my review.
>
> 1. In the Appendix E.1, you do not discuss about the method of Xiong et al. (2023). I'll ask you a more specific question: Why did you not run the federated ATE/ATT estimator from Xiong et al. (2023) on your synthetic data? If this is not possible it should be clearly explained why that is the case.
>
> 2. Your explanation about which variables the expectations are taken over, should be clearly stated in the manuscript when it is introduced (e.g. Equation 4).
>
> 3. Thank you for explaining why $C_{\tilde{\theta}}$ is positive-semi definite. Please add this to the manuscript.
>
> 4. You're explanation about proof of theorem 1, line 911, is incorrect. Line 911 defines the spectral norm of $\lvert\lvert \nabla^2_{\tau} U_{\hat{\theta}}^{\,i}(\hat{\theta}, \hat{\tau}) - \nabla^2_{\tau} U_{\bar{\theta}}(\hat{\theta}, \hat{\tau}) \rvert\rvert_{\sigma} = \lambda_{\max}(\nabla^2_{\tau} U_{\hat{\theta}}^{\,i}(\hat{\theta}, \hat{\tau}) - \nabla^2_{\tau} U_{\bar{\theta}}(\hat{\theta}, \hat{\tau}))$, and you said $\lambda_{\max}$ is the max eigenvalue as a real number, which implies that this equation is incorrect, since the spectral norm is the maximum over all absolute eigenvalues. The eigenvalue $\lambda_{\max}$ can be negative and this contradicts the definition of the spectral norm.
>
> 5. The second inequality on line 1088 is clearly not justified. You can't just replace an expectation with an empirical average. The Duchi (2017) reference does not replace an expectation with an empirical average in the proof of Theorem 1.1, they clearly use linear properties of the expectations and apply the law of total expectation to get rid of the nested expectation. In Mathematics, one can not simply justify a step in a proof because it bears resemblance to another proof.
>
> 6. I agree that the proof is similar and short, but for theoretical completeness I suggest that you add the proof at least to the appendix.
>
> 7. I meant the analysis of the discrete-time algorithm (Algorithm 1). I apologize, I just realized that I haven't seen Theorem 4 initially. However, I couldn't find any proof of Theorem 4 in the manuscript.
>
> > We appreciate your careful reading of the proofs. We will be happy and prompt to rectify and update if you could share specific points, without which, unfortunately, we are unsure how to address this concern. In any case, we will thoroughly proofread the theoretical results and will rectify every possible oversight in the revised version.
>
> Regarding justifications, you can see points 3, 4, and 5 again. However, they are not the only ones, but are the most obvious points that were not justified. I can give you additional points as you requested:
> * Line 762, there should be a square root for the dual norm.
> * Regarding the previous points, the last two lines in Equation 23.
> * Line 775 The Lagrange multiplier for a $\leq$ constraint is supposed to satisfy $\lambda \geq 0$ not $\lambda \leq 0$ unless I'm missing something.
> * Line 778-779 Strong duality is claimed via Slater’s condition with $v = 0$; however, the argument that strong duality holds for a general nonconvex functional optimization is not justified. The mapping to a dual problem requires convexity in $v$, which is not discussed.
> * Line 812, how $v_j^{i} = 0$ if $c \neq i$?
> * Line 839: Riesz Representation applies to Hilbert spaces, but your $\mathcal{F}$ is finite-dimensional but not explicitly equipped with an inner product.
> * Line 883 requires a missing step from the prior inequalities to be fully justified.
>
> The list still can be longer, and my argument is **not about listing every single detail here**. I am trying to tell the authors that theoretical part of the paper **needs a more careful rewrite**. As, I am clearly not able to judge the soundness of the theoretical claims in the current state of the manuscript.
>
> Further, I agree with a point raised by reviewer qJEf, that is the paper needs to be more accessible to a wider audience. Not only to the mathematically inclined.

---

> > ### Author Response · Authors · 2025-11-30
> > **Response the follow-up concerns**
> >
> > Dear Reviewer G6pz,
> >
> > We really appreciate your follow-up remarks. We have now updated our submission. To summarize the updates,
> > + The updated text is colored blue.
> > + In Sections 2 and 3, we have now tailored the terms to the motivation example of global-scale FL for the COVID-19 mortality prediction model. We have also clarified the positive semi-definiteness of the covariance matrix.
> > + We have updated the section on the open problem for federated mix-strategy equilibrium, clarifying the constraints and implications of such a solution.
> > + In Appendix A.1., we have given a detailed convergence proof for the FedGDA algorithm: Theorem 4, for a self-contained reading.
> >
> > Furthermore, below we address your specific concerns:
> >
> > > In Appendix E.1, you do not discuss the method of Xiong et al. (2023). I'll ask you a more specific question: Why did you not run the federated ATE/ATT estimator from Xiong et al. (2023) on your synthetic data? If this is not possible it should be clearly explained what that is the case.
> >
> > The method of Xiong et al. (2023) addresses a fundamentally different causal setting from ours, as it targets ATE/ATT estimation in observational studies under the **unconfoundedness** assumption, $ ( {Y(0), Y(1)} \perp W \mid X )$, which implies that treatment assignment $(W)$ is exogenous given observed covariates $(X)$. In contrast, FedDeepGMM is specifically designed for **endogenous treatment scenarios**, where this assumption is violated, and identification is achieved through valid instrumental variables satisfying $(E[\varepsilon \mid Z] = 0 )$ and corresponding moment conditions.
> >
> > In our synthetic data generation, an unobserved confounder explicitly enters both the treatment and outcome equations, inducing correlation between $(X)$ and the residual $(Y-g_0(X))$ and therefore violating unconfoundedness by construction. Consequently, the estimator proposed by Xiong et al. is not applicable under our data-generating process, as its core identification assumption does not hold.
> >
> > >Your explanation about which variables the expectations are taken over should be clearly stated in the manuscript when it is introduced (e.g. Equation 4).
> >
> > We have revised the section to enhance clarity regarding the random variables and their expectations.
> >
> > > Thank you for explaining why $C_{\tilde{\theta}}$ is positive-semi definite. Please add this to the manuscript.
> >
> >  Done.
> >
> >  > Your explanation about proof of theorem 1, line 911, is incorrect...
> >
> > We are grateful to you for bringing this to our attention. It was mainly a typo error, and we have edited it out. We did not intend to define the spectral norm by the maximum eigenvalue. The proof stands correct once the typo is corrected.
> >
> >  > The second inequality on line 1088 is clearly not justified. You can't just replace an expectation with an empirical average. The Duchi (2017) reference does not replace an expectation with an empirical average in the proof of Theorem 1.1, they clearly use linear properties of the expectations and apply the law of total expectation to get rid of the nested expectation. In Mathematics, one can not simply justify a step in a proof because it bears resemblance to another proof.
> >
> > Please note that for $m^i(\theta,\tau,\tilde{\theta}_n)$, we have replaced it with its empirical counterpart $m^i_k(\theta,\tau,\tilde{\theta}_n)$ and then used a ghost variable $\tilde{\theta}_n^\prime$. Therefore, we have replaced the true expectation with the empirical average. This is a standard technique in Rademacher complexity.
> >
> > For reference, please see Step 3 in Section 7.1 of Qiao (2022) on Rademacher complexity.
> >
> > (Xingye Qiao, 2022) Theory of Machine Learning, Math 605 (Spring 2022), Binghamton.
> >
> >  > I agree that the proof is similar and short, but for theoretical completeness I suggest that you add the proof at least to the appendix.
> >
> >  We have added the proof in Appendix C1.1.
> >
> >  > I meant the analysis of the discrete-time algorithm (Algorithm 1). I apologize, I just realized that I haven't seen Theorem 4 initially. However, I couldn't find any proof of Theorem 4 in the manuscript.
> >
> > Theorem 4 has been adapted for the deterministic version of LocalSGDA (Sharma et al., 2022), which we call FedGDA with bounded gradient variance $\sigma=0$. We have included the proof in Appendix A1.
> >
> > Pranay Sharma, Rohan Panda, Gauri Joshi, and Pramod Varshney. Federated minimax optimization:
> > Improved convergence analyses and algorithms. ICML 2022.
> >
> > >Regarding justifications, you can see points 3, 4, and 5 again. However, they are not the only ones, but are the most obvious points that were not justified. I can give you additional points as you requested:
> >
> > > Line 762, there should be a square root for the dual norm.
> >
> > Thanks for correcting us; it was a typo. We have made the changes in the manuscript.
> >
> > > Regarding the previous points, the last two lines in Equation 23.
> >
> > After correcting the previous typo, Equation 23 is correct.

---

> > > ### Author Response · Authors · 2025-11-30
> > > **Response continued..**
> > >
> > > > Line 775 The Lagrange multiplier for a $\le$ constraint is supposed to satisfy $\lambda\ge 0$ not $\lambda\le 0$ unless I'm missing something.
> > >
> > > We would like to point out that the primal problem here is a maximization problem, and thus $\lambda\le 0$ is justified.
> > >
> > > > Line 778-779 Strong duality is claimed via Slater’s condition...
> > >
> > > Again, please note that the Lagrangian is convex and Slater's condition is justified. We have now explicitly added the reason in the proof. See lines 1177-1178.
> > >
> > > > Line 812, how $v_j^i=0$ if $c\neq i$?
> > >
> > >  We have defined $f^i$ in line 812 explicitly, so that $f^i$ does not depend on the moment conditions of other clients $f_j^c$ when $c\neq i$. Thus, we have defined $v_j^i=0$ whenever $c\neq i$.
> > >
> > > > Line 839: Riesz Representation applies to Hilbert spaces, but your $\mathcal{F}$ is finite-dimensional but not explicitly equipped with an inner product.
> > >
> > >  We agree that the Riesz Representation Theorem applies to Hilbert spaces equipped with an explicit inner product, and that our space $\mathcal{F}^i$ is only a finite-dimensional linear span of basis functions without an explicitly defined inner product. The intended step is therefore a finite-dimensional coordinate identification: any vector $v \in \mathbb{R}^m$ corresponds **uniquely** to a function $f^i \in \mathrm{span}(\mathcal{F}^i)$ via $f^i = \sum_{j=1}^m v_j^i f_j^i$ (see line 812).
> > >
> > > This establishes a linear isomorphism between $\mathbb{R}^m$ and $\mathrm{span}(\mathcal{F}^i)$, which is what we use in the proof. We have updated the manuscript to remove the reference to the Riesz Representation Theorem and replaced it with this finite-dimensional identification.
> > >
> > > We note that this mapping is analogous in spirit to the parameter–function representation used in neural networks, where a weight vector defines a function via a linear combination of basis (feature) functions, although we do not rely on any Hilbert space structure here. See the last paragraph in the proof of Lemma 1 in Appendix A of Benette et al. (2019).
> > >
> > > Benette et al.,  Deep Generalized Method of Moments for Instrumental Variable Analysis. NeurIPS 2019
> > >
> > > > Line 883 requires a missing step from the prior inequalities to be fully justified.
> > >
> > > We have corrected the proof in line 879 to justify line 883. Thanks again for pointing this out.
> > >
> > > > The list still can be longer...
> > >
> > > Thank you for this comment. We have carefully revised the theoretical section by correcting identified mistakes and making assumptions and justifications explicit to improve readability and verification. While there may still be room for minor refinement, we believe the revised version is now significantly clearer and more sound.
> > >
> > > We will be happy to address further concerns that the reviewer or area chair may have.

---

### Official Review · Reviewer_tJXT · 2025-10-31

**Soundness:** 2
**Presentation:** 3
**Contribution:** 2
**Rating:** 4
**Confidence:** 3

**Summary:**

This paper investigates Instrumental Variable (IV) analysis within a federated learning context. The authors state that there are currently no algorithms for federated Generalized Method of Moments or IV analysis. To address this, the paper introduces federated IV analysis by way of federated GMM (FEDGMM). This FEDGMM is formulated as a federated zero-sum game, which is a non-convex non-concave minimax optimization problem. The paper theoretically characterizes the solution to this federated game using Stackelberg equilibrium. It shows that this solution satisfies client-local equilibria up to a heterogeneity bias. The consistency of the federated GMM estimator across clients is then shown to depend on this bias.

**Strengths:**

1.	The paper addresses a novel and highly important problem. Applying IV analysis and GMM estimation in a privacy-preserving, federated setting opens up new possibilities for causal inference in sensitive domains like healthcare and economics, where data is decentralized and cannot be pooled.
2.	The paper frames the federated GMM problem as a non-convex non-concave minimax game. The subsequent analysis uses the concept of Stackelberg equilibrium to characterize the properties of the game's solution.

**Weaknesses:**

1.	The entire theoretical framework and the algorithm are analyzed under a "full client participation" setting (Line 194). This is a major limitation and is unrealistic for most practical FL systems, where client sampling (partial participation) is a defining and non-negotiable characteristic.
2.	The paper does not propose a new federated algorithm. The algorithm used for the analysis (FEDGDA) is a standard, synchronous federated adaptation of gradient descent-ascent. The contribution is thus limited to analyzing a standard algorithm in a new problem setting, rather than designing an algorithm that is robust to the specific challenges of this federated game (e.g., mitigating the heterogeneity bias, handling partial participation, or reducing communication).
3.	The experimental setup is insufficient. The experiments only compare the federated algorithms against their centralized counterparts, lacking comparisons to simpler federated alternatives. Furthermore, the validation relies on synthetic or simple image datasets (FEMNIST, CIFAR10) with artificial IV structures. The study should be extended to include more complex models and real-world datasets to demonstrate practical applicability.

**Questions:**

Please refer to _Weaknesses_.

---

> ### Author Response · Authors · 2025-11-19
> **Rebuttal on weaknesses**
>
> Thank you for your remarks. We are happy to address your concerns.
>
> >The entire theoretical framework and the algorithm are analyzed under a "full client participation" setting (Line 194). This is a major limitation and is unrealistic for most practical FL systems, where client sampling (partial participation) is a defining and non-negotiable characteristic.
>
> This is indeed a technical necessity. As responded to reviewer qJEf, we do not yet have a mixed-strategy solution. A pure strategy equilibrium solution for the federated minimax game can correspond to only a full client participation setting. Allowing for stochasticity, whether arising from stochastic gradients at clients or client sampling for each communication round, would necessitate accommodating a distribution over multiple actions at each communication round. Effectively, multiple sets of clients will become available for selection at each round, with a probability associated with each set. This game then no longer remains a pure strategy game, as the actions become non-deterministic.
>
> As mentioned in the paper, even if we adopt the notion of mixed strategy, to our knowledge, there is **no characterization of the federated mixed-strategy equilibrium solutions**. Therefore, we required the setting of full gradient and full client participation to understand IV analysis via FedDeepGMM. We emphasize that this is **purely a theoretical limitation**; our experiments include the results of FedSGDA used in our FedDeepGMM framework. Additionally, the benchmarks enable client selection from a large pool of clients.
>
> Several prior works on first theoretical analysis in federated optimization and federated minimax settings have also assumed full participation and full gradient computation to isolate and study core theoretical phenomena, e.g., Sharma et al. (2022) and Wu et al. (2023). To our knowledge, most standard centralized statistical machine learning algorithms have been adapted to federated settings with full client participation and full gradient updates, e.g., FedGMM (Federated Gaussian Mixture Model) (Wu et al., 2023), primarily due to foundational theoretical discussions. Indeed, this is a rapidly evolving area, and for us, the Federated Generalized Method of Moments with partial client participation remains future work, along with Federated Mixed-Strategy Equilibrium.
>
> References:
> (Sharma et al., 2022) Sharma, P., Panda, R., Joshi, G., and Varshney, P. (2022). Federated minimax optimization: Improved convergence analyses and algorithms. ICML, pages 19683–19730. PMLR.
>
> (Wu et al., 2022) Wu, X., Sun, J., Hu, Z., Zhang, A., and Huang, H. (2023). Solving a class of non-convex minimax optimization in federated learning. arXiv preprint arXiv:2310.03613.
>
> (Wu et al., 2023) Wu, Yue, et al. "Personalized federated learning under mixture of distributions." International Conference on Machine Learning. PMLR, 2023.
>
> > The paper does not propose a new federated algorithm. The algorithm used for the analysis (FEDGDA) is a standard, synchronous federated adaptation of gradient descent-ascent. The contribution is thus limited to analyzing a standard algorithm in a new problem setting, rather than designing an algorithm that is robust to the specific challenges of this federated game (e.g., mitigating the heterogeneity bias, handling partial participation, or reducing communication).
>
> We have clarified a similar misunderstanding regarding our key contributions noted by the reviewer qJEf. We kindly request that you read our rebuttal to qJEf.
>
> >The experimental setup is insufficient. The experiments only compare the federated algorithms against their centralized counterparts, lacking comparisons to simpler federated alternatives. Furthermore, the validation relies on synthetic or simple image datasets (FEMNIST, CIFAR10) with artificial IV structures. The study should be extended to include more complex models and real-world datasets to demonstrate practical applicability.
>
> We appreciate the reviewer's feedback regarding the experimental setup and would like to clarify the goals and scope of our empirical evaluation. We request that you review our benchmark selection as detailed in Appendix E.1. Reiterating, our benchmarks serve as proof of concept for the foundational existence results. We have presented experimental results that are consistent with the standard literature. We would be more than happy and prompt to implement a federated IV analysis for a real-life example if you could share some relevant references, even for the centralized setting.

---

### Official Review · Reviewer_qJEf · 2025-10-31

**Soundness:** 2
**Presentation:** 2
**Contribution:** 2
**Rating:** 2
**Confidence:** 3

**Summary:**

This paper introduces FEDIV, the first framework for federated instrumental variable (IV) analysis, and proposes a federated version of the Deep Generalized Method of Moments (DEEPGMM) algorithm, termed FEDDEEPGMM. The method formulates the federated GMM estimation problem as a non-convex non-concave minimax optimization—a federated zero-sum game—and characterizes its solution through Stackelberg (local minimax) equilibria.

Theoretical results establish that under bounded gradient and Hessian dissimilarity across clients, FEDDEEPGMM converges to an approximate federated equilibrium that ensures client-level consistency of GMM estimators, up to a heterogeneity bias. The paper also analyzes the relationship between the stable points of the federated gradient descent-ascent (FEDGDA) flow and the local minimax equilibria, showing their equivalence under smoothness conditions. Experiments on both synthetic (1D regression) and high-dimensional (FEMNIST, CIFAR-10) datasets demonstrate that FEDDEEPGMM achieves comparable or better estimation accuracy than centralized DEEPGMM baselines, even under non-i.i.d. data.

**Strengths:**

The paper is the first to explicitly tackle federated instrumental variable analysis, bridging causal inference, GMM estimation, and federated learning. Framing the problem as a federated minimax game is conceptually original and mathematically elegant.

The derivation of E-approximate federated equilibria and theorems establishing connections between equilibrium solutions, consistency, and heterogeneity are technically solid and carefully motivated. The proofs adapt and extend the theory of local minimax optimization (Jin et al., 2020) to the federated context.

**Weaknesses:**

1. While the technical contribution is significant, the motivation is weakly communicated. The paper assumes familiarity with both federated learning and econometric GMM/IV analysis, offering limited intuition on why federated IV analysis matters or what real-world settings require it (e.g., distributed healthcare causal studies). A short example motivating the use case (like the one in the introduction) could be more tightly tied to the technical results.

2. The main methodological component, FEDDEEPGMM, is a straightforward federated adaptation of DEEPGMM using existing FEDGDA optimization schemes. While the theoretical characterization is new, the algorithmic structure itself does not introduce a fundamentally new federated optimization mechanism.

3. The experiments mainly reproduce DEEPGMM setups under federated conditions. However, they do not deeply analyze the heterogeneity bias—the paper’s key theoretical insight. There is no quantitative or visual exploration of how bias or consistency deteriorates as client heterogeneity increases. This omission weakens the empirical validation of the central theoretical claim.

4. The computational and communication cost of running deep minimax optimization in a federated environment is non-trivial. The paper does not report runtime, convergence speed, or communication overhead compared to baselines such as FedAvg or SCAFFOLD.

5. The manuscript is highly mathematical, with minimal narrative or conceptual explanations. This makes it difficult for non-theoretical readers to grasp the significance of the results or their connection to causal inference practice.

**Questions:**

1. Could the authors provide quantitative experiments on how the heterogeneity bias (as defined in Theorem 1) affects the consistency of estimators? For instance, varying the Dirichlet α parameter and plotting estimation error vs. heterogeneity.

2. The current equilibrium analysis focuses on pure strategies; could the authors elaborate on the practical implications of the open problem of mixed-strategy equilibria in federated settings?

3. How sensitive is FEDDEEPGMM to communication frequency or local iteration count (R)? Some insight into optimization stability would be helpful.

4. Could the authors provide a simplified algorithmic summary or intuition—perhaps a diagram—of how the theoretical results (equilibrium → consistency) translate into practical algorithm behavior?

5. In the experiments, why do some high-dimensional cases (e.g., CIFAR10z) show large variance or poor performance? Is this related to curvature mismatch or gradient dissimilarity?

---

> ### Author Response · Authors · 2025-11-18
> **Rebuttal to concerns on weakly communicated motivation, new algorithmic contribution,  experiments**
>
> Thank you for taking the time to read our submission and noting that our derivation of E-approximate federated equilibria, as well as the theorems establishing connections between equilibrium solutions, consistency, and heterogeneity, are technically solid and carefully motivated. Below, we address your concerns.
>
> >While the technical contribution is significant, the motivation is weakly communicated. The paper assumes familiarity with both federated learning and econometric GMM/IV analysis, providing limited insight into why federated IV analysis is relevant or what real-world settings necessitate it (e.g., distributed healthcare causal studies). A short example motivating the use case (like the one in the introduction) could be more tightly tied to the technical results.
>
> While we appreciate your concern for further motivation, we are not sure what is meant by "more tightly tied to the technical results". The theoretical results quantify the effects of heterogeneity on the consistency of the federated GMM estimators for the clients, in addition to characterizing their existence via limit points. Providing a proof of concept for the results is exactly the standard in the literature. We would appreciate it if you could clarify your question further, especially what you would like to see in the limited space of the paper.  Do you have any specific examples in mind from the related literature that you would like us to implement? We would be happy to address your concern if you provide a reference. In this context, we would greatly appreciate it if you could review Appendix E.1 of our submission, where we detail the selection of our benchmark.
>
> >The main methodological component, FEDDEEPGMM, is a straightforward federated adaptation of DEEPGMM using existing FEDGDA optimization schemes. While the theoretical characterization is new, the algorithmic structure itself does not introduce a fundamentally new federated optimization mechanism.
>
> Very humbly, this concern may have stemmed from overlooking the main contribution of our work and asking for something that we specifically highlighted, see lines 92 to 100, which was not our goal. We are unsure what is meant by 'straightforward' here. Are you indicating that proving the consistency of estimators on clients under heterogeneity is a straightforward adaptation of a centralized optimization solution, when the federated equilibrium is not even defined? Again, we would like to humbly state that from a GMM application's perspective, the entailment of a new optimization algorithm is somewhat stretched when it was previously unknown whether an existing basic federated minimax optimization method, i.e., FedGDA, could retrieve a consistent GMM estimator.
>
> We would like to request that you refer again to lines 92-100 to clarify any misunderstandings about our key contribution. Our focus is not to propose a new federated minimax algorithm; instead, we formalize the GMM problem for IV analysis in the federated context and show that it reduces to solving a non-concave, non-convex minimax federated optimization problem, as shown in Equation (17). This, in its own merit, is non-trivial, which can be amply realized as you go through the construction of the pipeline of results. The FedDeepGMM framework is agnostic to the use of any federated minimax algorithm. However, we conjecture that for a new federated minimax solver to solve the FedDeepGMM problem, characterizing its limit point will be a challenge in itself. Its convergence theory, though important, will always remain orthogonal to the context of a GMM estimate solver.
>
> >The experiments mainly reproduce DEEPGMM setups under federated conditions. However, they do not deeply analyze the heterogeneity bias—the paper’s key theoretical insight. There is no quantitative or visual exploration of how bias or consistency deteriorates as client heterogeneity increases. This omission weakens the empirical validation of the central theoretical claim.
>
> This comment appears arisen from overlooking the experimental results investigating the effects of heterogeneity, as presented in Appendix E. In the main paper, we use $Dir_S(\alpha)=0.3$. The results of Appendix E.2 use $Dir_S(\alpha)=0.1$ (highly hetterogenous) and $Dir_S(\alpha)=1$ (homogeneous)  setups. The results of Appendix E.2 show that on decreasing $Dir_S(\alpha)$ from 0.3 to 0.1, i.e., increasing heterogeneity, the Test MSE achieved increases marginally. Whereas, upon increasing $Dir_S(\alpha)$ from 0.3 to 1.0, i.e., decreasing heterogeneity, the Test MSE decreases. This set of observations corroborates our theoretical insight that the consistency of the GMM estimator depends on the heterogeneity bias. The marginal change in the MSE values can be attributed to the overparameterized setting provided by the CNN on a small dataset for each client, as well as hyperparameter tuning. In the final version, we will include these results in the paper's main body.

---

> ### Author Response · Authors · 2025-11-19
> **Rebuttal to concerns on computational and communication cost and mathematical flavor**
>
> > The computational and communication cost of running deep minimax optimization in a federated environment is non-trivial. The paper does not report runtime, convergence speed, or communication overhead compared to baselines such as FedAvg or SCAFFOLD.
>
> Very humbly, we have exactly implemented the minimax counterpart of FedAvg. As we pointed out, our contribution is not a new federated minimax optimization algorithm; instead, we proposed FedDeepGMM- a causal effect estimation framework via IV analysis. We have implemented Local GDA and Local SGDA (Deng \& Mahdavi, 2021) algorithms for FedIV as FedDeepGMM-GDA and FedDeepGMM-SGDA, respectively. The computation and communication complexity of these algorithms has been sufficiently explored in the literature (Deng \& Mahdavi, 2021; Sharma et al., 2022) and is not the focus of our work. Moreover, these algorithms are comparable to other federated minimax algorithms, as explored in the literature, and are distinct from federated minimization algorithms such as SCAFFOLD and FedAvg. We doubt that you would have anything new to appreciate if we implemented three different optimizers, which are not related to the core technical contribution. At the same time, we would urge you to kindly consider the limitations of space in a conference paper, which differ from those of a thesis or technical report.
>
>
> > The manuscript is highly mathematical, with minimal narrative or conceptual explanations. This makes it difficult for non-theoretical readers to grasp the significance of the results or their connection to causal inference practice.
>
> Thank you for the feedback. We acknowledge that the manuscript is mathematically rich. However, due to space limitations, we were unable to include further high-level takeaways at the expense of the core contributions. We indeed have plans to utilize the additional page available in the CR version to include a discussion on the theoretical results, providing connections to causal inference practice.
>
> To underscore from our submission, our theoretical findings are summarized as:
> * Theorem 1 establishes the condition under which the federated solution of the optimization problem (17) satisfies the approximate equilibrium condition for each client $i$ with approximation error $\varepsilon^i$. $\varepsilon^i$ depends on the bound on gradient and hessian dissimilarity. For a client with high heterogeneity, the federated solution fails to converge to an approximate equilibrium solution.
> * Subsequently, Theorem 2 formalizes the tradeoff between data heterogeneity and the consistency of the global solution for each client, and it depends on $\varepsilon^i$, see Remark 2. If the error $\varepsilon^i$ is large for a client $i$, then the solution may fail to be a consistent estimator for that client. With low heterogeneity, the solution is a consistent estimator across the clients.
> * Since Theorems 1 and 2 are based on the $\varepsilon$-equilibrium solutions for the federated minimax optimization problem (Equation 17), Theorem 3 shows that the limit points of FEDGDA are the local minimax solutions, and thereby the equilibrium solution of the federated zero-sum game, up to some degenerate case. Thus, making the discussion complete.
>
> In layman's terms, the goal of this work is to obtain a consistent GMM estimator for each client using only its local data within a federated learning framework. For example, consider the case where each client represents a hospital that holds private data on COVID-19 patients. Estimating the effect of oxygen therapy on patient outcomes using standard regression methods may lead to biased results due to confounding factors. To address this, we employ IV analysis using the GMM estimators. In this context, the HIF gene factor serves as a valid instrument for oxygen therapy. Since the data is distributed across hospitals, we use FedDeepGMM to train collaboratively on the decentralized datasets. This approach allows us to estimate the true causal effect of oxygen therapy by pooling information across hospitals while preserving data privacy. The federated estimator provides a consistent estimate of the true parameter at each client, with an estimation error that depends on the level of heterogeneity across client datasets. We have described this motivating example in the introduction section. Indeed, for us, maintaining the rigour of formal discussion was a concern on par with writing the applications, which is standard for submissions to ICLR in this domain; see (Bennett et al., 2019).
>
> (Bennett et al. 2019) Bennett, Andrew, Nathan Kallus, and Tobias Schnabel. "Deep generalized method of moments for instrumental variable analysis." NeurIPS 32 (2019).

---

> > ### Author Response · Authors · 2025-11-19
> > **Response to questions on heterogeneity bias, mixed-strategy equilibrium and sensitivity to number of local steps**
> >
> > >  Could the authors provide quantitative experiments on how the heterogeneity bias (as defined in Theorem 1) affects the consistency of estimators? For instance, varying the Dirichlet α parameter and plotting estimation error vs. heterogeneity.
> >
> > We responded above -- the test MSE values for datasets under varying heterogeneity are reported in Appendix E.2.
> >
> > > The current equilibrium analysis focuses on pure strategies; could the authors elaborate on the practical implications of the open problem of mixed-strategy equilibria in federated settings?
> >
> > This is an excellent question. First, let's revisit the purpose of FL from a client's/agent's perspective, as discussed in Murhekar et al. (2023). In FL, clients collaborate to solve local optimization problems more effectively than they could with their limited local data. Participation is thus utility-driven.
> >
> >
> > In a centralized setting, the mixed-strategy solution has been recently applied to adversarial fine-tuning for robustness (Zhong et al., 2023), where a mixed-strategy solution represents a probability distribution over parameter states and adversarial perturbations. Extending this idea to a federated setting, a mixed-strategy solution would correspond to a global distribution over the global model states and perturbations synchronized across clients. This could improve robustness compared to pure-strategy solutions by accounting for uncertainty across clients.
> >
> >
> > In the context of federated GMM estimation, a mixed-strategy solution would imply a distribution over GMM estimators rather than a single deterministic solution. For practical use, each client would still require a consistent local estimator derived from this global distribution. Achieving this raises several challenges:
> >
> > 1. **Characterizing a global mixed-strategy equilibrium** in a federated setting remains an open problem, especially when clients act strategically (e.g., based on their local utility function) and under statistical/system heterogeneity.
> > 2. **Quantifying the approximation error** between the global mixed-strategy solution and each client’s local mixed-strategy solution is non-trivial. In the pure-strategy case, we analyzed this approximation error in terms of the heterogeneity bias using the gradient and Hessian dissimilarities; a similar approach would be needed here, but for distributions over estimators, which is not direct (we do not have a methodology for that).
> >
> > The key limitation is that we currently lack both:
> >
> > * A federated algorithm that explicitly computes a mixed-strategy equilibrium, and
> > * A theoretical framework for analyzing the consistency and approximation error between probability distributions under client heterogeneity.
> >
> > Addressing these gaps would be a promising direction for future work in federated learning.
> >
> > (Murhekar et al. 2023) Murhekar, Aniket, et al. "Incentives in federated learning: Equilibria, dynamics, and mechanisms for welfare maximization."  NeurIPS 2023
> >
> > (Zhong et al. 2023) Zhong, Zhehua, Tianyi Chen, and Zhen Wang. "MAT: mixed-strategy game of adversarial training in fine-tuning." IJCAI. 2023.
> >
> > > How sensitive is FEDDEEPGMM to communication frequency or local iteration count (R)? Some insight into optimization stability would be helpful.
> >
> > FedDeepGMM is essentially the framework that gives equation (17). Solving the minimax optimization problem of equation (17) amounts to solving the federated GMM problem. The framework in itself is algorithm-agnostic; essentially, its sensitivity to communication frequency or local iteration count depends on the algorithm used. A higher number of local iterations increases the convergence error, as shown in Theorem 4 of Appendix A. This is a standard result in FL that increasing the local training steps increases the client drift. To demonstrate the effectiveness of our framework, we implemented the Local GDA and Local SGDA (Deng & Mahdavi, 2021) algorithms as FedDeepGMM-GDA and FedDeepGMM-SGDA, respectively. In the revised version, we will include this standard remark after Theorem 4.
> >
> > (Deng & Mahdavi 2021) Yuyang Deng and Mehrdad Mahdavi. Local stochastic gradient descent ascent: Convergence analysis and communication efficiency. 24th AISTATS, PMLR: Vol 130, 2021.

---

> ### Author Response · Authors · 2025-11-19
> **Response to questions on algorithmic summary and some experimental observations**
>
> > Could the authors provide a simplified algorithmic summary or intuition—perhaps a diagram—of how the theoretical results (equilibrium →consistency) translate into practical algorithm behavior?
>
> Thank you for this insightful question.
>
> To summarize the core insight: our goal is to estimate causal effects using IV analysis in a federated setting, where data is localized and private. We solve a federated minimax problem using **FedGDA**, which outputs a saddle point solution $(\hat\theta, \hat\tau)$.
>
> * **Theorem 3** shows that FedGDA converges to a saddle point of the **federated objective**, which corresponds to an equilibrium solution for the federated setup.
> * **Theorem 1** then quantifies how well this solution satisfies the equilibrium condition for **each individual client**, with error $\varepsilon^i$, which depends on gradient and Hessian dissimilarity- i.e., the degree of **client heterogeneity**.
> * **Theorem 2** links this error $ \varepsilon^i $ to the **consistency** of the estimator for each client. If heterogeneity is low, $\varepsilon^i$ is small, and the estimator remains consistent. If heterogeneity is high, the error grows, and consistency can fail.
> ---
>
> **Theory to Practice Flow**
>
> ```
> FedGDA → Federated Saddle Point $(\hat\theta,\hat\tau)$
>              ↓ (Theorem 1)
>     $\varepsilon^i$= Client-wise Equilibrium Approximation Error
>              ↓ (Theorem 2)
>    Consistency of Estimator for Client $i$
> ```
>
> ---
>
> This pipeline shows how the output of the algorithm (FedGDA) translates to client-level guarantees through the lens of equilibrium and heterogeneity. We will include this diagram and a brief explanation in the final version to better convey the practical implications of our theoretical results.
>
> > In the experiments, why do some high-dimensional cases (e.g., CIFAR10z) show large variance or poor performance? Is this related to curvature mismatch or gradient dissimilarity?
>
> The high variance on CIFAR10z (1.70$\pm$2.60) likely results from a combination of gradient dissimilarity and curvature mismatch across clients due to the high-dimensional model and non-i.i.d. data (Dirichlet $Dir_S(\alpha)=0.3$). Using Theorem 1, the gradient and Hessian differences directly affect the approximation error $\varepsilon^i$, which can lead to instability and degraded consistency (Theorem 2). However, we cannot attribute the variance conclusively to a single source.

---

### Author Response · Authors · 2025-12-04
**Summary**

Dear AC & Reviewers,

We appreciate your feedback and constructive discussion on our submission. We briefly summarize our contributions and the discussion regarding the concerns raised.

We propose FedDeepGMM- a framework for federated IV analysis via GMM. We establish that solving for a GMM estimator in a federated setting is equivalent to solving a federated minimax problem using **FedGDA**, which outputs a saddle point solution $(\hat\theta, \hat\tau)$. Our theoretical contributions are summarized as:

(1)  **Theorem 1** quantifies how well a federated minimax solution satisfies the equilibrium condition for **each individual client**, with error $\varepsilon^i$, which depends on gradient and Hessian dissimilarity- i.e., the degree of **client heterogeneity**.

(2) **Theorem 2** links this error $ \varepsilon^i $ to the **consistency** of the estimator for each client. If heterogeneity is low, $\varepsilon^i$ is small, and the estimator remains consistent. If heterogeneity is high, the error grows, and consistency can fail.

(3) **Theorem 3** shows that FedGDA converges to a saddle point of the **federated objective**, which corresponds to an equilibrium solution for the federated setup.

---

Reviewer **qJEF** notes that "*the paper is the first to explicitly tackle federated instrumental variable analysis, bridging causal inference, GMM estimation, and federated learning. Framing the problem as a federated minimax game is conceptually original and mathematically elegant.*"

The reviewer’s main concerns involved motivation, the novelty relative to existing federated minimax solvers, the empirical study of heterogeneity bias, communication/runtime overhead, and accessibility of the exposition. We clarified that the core contribution lies not in proposing a new optimizer, but in establishing the theoretical conditions under which **each client obtains a consistent GMM estimator** in a federated setting, and demonstrating that standard FedGDA provably recovers such equilibria for the FedDeepGMM formulation. We highlighted that Appendix E includes experiments quantifying heterogeneity effects. We also explained why comparing runtime to federated minimization methods (e.g., FedAvg/SCAFFOLD) is not meaningful in this context. Theorem 4 in Appendix A discusses the communication complexity of FedGDA for $\mu$-PL and non-concave objectives. We have included a discussion in the updated version on the practical implications and the open problem of the mixed-strategy equilibria in FL.

---

Reviewer **tJXT** remarks that "*the paper addresses a novel and highly important problem.*" The reviewer raises concern that the theoretical framework is analyzed under a full client participation setting. We clarify that this is a common practice in FL theory, which simplifies the analysis of complex problems. In our case, this is a technical necessity as allowing stochasticity would require analyzing the federated game under mixed strategies, which is challenging. We have discussed the potential challenges in the updated version. We also clarify that our experiments align with the standard benchmarks used in prior IV literature (Appendix E.1).

---

Reviewer **G6Pz** acknowledges the results on quantifying the effect of heterogeneity on the consistency of the FedDeepGMM estimator for each client, as well as the empirical validation demonstrating performance comparable to that of the centralized DeepGMM. The reviewer raised concerns regarding missing comparisons, clarity of presentation, and gaps in proof details. We have addressed these by: (i) clarifying why ATE/ATT estimators like Xiong et al. (2023) are not applicable under endogenous treatment in our benchmark, (ii) substantially revising notation and explicitly specifying expectations and PSD assumptions in the updated version, and (iii) adding the full convergence proof of FedGDA in Appendix A, correcting theoretical typos, and enhancing justification steps in Lemma 1, and including the proof of Proposition 1. These improvements significantly strengthen rigour and accessibility, and we are thankful to the reviewer for helping us improve the manuscript.

---

We appreciate the reviewers' engagement throughout the discussion and hope that the clarifications and revisions lead to a positive reassessment.

---

### Note · Authors · 2026-01-26

**Comment:**

The reviews did not reflect the paper's merit.

**Withdrawal Confirmation:**

I have read and agree with the venue's withdrawal policy on behalf of myself and my co-authors.

---

### Meta-Review · Area_Chair_Vb9c · 2026-01-02

**Summary:**

Reviewers agreed that the problem is novel and important, and overall the idea (and theory) was good.

However, there were concerns about
1. Sufficient motivation
2. Algorithm being too simple
3. Missing empirical analysis linking theory to experiments
4. Lack of computation and communication cost
5. Paper too mathematically written and hard to read
6. Lacking sufficient baselines
7. Assumption of full client participation
8. Missing theoretical steps / justification

In my opinion, many of these were sufficiently rebutted by the authors, and also, some of these points, I disagreed with being so important (eg 2, 7).

However, some major concerns remain in this version of the paper. In particular, point 3 (eg a few small experiments plotting consistency and linking to theory and MSE), and point 5 (I found the paper unnecessarily mathematically dense and difficult to go through). Some minor remaining concerns are point 6 (can do with more simple baselines) and point 4 (can provide a computation and communication cost analysis): these would improve the paper but are not the reasons for rejection.

**Reviewer Concerns:**

See meta-review summary. Going over in turn:
1. Sufficient motivation: I felt was mostly addressed, or rather, can be fixed in the writing easily (eg I would expand the COVID example in a few more lines, such as given during the rebuttal).
2. Algorithm being too simple: I do not think this is a weakness or concern.
3. Missing empirical analysis linking theory to experiments: still outstanding (the experiments in the Appendix are not sufficient, see meta-review summary for more detail).
4. Lack of computation and communication cost: a minor point but can be addressed by providing an analysis.
5. Paper too mathematically written and hard to read: this has been improved in the new version but still I found it hard to read.
6. Lacking sufficient baselines: a minor point but the experiments would be stronger with some simple baselines.
7. Assumption of full client participation: sufficiently rebutted.
8. Missing theoretical steps / justification: sufficiently rebutted.

**Reviewer Scores:**

qJEf: increase to 4 as some points rebutted but others remain.

tJXT: either remain at 4 or increase to 6.

G6pz: increase to 4 after the discussion, but that's assuming that the authors sufficiently rebutted the mathematical details, which after my read through (without a detailed read), seemed the case.

---

### Decision · Program_Chairs · 2026-01-26

Reject